# GSFLOW–GRASS v1.0.0: GIS-enabled hydrologic modeling of coupled groundwater–surface-water systems

G.-H. Crystal Ng[1,2], Andrew D. Wickert[1,2], Lauren D. Somers[3], Leila Saberi[1], Collin Cronkite-Ratcliff[4], Richard G. Niswonger[5], and Jeffrey M. McKenzie[3]

[1]Department of Earth Sciences, University of Minnesota, Minneapolis, Minnesota, USA
[2]Saint Anthony Falls Laboratory, University of Minnesota, Minneapolis, Minnesota, USA
[3]Department of Earth and Planetary Sciences, McGill University, Montreal, Quebec, Canada
[4]Geology, Minerals, Energy and Geophysics Science Center, United States Geological Survey, Menlo Park, California USA
[5]Earth Systems Modeling Branch, United States Geological Survey, Menlo Park, California, USA

*Correspondence to:* G.-H. Crystal Ng (gcng@umn.edu)

**Abstract.**

The importance of water moving between the atmosphere and aquifers has led to efforts to develop and maintain coupled models of surface water and groundwater. However, developing inputs to these models is usually time-consuming and requires extensive knowledge of software engineering, often prohibiting their use by many researchers and water managers, and thus reducing these models' potential to promote science-driven decision-making in an era of global change and increasing water-resource stress. In response to this need, we have developed GSFLOW–GRASS, a bundled set of open-source tools that develops inputs for, executes, and graphically displays the results of GSFLOW, the U.S. Geological Survey's coupled groundwater and surface-water flow model. In order to create a robust tool that can be widely implemented over diverse hydro(geo)logic settings, we built a series of GRASS GIS extensions that automatically discretizes a topological surface-water flow network that is linked with the underlying gridded groundwater domain. As inputs, GSFLOW–GRASS requires at a minimum a digital elevation model, a precipitation and temperature record, and estimates of channel parameters and hydraulic conductivity. We demonstrate the broad applicability of the toolbox by successfully testing it in environments with varying degrees of drainage integration, landscape relief, and grid resolution, as well as the presence of irregular coastal boundaries. These examples also show how GSFLOW–GRASS can be implemented to examine the role of groundwater–surface-water interactions in a diverse range of water resources and land management applications.

# 1 Introduction

Predicting and understanding the hydrologic impacts of climate, land use, and other natural and anthropogenic change is a scientific endeavor that is increasingly necessary to manage water resources. Addressing this need requires streamlined access to models that integrate surface and subsurface processes across a watershed. This integrated approach is required because traditional hydrologic models that focus only on a single component within a watershed cannot properly predict the effects of changing conditions and feedbacks across their boundaries. The widespread use of integrated models is stymied, however, by labor-intensive requirements for creating consistent sets of extensive model inputs, including the challenges of generating computationally robust surface and sub-surface model domains.

Driven by the growing recognition of tightly coupled groundwater and surface water dynamics and the need to evaluate and manage the two as a single resource (Winter et al., 1998), the United States Geological Survey (USGS) developed and released GSFLOW. This integrated hydrologic model couples the groundwater flow model MODFLOW with the rainfall–runoff model PRMS (Precipitation Runoff Modeling System) (Markstrom et al., 2008). Both MODFLOW (Harbaugh, 2005; Niswonger et al., 2011) and PRMS (Leavesley et al., 1983; Markstrom et al., 2015) are popular models with significant user bases. GSFLOW has been previously applied to various watersheds in the US, for example in California (Essaid and Hill, 2014), Wisconsin (Hunt et al., 2013), Pennsylvania (Galeone et al., 2016), and Oregon (Surfleet and Tullos, 2013; Gannett et al., 2017), as well as to applications outside of the US (e.g., Hassan et al., 2014; Tian et al., 2015).

The process of implementing GSFLOW includes many hurdles that require significant time and computational knowledge to overcome. GSFLOW is not "fully integrated" in the sense that it does not simultaneously solve surface and subsurface flow equations; instead it consists of an iterative coupling between MODFLOW and PRMS that requires nearly all the individual input files for each of the two original models as well as an additional GSFLOW-specific linkage file. While a fully integrated model may have all the input information streamlined into a small number of internally consistent and efficiently organized files, to run GSFLOW, the user bears the burden of generating a multitude of diversely formatted ASCII files and ensuring that they contain inputs that correctly correspond with each other and can produce convergent coupled simulations. Freely available USGS GUIs – ModelMuse (Winston, 2009) and the PRMS GUI (Markstrom et al., 2015) – and proprietary GUIs (mostly for MODFLOW) can help users separately develop inputs to the two individual base models but do not offer support for creating the GSFLOW linkage file. The company Earthfx (http://www.earthfx.com/) provides full GSFLOW support as part of their "VIEWLOG" package, designed for the environmental consulting industry. More openly accessible software endeavors have also improved the usability of integrated hydrologic models (Bhatt et al., 2014; Tian et al., 2016; Gardner et al., 2018), but the community still lacks a free and complete package spanning pre- to post-processing for heterogeneous surface and subsurface domains. This lack of support for developing integrated hydrologic models such as GSFLOW motivates our present work, which we anticipate will enable more widespread hydrologic modeling.

Our overarching goal is to develop a bundled package – "GSFLOW–GRASS" – to handle the complexity of the coupled GSFLOW model, thus tackling the grand challenge of accessibility plaguing many integrated modeling systems. We develop an integrated toolbox featuring fully automated, robust, and open-source codes that cover the entire model implementation

process within a consistent and efficient framework, from building topologically linked hydrologic domains and assembling model input parameters to visualizing model outputs. Our use of only free and open-source programming languages and software is a key feature of the toolbox's accessibility. Python scripts generate model input files and model output graphics, and extensions using the open-source GRASS GIS platform build topographically defined sub-watersheds linked to subsurface grid cells. Open-source software facilitates implementation of GSFLOW–GRASS by diverse academic, government, and individual entities, enables further community development of GSFLOW–GRASS, and aligns with the USGS's goal to make its resources publicly accessible.

Developing a fully automated toolbox that can be readily executed for diverse physical settings raises the key technical obstacle of how to robustly build stream networks and sub-basins linked to subsurface computational domains without labor-intensive user intervention. Whereas overland flow routing and the calculation of drainage basins from topography are standard GIS capabilities, our tool improves upon these by automatically building topologically structured vectorized drainage networks without manual corrections using a least-cost path approach (Metz et al., 2011), while also including information on adjacency and routing pathways through the network that is required by integrated hydrologic models. The main technical advancement of GSFLOW–GRASS is the development of streamlined GRASS GIS extensions that have passed a diverse range of stress tests, including steep to low-relief topographies, large and intricate to small and simple drainage systems, incomplete to full topographic drainage integration, and inland to coastal watersheds. These new capabilities enable rapid, automated delineation of surface-water drainage networks linked to subsurface domains across any generalized landscape and computationally feasible resolution within the range of scales permissible by GSFLOW. By doing this all within a framework that also includes open-source model input and post-processing tools, GSFLOW–GRASS presents a solution toward more accessible integrated hydrologic modeling.

## 2 Background

### 2.1 GSFLOW

GSFLOW simulates spatially distributed surface to subsurface water flow in a watershed using modified model codes from PRMS and MODFLOW. It is designed for simulations of watersheds with areas of a few square kilometers to several thousand square kilometers (Markstrom et al., 2008, p. 2). Although GSFLOW can run in modes equivalent to the stand-alone PRMS-IV model and the stand-alone MODFLOW model, only the "integrated" version is described here. Near-surface watershed processes within the shallow "soil zone," including evapotranspiration, infiltration, runoff, and interflow, are represented by the PRMS sub-component of GSFLOW. Groundwater flow below the "soil zone," including vertical soil water movement in the deeper unsaturated zone and saturated flow through horizontal aquifer layers, is represented by the MODFLOW sub-component. Streamflow and exchange between streams and underlying groundwater systems are also represented by the MODFLOW sub-component. We describe here the key features of GSFLOW in order to guide new users in implementing it and interpreting its results; Markstrom et al. (2008) document the full details of the model.

### 2.1.1 Domain discretization

GSFLOW adopts a hybrid spatial domain discretization approach (Figure 1) to establish its computational units. Stream segments are links in a river network that are used in both the PRMS and MODFLOW sub-components of GSFLOW (Figure 1A). Horizontally, the PRMS sub-component uses hydrological response units (HRUs) of any shape as its fundamental discretized unit (Figure 1B). These are used for calculations of the upper soil zone and the part of the surface not covered by the stream network. The MODFLOW sub-component uses rectangular grid cells for the deeper subsurface (Figure 1C) and to further discretize the stream network into reaches (Figure 1D). Establishing reaches as the fundamental unit of computation for the stream network instead of segments makes it possible to resolve fine spatial resolution groundwater-surface exchanges. Like MODFLOW grid cells, HRUs can be set to rectangles, but they are also commonly defined topologically to correspond to sub-basins, as they are in our approach (Figure 1). Model intercomparison projects have included both representatives that use gridded domains and those that use irregular domains (Reed et al., 2004; Maxwell et al., 2014). In general, gridded domains are easier to construct and extend readily to parallelized computational systems, and they allow flexible spatial specification of soil and land-cover heterogeneity. In contrast, ungridded domains, such as triangulated irregular networks (TINs) used in models including tRIBS (Vivoni et al., 2004) and PIHM (Qu and Duffy, 2007), can conform more efficiently to complex terrain. In the case of PIHM (Qu and Duffy, 2007), TINs were also implemented for better water balance performance through the mass-conserving finite volume method (LeVeque, 2002); further, nested TINs can provide efficient solutions when higher resolution is desired for certain target areas (Wang et al., 2018). Other hydrological models with ungridded domains use topographically defined sub-basins as efficient computational units, including SWAT (Arnold and Fohrer, 2005), SAC-SMA (Ajami et al., 2004), HEC-HMS (Feldman, 2000), and TOPNET (Bandaragoda et al., 2004).

Vertically, the PRMS sub-component of GSFLOW is discretized into conceptual shallow soil zone reservoirs, which do not correspond directly to physical locations within the soil column but are instead based on user-specified conceptual thresholds. Specifically, within an HRU, the "soil zone" is subdivided into three reservoir types – the capillary reservoir, gravity reservoir, and preferential-flow reservoir, which are filled in order of increasing water storage using efficient water-accounting calculations (Section 2.1.2) (Figure 2). Underlying the PRMS soil zone are MODFLOW grid cells representing the deeper unsaturated zone and the saturated zone. While grid cells have uniform horizontal discretization, vertical layer thicknesses can be variable in order to accommodate different hydrostratigraphy. To link the PRMS and MODFLOW grids, the user must define gravity reservoirs at each different intersection of an HRU and a grid cell (Figure 1D). The MODFLOW component of GSFLOW also relies on a user-specified stream network; stream segments represent tributaries, and the intersection of a stream segment with MODFLOW grid cells defines stream reaches (Figure 1A, D).

GSFLOW uses a daily computational time step for both the PRMS component and MODFLOW component. Flows are exchanged between each component at each time step. Multiple MODFLOW "stress periods" can be invoked to represent different subsurface boundary conditions within a simulation period, but their lengths must be integer days.

**A. Segments (MODFLOW)**
Links in river network

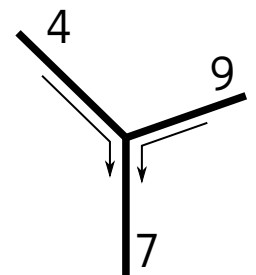

**B. HRUs (PRMS)**
Sub-catchments

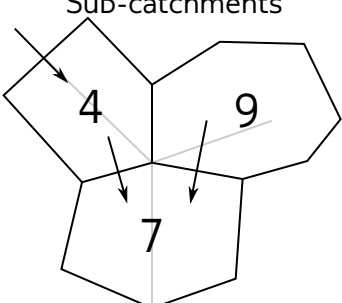

**C. Grid (MODFLOW)**
Finite difference grid

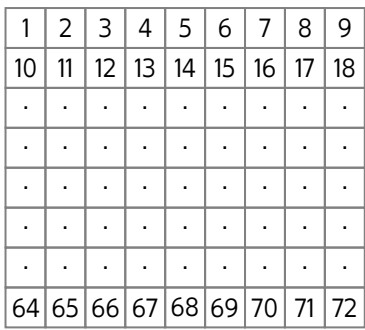

**D. Reaches & Grav. Res. (GSFLOW)**
Grid intersection of HRUs and segs.

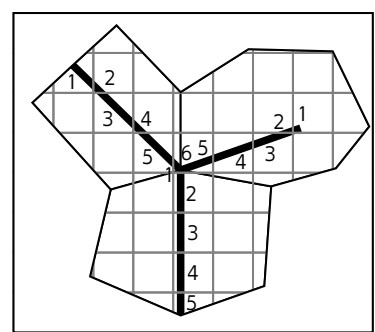

**Figure 1.** Major features of the GSFLOW geometry. **A.** Each segment is one link in the network. At each node, two tributary segments combine to flow into a single segment. Each is numbered. They need not be in any particular order, as indicated, but a downstream-increasing numbering scheme is required for updated inflows to all segments to be computed during the same iteration. **B.** Flow in each of the sub-basin HRUs is routed directly to a corresponding stream segment. The arrow on the upper left indicates that flow from outside of the representative tributary junction may also be part of the drainage network. Our topological approach to defining HRUs allows HRUs to be numbered the same as the stream segments that they enclose. Our code is written in such a way that future developments can relax this symmetry. **C.** MODFLOW operates on a grid that underlies the PRMS-based stream network and HRUs; each cell has a unique ID that is sequentially numbered. **D.** Gravity reservoirs are defined by the intersection of the PRMS HRUs and the MODFLOW grid. "Reaches" are defined as the section of each PRMS stream segment that lies within a single MODFLOW grid cell, and are numbered sequentially downstream as shown.

### 2.1.2 Process description

This section includes a brief description of the main hydrologic processes represented in GSFLOW, with select parameters listed in Table 1. Full details can be found in the GSFLOW manual (Markstrom et al., 2008). In particular, Table 1 from Markstrom et al. (2008) summarizes all the surface-water processes captured by PRMS modules, groundwater processes captured by MODFLOW stress packages, and model coupling procedures captured by GSFLOW.

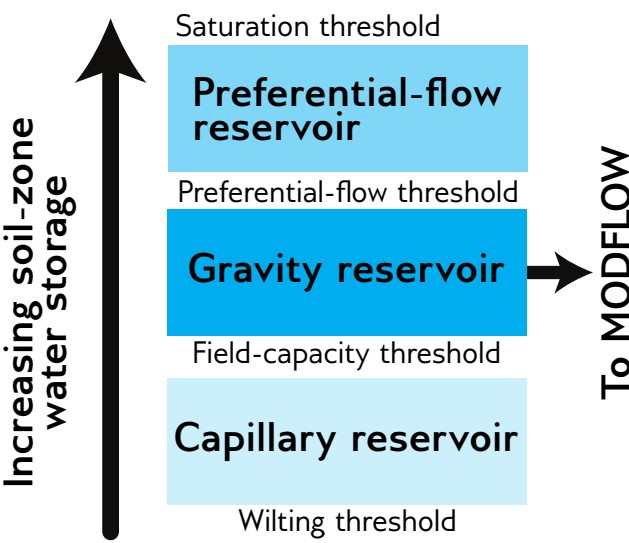

**Figure 2.** Soil-water storage reservoirs in the PRMS component of GSFLOW. Within each HRU, soil-water accounting calculations are carried out for three conceptual reservoirs in the order of increasing water storage and according to user-specified parameters. Climate forcing applies to the capillary reservoir, the gravity reservoir exchanges water with the deeper unsaturated and saturated zones represented by the MODFLOW component of GSFLOW, and Dunnian runoff and fast interflow occur in the preferential-flow reservoir. (Adapted from Markstrom et al., 2008, Figure 12.)

.

The PRMS component of GSFLOW includes modules that can convert commonly available climate data into complete forcing inputs needed for model simulations. These include methods for determining potential solar radiation, potential evapotranspiration, and snow accumulation or depletion; they also include different for spatially distributing data from one or a few observations points over the entire watershed.

5    For unsaturated zone flow, PRMS does not implement the Richards equation but instead applies computationally fast soil-water routing calculations to determine inputs and outputs for each HRU as well as exchanges among the three conceptual reservoir types within an HRU (GSFLOW manual Fig 19, Table 9). The "capillary zone" reservoir represents water held by capillary forces; it receives water through infiltration (based on parameter *pref_flow_den*) and loses water through evaporation and transpiration (based on parameters *soil_moist_max*, *soil_rechr_max*, and *soil_type*). After reaching field capacity (param-
10   eter *soil_moist_max*), water transfers from the capillary zone to "gravity reservoirs", where water can flow horizontally as slow interflow (based on parameters *slowcoef_lin* and *slowcoef_sq*) or drain vertically into the deeper subsurface domain that is handled by MODFLOW (based on parameters *ssr2gw_rate*, *ssr2gw_exp*, and *ssrmax_coef*). Gravity reservoirs can also receive groundwater discharge from the MODFLOW component when hydraulic head values exceed the lower limit of the soil zone. A fraction of gravity reservoir storage moves to the "preferential-flow reservoir" (based on parameters *pref_flow_den* and
15   *sat_threshold*), where fast interflow occurs (based on parameters *fastcoef_lin* and *fastcoef_sq*). If the preferential-flow reservoir becomes full (based on parameter *sat_threshold*), then water exits the soil zone as Dunnian (saturation-excess) runoff. Horto-

**Table 1.** Select GSFLOW parameters (adapted from Markstrom et al., 2008, Appendix 1).

| Parameter | Description |
|-----------|-------------|
| *pref_flow_den* | Decimal fraction of the soil zone available for preferential flow versus capillary zone flow |
| *soil_moist_max* | Maximum available capillary water-holding capacity of soil zone |
| *soil_rechr_max* | Maximum quantity of water in the capillary reservoir (value must be less than or equal to *soil_moist_max*) |
| *soil_type* | Soil type: 1=sand; 2=loam; 3=clay |
| *soil_moist_max* | Maximum volume of water per unit area in the capillary reservoir |
| *slowcoef_lin* | Linear flow-routing coefficient for slow interflow |
| *slowcoef_sq* | Non-linear flow-routing coefficient for slow interflow |
| *ssr2gw_rate* | Linear coefficient in the equation used to compute gravity drainage to MODFLOW finite-difference cell |
| *ssr2gw_exp* | Exponent in the equation used to compute gravity drainage to MODFLOW finite-difference cell |
| *ssrmax_coef* | Maximum amount of gravity drainage to MODFLOW finite-difference cell |
| *sat_threshold* | Maximum volume of water per unit area in the soil zone, between field capacity and saturation thresholds |
| *hru_percent_imperv* | Decimal fraction of HRU area that is impervious |
| ICALC | An integer value used to indicate method used to calculate stream depth in this segment |
| IRTFLG | An integer value that flags whether transient streamflow routing is active |

nian (infiltration-excess) runoff calculations apply for impervious fractions of HRUs (set by parameter *hru_percent_imperv*). Surface runoff and interflow are routed between HRUs, using a cascading flow scheme that follows user-specified indexing of linked HRUs, and eventually reaches the stream network.

The MODFLOW component of GSFLOW computes water flow in the deeper unsaturated zone (UZF stress package),
streams (SFR package), and saturated groundwater units (BCF, LPF, or UPW flow packages). Unsaturated zone flow is calculated using a kinematic-wave approach, which assumes that capillary (pressure gradient) flow is negligible compared to gravity-driven flow. Capillary-dominated effects are instead represented in the soil zone of the PRMS component described above. Unsaturated zone flow in the MODFLOW component is calculated as waves representing wetting and drying fronts. Gravity reservoir drainage from the PRMS component flows to the top of the unsaturated zone of the MODFLOW component,
unless the water table is above the soil-zone base – defined by the top of the MODFLOW domain – in which case the gravity reservoirs drain directly to the saturated zone. Saturated zone simulations (MODFLOW) employ the finite difference method to the groundwater flow equation.

Streamflow, as calculated by the MODFLOW component, includes inputs from upstream reaches, surface runoff and interflow from the PRMS component, base flow from the saturated zone discharge, and flows from possible underlying unsaturated
areas. Outputs include flow to downstream reaches, leakage to groundwater, and flows to possible underlying unsaturated areas. Discharge across the streambed follows Darcy's law with specified streambed hydraulic properties. Five different options exist for stream discharge and head computations (parameter *ICALC*). The user can specify stream depths for each reach; apply Manning's equation to an assumed wide rectangular channel; apply Manning's equation for an eight-point-based channel and

floodplain geometry; apply at-a-station power-law relationships between discharge, flow width, and flow depth (Leopold and Maddock, 1953); or specify an input look-up table of hydraulic geometries for each segment. Streamflow can be simulated as either steady-state flow (parameter *IRTFLG* = 0), where outflow to the next stream reach balances inputs, or as transient flow (parameter *IRTFLG* > 0), using a kinematic wave formulation for surface-water routing in channels, which applies the
assumption that the water surface slope approximates the friction slope, and therefore negates backwater effects.

  Some modifications were made to the original stand-alone PRMS and MODFLOW codes for their use in GSFLOW. Notably, the soil-zone structure of PRMS was significantly altered to facilitate its linkage with a MODFLOW subsurface domain. Other modifications are noted in the GSFLOW manual (Markstrom et al., 2008, see sections on "Changes to PRMS" and "Changes to MODFLOW-2005"). An additional feature starting in version 1.2.0 that is not described in the original manual is the inclusion
of MODFLOW-NWT (Niswonger et al., 2011), a more numerically robust update to MODFLOW-2005 (Harbaugh, 2005) for groundwater flow.

## 2.2 GRASS GIS

GRASS GIS is an open-source, multi-purpose, and cross-platform geographic information system (Neteler and Mitasova, 2008; Neteler et al., 2008, 2012) that supports utilities for efficient raster and vector computations (Shapiro and Westervelt, 1994;
Mitasova et al., 1995; Ŝúri and Hofierka, 2004; Hofierka et al., 2009). It includes both graphical and command-line interfaces, and may be driven by shell or Python scripts. It supports both 2D and 3D raster and vector data and includes SQL-based attribute table database management. GSFLOW–GRASS utilities are written for the most recent stable release version of GRASS GIS, v7.4. This supports Python scripting for both high-level built-in commands and for low-level access to database entries and vector geometries (Zambelli et al., 2013). We take advantage of these capabilities to develop an automated workflow to build
GSFLOW inputs through GRASS GIS.

  We chose GRASS GIS as the interface to develop inputs because (1) it is open-source and cross-platform; (2) it enforces rigid vector topology, which is critical for building stream networks; (3) its broad library of built-in hydrologic tools include those for vectorized drainage network development with downstream-increasing indexing (Jasiewicz and Metz, 2011), which is essential for setting flow paths and adjacencies; (4) its generic Python scripting library and PyGRASS Application Program-
ming Interface (API) make it easy to develop new extensions; (5) these extensions may be added to the official subversion (svn) repository, from which they can be automatically downloaded and installed on users' computers using the **g.extension** command; and (6) it provides a GUI and command-line interface (CLI) that are consistent with one another. The GUI and CLI interfaces are not required for GSFLOW–GRASS because the GRASS GIS component is handled mostly behind the scenes by a batch-processing Python script (**buildDomainGRASS.py**, Section 3.2); however, they allow end-users to re-run certain
portions of the process and/or produce their own workflows using the GSFLOW–GRASS extensions as building blocks. The open-source aspect of the present work is in part motivated by the need for water assessment and planning tools in the developing world (Pal et al., 2007), and these extensions, combined with the interchangeable and consistent GUI and CLI, can help users to generate their own advanced customizations of GSFLOW–GRASS.

## 3 Methods

We adopt a heterogeneous surface and subsurface computational domain for GSFLOW–GRASS that employs sub-basin surface HRUs that are linked to subsurface grid cells. In addition to the computational efficiency of discretizing complex terrain into sub-basins with complex shapes rather than using a gridded surface domain at the resolution required to resolve the HRUs, the use of sub-basin HRUs that route surface runoff directly to stream segments also eliminates the need for establishing a cascading network (Section 2.1.2). Because of GSFLOW's conceptual (rather than gradient-based) surface-water-routing scheme, numerical differences between sub-basin and gridded HRU's are difficult to predict, but the automated GSFLOW– GRASS toolbox can help enable future testing to rigorously interrogate their respective performances.

GSFLOW–GRASS strikes a balance between generating a ready-to-go GSFLOW implementation and providing flexibility to customize applications. With a newly developed set of automated and robust GIS domain builder tools, GSFLOW–GRASS can be applied to any digital elevation model (DEM) to produce GSFLOW model simulations. Only a few steps are required to set up a GSFLOW model on the user's computer system. For further model-tuning, all scripts in the toolbox are open-source and commented to allow changes to any parameter as well as development of optional GSFLOW capabilities not included in the default GSFLOW–GRASS implementation. Many popular hydrologic model implementation programs have GUIs, including ModelMuse (Winston, 2009), Visual Modflow (Waterloo Hydrogeologic Inc., 2011), Hydrus (Simunek et al., 2009), ArcSWAT (Neitsch et al., 2002), and MIKE-SHE (Butts and Graham, 2005). While these are easiest for novice model users, GUIs can be challenging to develop for cross-platform implementations and generally support less flexibility for customization. Thus, we chose a mostly command-line approach, which has been designed and tested for use on Linux and Windows operating systems.

### 3.1 User-specified settings and model inputs

To seamlessly unify the different GSFLOW–GRASS functionalities, including the automated GRASS GIS domain builder, GSFLOW input-file builder, and visualization components, users specify model inputs and configurations using a Settings text file. All inputs from the Settings file are read in and processed by the **ReadSettings.py** script. GSFLOW requires a daunting number of different model inputs (nearly 200 parameters for the PRMS sub-component alone). For ease of use, only a handful of application-specific and commonly adjusted inputs may be assigned using the Settings file, and default parameter values are applied elsewhere. While the default (and simplest) approach to GSFLOW–GRASS is to modify only the Settings file, other parameters (including those mentioned in Section 2.1.2) may be readily changed in its input-file builder by searching for the parameter names defined in the GSFLOW manual and changing their values. The open-source nature of our toolbox also allows users to add parameters to the Settings files for future extensions of GSFLOW–GRASS.

Specifying and including spatially variable properties is a major challenge to distributed modeling. The Settings file accommodates the use of variable aquifer hydraulic conductivity, channel width, and Manning's $n$ parameters, which are described further in Section 3.3.3. Universal solutions are beyond the scope of the default toolbox, but we do provide a generalized GRASS-GIS extension called **v.gsflow.mapdata** to facilitate the generation of heterogeneous model inputs. **v.gsflow.mapdata**, further described in Section 3.2.4, can take any spatially variable data in a raster or vector GIS format and map it to one of

the GSFLOW discretization structures: sub-basin HRUs for PRMS surface-water processes, regular grid cells for MODFLOW groundwater processes, gravity reservoirs that link the HRUs and MODFLOW grid cells, or stream segments or reaches for MODFLOW streamflow processes. This allows users to add data from any source – e.g., meteorological forcing, soil properties, hydrogeologic stratigraphy, or vegetation / land cover – to the GSFLOW–GRASS data structures. Other software tools have facilitated hydrologic modeling by automating the connection with established databases (Viger and Leavesley, 2007; Leonard and Duffy, 2013; Bhatt et al., 2014; Gardner et al., 2018). The USGS's GIS Weasel tool (Viger and Leavesley, 2007) may be used for deriving PRMS parameters from physical data sets such as STATSGO, which can then be mapped to the appropriate GSFLOW data structure using **v.gsflow.mapdata**. The current GSFLOW–GRASS release aims to provide a general set of tools and does not directly link with any specific databases, which are typically only available in observation-rich regions and countries.

The Settings file is divided into subsections, each of which drives a portion of the model setup and organization. The "paths" section defines the computer directory structure for the project and GSFLOW executable, as well as the project name and GSFLOW version. Three GRASS GIS sections, "GRASS_core", "GRASS_drainage", and "GRASS_hyrdaulics", set the GIS location and path to the DEM, the surface and subsurface flow discretization parameters, and open-channel flow geometry and resistance, respectively. The "run_mode" section allows the user to execute GSFLOW in either "spin-up" or "restart" mode (Regan et al., 2015). Spin-up simulations start with a preliminary MODFLOW steady-state execution using a specified infiltration rate (see below) to calculate reasonable initial groundwater head conditions for the subsequent transient simulation that includes both the surface and subsurface domains; the steady-state step can be essential for obtaining numerically convergent groundwater results and more realistic solutions for the entire coupled system. At the end of a spin-up run, final PRMS and MODFLOW state variables are saved in files that can be specified in the run_mode section to initiate "restart" coupled runs without the preliminary groundwater steady-state period. The "time" section is used to specify the temporal window of the simulation. The "climate inputs" section sets input parameters for the PRMS "climate_hru" option, which is the standard climate implementation supported by GSFLOW–GRASS (see Section 3.3.1) . Finally, the "hydrogeologic_inputs" section defines the preliminary steady-state MODFLOW infiltration rate, used for "spin-up" runs, and either a layered or fully distributed subsurface hydraulic conductivity structure. The **ReadSettings.py** script uses these inputs to create a directory structure and organize all GIS and simulations files. This imposed directory structure supports easy exchange between the different toolkit modules and allows the use of relative directory names, which facilitates the sharing of model files across computers systems and between users.

## 3.2  GRASS GIS domain builder

A critical challenge for any distributed hydrologic model is the fully automated development of a reproducible, topologically correct, and interlinked data structure that describes water flow through a catchment in a computationally efficient manner. Semi-automated approaches to building surface flow networks are common (e.g., Luzio et al., 2006; Arnold et al., 2012), but the development of a fully automated approach has been impeded by the mathematical and logistical difficulties of building a topologically ideal drainage network (i.e. one whose fundamental unit is a tributary junction). Many standard GIS tools en-

counter problems when handling complex digital topography (represented using a DEM) that may contain natural or artificial depressions and whose grid cells are often much larger than real topographic features. Further complications arise when incorporating surface flow networks into integrated hydrologic models, because each link within the network must then be tagged with sufficient information to identify drainage pathways through the whole network, and the stream network must also be linked with sometimes different geometries and resolutions for surface-water and the groundwater-flow grids.

We addressed this challenge by creating eleven new GRASS GIS "extensions", also called "add-ons", that work alongside core GRASS GIS commands to transform user inputs (including a single DEM) into a set of GSFLOW inputs. This workflow creates an automatically generated network of streams and HRUs that is spatially linked to a MODFLOW grid. The domain-building procedure is automated through the **buildDomainGRASS.py** script, which takes inputs from the Settings text file, implements the domain-building workflow, and produces ASCII files used by GSFLOW–GRASS's Python input-builder scripts.

### 3.2.1 Surface-water network

In the first step of the fully automated domain-building workflow, GRASS GIS imports a user-provided DEM to define the drainage network and HRUs. After hydrologically correcting the DEM by filling pits and removing cells that have flow inputs from outside the map area (GSFLOW–GRASS requires the full topographical catchment to be included in the model domain), a Hortonian drainage network is constructed using the **r/v.stream.\*** toolkit (Jasiewicz and Metz, 2011) that relies on a single-flow-direction implementation of the **r.watershed** flow-routing algorithm (Metz et al., 2011). Sub-basins associated with each stream segment are designated as HRUs in order to follow both the natural discretization of the landscape and the architecture of PRMS (Markstrom et al., 2015). River headwaters are defined based on a threshold drainage area that may be weighted by the user to represent, for example, nonuniform precipitation or snowmelt inputs. Such weights permit a more realistic representation of drainage density and, as a result, increased model resolution in areas that contribute more water to the catchment.

The GRASS GIS drainage-network-creation algorithm, **r.watershed** (Metz et al., 2011), is both efficient and accurate. For computations that can take place entirely within memory, its speed exceeds that of both Terraflow and the D8 routing used by ArcGIS (Jenson and Domingue, 1988; Maidment and Morehouse, 2002; Arge et al., 2003; Magalhães et al., 2014). This speed results from its sorting algorithm and priority queue, and a standard desktop workstation today can process DEMs in memory with tens of thousands of cells on each side. The least-cost path approach taken by **r.watershed** does not require any pit-filling step, but we do so in order to create a more consistent DEM with downslope-routed flow for the remainder of the analysis. The flow-routing component of the more recent "Fastscape" algorithm by Braun and Willett (2013) could be faster than **r.watershed**, but these have not been benchmarked, and Fastscape is not yet integrated into the GRASS GIS toolchain, which is necessary for all of the subsequent steps. Kinner et al. (2005) demonstrated that **r.watershed** is more accurate than Terraflow (Arge et al., 2003), especially in low-relief areas and those in which tree canopy elevations are mistakenly interpreted as ground-surface elevations; this latter issue is pervasive across many digital elevation models, including the widely used Shuttle Radar Topography Mission (SRTM) DEM (Farr et al., 2007; Miliaresis and Delikaraoglou, 2009).

In spite of these advantages, **r.watershed** has not before been used to build flow networks for integrated hydrologic models. Other integrated hydrologic model domain-building tools use local drainage direction information (Bhatt et al., 2014; Maxwell et al., 2017; Gardner et al., 2018). While not an integrated hydrologic model due to its limited subsurface modeling capabilities, Srinivasan and Arnold (1994) integrated GRASS GIS version 4, including an earlier and much slower version of **r.watershed**,

into the Soil & Water Assessment Tool (SWAT). Beyond this, **r.watershed** is typically discussed in the drainage algorithm literature (e.g., Barnes et al., 2014; Magalhães et al., 2014; Schwanghart and Scherler, 2014; Sangireddy et al., 2016), directly applied to flow-routing and cost-path calculations (e.g., Wickert et al., 2013; Bird et al., 2016), or included as a component of an assessment tool (e.g., Bhowmik et al., 2015; Rossi and Reichenbach, 2016). By integrating **r.watershed** into GSFLOW–GRASS, via the **r/v.stream.\*** toolkit for Hortonion drainage network analysis (Jasiewicz and Metz, 2011), we are able to

harness the capabilities and efficiency of the hydrologic computational engine within GRASS GIS for integrated hydrological modeling.

Following drainage network construction, the next step in the automated workflow is to map the connections between each segment in the tributary network. To do this, we developed an extension called **v.stream.network**, which builds atop the upstream-to-downstream stream-segment and HRU indexing in the existing **r/v.stream.\*** toolkit (Jasiewicz and Metz, 2011).

This index is a unique positive integer identifier applied to each segment and its overlapping HRU, and is called a "category" in GRASS GIS. For each segment and overlapping HRU in the drainage network (Figure 1A,B), **v.stream.network** writes the category value of the immediately downstream stream segment to the "tostream" column in its associated attribute table row. Any stream segment exiting the map area is given a "tostream" value of 0. This links the stream segments and HRUs in the watershed as a directed graph (e.g., Czuba and Foufoula-Georgiou, 2014; Heckmann et al., 2014; Tejedor et al., 2015).

At this point, the user may optionally break out of the automated workflow and edit the vector geometries that define the streams and sub-basins. While we expect that many users will find the fully automated approach to be a major advantage over those that require manual intervention – these add a source of subjectivity and laborious processing time – Gardner et al. (2018) note that human-developed drainage structures may cause a discrepancy between topographically routed flow and actual flow paths. This manual adjustment is not standard, and requires the addition of a break point in **buildDomainGRASS.py**, as well

as for the user to manually adjust the category numbers (indices) and "tostream" network topology values in the attribute tables for the segments and HRUs if the changes are substantial enough to change the flow network.

After this, the study area is limited to a single drainage basin using the new **v.stream.inbasin** extension, completing the development of the drainage network geometry and topology. This step is included because the goal of many hydrologic studies is to understand a single watershed basin. If this is not the case, **buildDomainGRASS.py** may be edited to skip this

step and to analyze all complete drainage networks within the domain.

Each stream segment is then supplied with attribute values required for GSFLOW through the **v.gsflow.segments** extension. This numbers each segment for GSFLOW (Figure 1A) and populates the associated database table with hydraulic geometry, channel roughness (constant or spatially distributed), and channel and floodplain width (constant or spatially distributed). Additional less-commonly used options are also available, including additional input discharge for the upstream-most stream

segments (e.g., from human intervention), input diffuse runoff, and direct precipitation on the stream.

### 3.2.2 Groundwater-flow grid

Following the completion of the surface-water domain, the next step is to build the groundwater domain. MODFLOW-NWT uses a rectangular finite-difference grid structure (Harbaugh, 2005; Niswonger et al., 2011). The cell size for this grid is selected by the user in the Settings file. It is often desirable to discretize the MODFLOW groundwater domain on a grid that is coarser than the DEM used for surface flow routing in order to increase computational efficiency while still allowing GSFLOW–GRASS to generate a complex surface-water network; the proper grid cell size depends on the size of the HRUs and the strength of the surface-water–groundwater coupling. **v.gsflow.grid** builds the MODFLOW grid (Figure 1C) using the built-in **v.mkgrid** command while enforcing that it must contain only whole DEM grid cells, and that its edges must align with cell edges in the DEM.

Following grid creation, which often includes coarsening of the original DEM, **r.gsflow.hydrodem** then hydrologically corrects the elevations of the MODFLOW grid cells. Cells that contain stream segments are given a surface elevation corresponding to the lowest-elevation overlapping pixel on the fine-scale flow-routing DEM, while all other MODFLOW cells are assigned the mean elevation from the corresponding cells in the flow-routing DEM. This is crucial where a river valley is less than two grid cells wide: in this case, it is possible that the cell corresponding to the flow path will average over both the valley wall and the valley bottom, creating a bumpy valley floor that contains artificial dams. Thus, both the flow-routing DEM (high resolution) and the MODFLOW grid (typically, though not necessarily, lower resolution) are hydrologically corrected to enforce decreasing elevations down the drainage network.

### 3.2.3 Surface-water–groundwater coupling

The final step in developing the GSFLOW domain is to link the surface-water geospatial data structures (HRUs and segments) with the MODFLOW rectangular grid. **v.gsflow.reaches** and **v.gsflow.gravres** construct the reaches and gravity reservoirs (Section 3.1), which are the intersection of segments and HRUs, respectively, with each MODFLOW grid cell (Figure 1D). The database table for the reaches includes values for the thickness of the stream-bed sediment (defaults to 1 m) and its hydraulic conductivity (defaults to 5 m/d, characteristic of sand and gravel).

### 3.2.4 Accessing additional GSFLOW functionality

GSFLOW supports more input options than we have defined for our GRASS GIS **v.gsflow.\*** commands, though we have included many of the most common options. These are sufficient to set up and run a GSFLOW simulation. However, they may not encompass all of the variables that some users may consider to be important.

Therefore, GSFLOW–GRASS includes the **v.gsflow.mapdata** tool for users to add other attributes to database tables, with a focus on spatial distributions. These attributes can include spatially variable precipitation and temperature, parameter choices for model spin-up, and fully distributed maps of hydraulic conductivity, specific yield, streambed hydraulic properties, soil texture, vegetation type, and evapotranspiration parameters. The core capability of **v.gsflow.mapdata** is the use of averaging and nearest-neighbor methods to connect input data from raster grids, vector areas (polygons), or vector points, to the attribute

tables of the HRUs, segments, gravity reservoirs, reaches, and/or MODFLOW grid cells. As these are custom additions, calls to **v.gsflow.mapdata** must be added by end users to **buildDomainGRASS.py**. Once added, the end user can follow our template code in the input-file builder to add these to the GSFLOW input files. **v.gsflow.mapdata** therefore adds user-driven flexibility which input data can be supported by GSFLOW–GRASS, and a starting point for users who may want to expand on its capabilities.

### 3.2.5 Geospatial data export

In the final step, the generated attributes and geometries are exported. This information is stored in GRASS GIS as raster grids and vector geometries associated with SQL database tables. **buildDomainGRASS.py** exports a rasterized "basin mask" (1 in the basin, 0 outside) and the hydrologically corrected DEM at the MODFLOW grid resolution, as well as vectorized GIS data (shapefile format) for the HRUs, gravity reservoirs, MODFLOW grid, stream segments, stream reaches, pour point, full study basin area, and downstream boundary-condition cells. **v.gsflow.output** exports the database tables associated with the vectorized GIS data in comma-separated variables (CSV) files that can be read in by the input-file builder scripts (Section 3.3) for use in GSFLOW. These exported data are then ready to be parsed into GSFLOW inputs using the Python input-file builder scripts (Section 3.3) and/or to be used for data visualization (Section 3.5).

This separation between the GIS and ASCII-input-file components is intentional. The GRASS domain-builder component typically requires several minutes to run, and often only needs to be executed once for a watershed. The ASCII files, on the other hand, can form an effective basis for ensembles of runs. These can be used to calibrate parameters or explore hydrologic sensitivity to variable forcing scenarios.

### 3.3 GSFLOW Input File Builder

GSFLOW–GRASS includes a set of input-file builder scripts that are streamlined to incorporate the model domain discretization constructed by the GRASS GIS workflow and generate corresponding model inputs for the GSFLOW control file, PRMS-type input files, and MODFLOW-type input files. Most of the new features in GSFLOW that are not in stand-alone PRMS or MODFLOW follow the same Modular Modeling System input-data file format (Leavesley et al., 1996) as PRMS, which includes use of a "control file" as the main interface file, "modules" for different computational options, and the PRMS input file syntax. In contrast, MODFLOW uses a "name file" as its main interface file, implements "packages" for computational options, and follows its own file syntax. The following builder scripts handle these different formats and are automatically executed through the toolkit's Run file (Section 3.4). The builder scripts may also be customized for extensions beyond the default implementation.

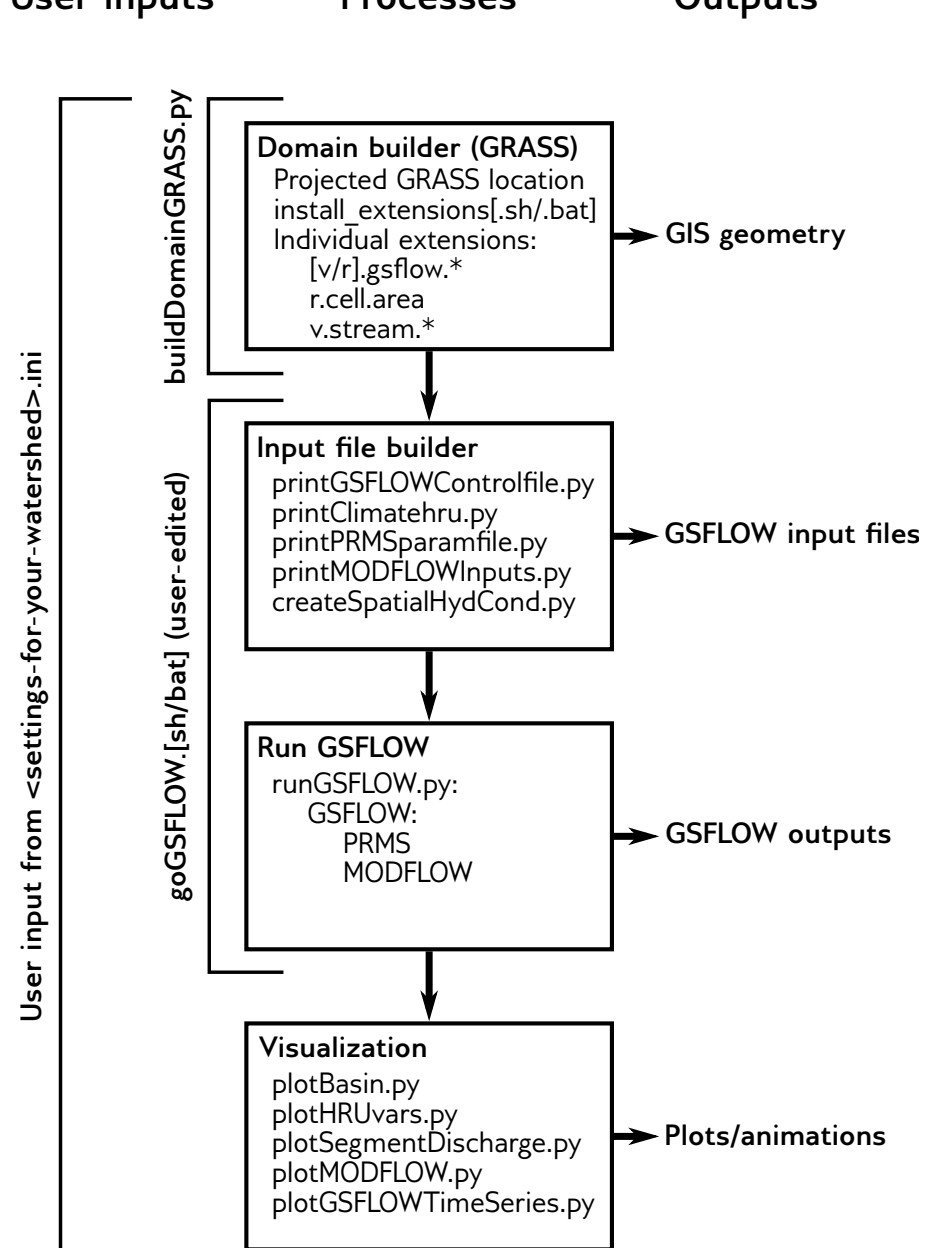

**Figure 3.** GSFLOW–GRASS workflow. The user: (1) creates a **\*.ini** file based on their study catchment; (2) creates a projected GRASS GIS location; (3) runs **buildDomainGRASS.py**; (4) edits and runs **goGSFLOW.py**. After this, they may use GSFLOW–GRASS's visualization tools to study the GIS and model outputs.

.

### 3.3.1 GSFLOW control file

The GSFLOW control file is the highest level input file and is created by the **printGSFLOWControlfile.py** script in the GSFLOW–GRASS toolkit. The toolkit is streamlined for configuring the integrated mode of GSFLOW (set through the "model_mode" parameter).

Inputs for the control file parameters are organized under six numbered sections in **printGSFLOWControlfile.py**. The script sets parameters related to climate forcing, time domain, and run mode based on what the user specifies in the Settings file; all other parameters are pre-set to default values. Further customization of control file parameters (stored in the list variable *con_par_name*) requires simply changing default values (in the corresponding list variable *con_par_values*) in the script; spatially variable entries can be generated with the aid of the **v.gsflow.mapdata** tool. The first two sections are required and

include details about the simulation execution and module choices. The third section establishes spin-up versus restart run modes based on Settings file entires. Sections 4 and 5 contain customizable lists of output variables to be printed, which can be used by visualization scripts in GSFLOW–GRASS (Section 3.5). The last optional section is for running the model in a debugging mode.

    Note that the default implementation of this toolkit uses the "climate_hru" module for precipitation and minimum and maxi-

mum daily temperature; this means that the model will employ pre-existing files containing data already specified by HRU. The PRMS component of GSFLOW does include other modules for distributing data from one or a handful of weather stations, but these typically require application-specific empirical parameters that are difficult to incorporate in a generic toolkit. Use of the "climate_hru" module provides flexibility for the user to implement their own spatial interpolation or extrapolation methods, which can then be transferred to the GSFLOW domain with the **v.gsflow.mapdata** tool. GSFLOW–GRASS's default imple-

mentation also uses the Priestley-Taylor formulation for potential evapotranspiration calculations (Markstrom et al., 2008). This module was chosen because of its reliance on only air temperature and solar radiation (calculated by the PRMS component of GSFLOW), and because of the relative ease of accounting for different vegetation properties through the parameter *pt_alpha* (in the PRMS parameter file, Section 3.3.2).

    After the six parameter input sections in **printGSFLOWControlfile.py**, the script builds the control file and then generates

an executable file (shell script for Linux or batch file for Windows) for running GSFLOW with the control file. After all other input files are created, this executable is called by the toolkit's automated Run file (Section 3.4). The executable can also be used to run GSFLOW outside of the GSFLOW–GRASS toolkit.

### 3.3.2 PRMS-type input files

Input files required for the PRMS component of GSFLOW are the parameter file ("param_file" in the control file), which

includes empirical surface and soil zone properties, and the data file ("data_file" in the control file), which includes climate observations for the spatial interpolation/extrapolation algorithms. If the "climate_hru" module is selected, as it is in the toolkit's default implementation (Section 3.3.1), then individual input files with HRU-distributed climate variables must also be specified. For a quick set-up of GSFLOW–GRASS, the script **printClimatehru.py** takes daily observations from a single file

and distributes them uniformly over all HRUs. The toolkit handles the minimum required climate variables – daily precipitation, maximum temperature, and minimum temperature, and it is set up to be readily extended to also include humidity, solar radiation, and/or wind speed. A spatially uniform approach may be acceptable where the size of a rainstorm is typically greater than the size of a catchment and climatic variables vary only weakly with slope and aspect. Larger and higher-relief catchments

require spatially distributed climate inputs for realistic outputs; these require custom inputs from the end-user, which can be ported from any discretization to the HRU domain with the aid of the **v.gsflow.mapdata** tool.

The parameter file is created by the script **printPRMSparamfile.py**. The script includes sections for domain dimensions and for parameters inputs, both of which are streamlined to take values parsed from the GRASS GIS domain builder outputs (as indicated in the comments in the script). Because of PRMS's conceptual soil moisture regimes, the parameter file requires a

substantial number of parameter inputs related to the soil and vegetation that cannot easily be specified without calibration. As a default to help the user get GSFLOW up and running, most parameter values in **printPRMSparamfile.py** are preset, mostly using calibrated values from the Sagehen watershed example that was distributed with the GSFLOW model version 1.2.1. We have indicated with the comment "# *** CHANGE FOR SPECIFIC SITE" those parameters that could also be altered based on known characteristics of one's watershed site. This includes various soil and land-cover inputs, such as *soil_type* (sand, loam,

or clay), *cov_type* (bare soil, grasses, shrubs, or trees), *transp_end* (end month of transpiration, for phenology), and *pt_alpha* (Priestley-Taylor parameter $\alpha$, which can be based on literature values). In addition to these highlighted parameters, users can review all parameters to determine whether others could be particularly important for their specific application. These may include some of the parameters mentioned in Section 2.1.2 that determine exchanges between different soil-zone reservoirs. Spatially variable information can be transferred to the HRU domain using the **v.gsflow.mapdata** tool. Rigorous calibration of

PRMS parameters can eventually be carried out with inverse codes such as PEST (Doherty, 1994) or UCODE (Poeter and Hill, 1998, 1999).

### 3.3.3    MODFLOW-type input files

GSFLOW requires input files for each MODFLOW package utilized, which can include any of the packages listed in Table 1 of the GSFLOW manual (Markstrom et al., 2008, Appendix 1, p. 176-226 provides details). Our toolkit by default creates a

relatively general MODFLOW set-up, which includes required input files and omits most optional ones, such as the Well package. Our Python library **MODFLOWLib.py** consists of functions for creating: four Basic package input files (name file, basic package file, discretization file, and the optional output control file for customizing output files), two different groundwater flow package options (the Layer-Property Flow (LPF) from MODFLOW-2005 and the Upstream Weighting Package (UPW) from MODFLOW-NWT), the numerical solver package (Preconditioned Conjugate Gradient (PCG) for LPF and Newtonian

(NWT) input file for UPW), the Streamflow-Routing package (SFR), and the Unsaturated-Zone Flow package (UZF).

The script **printMODFLOWInputs.py** calls the functions from **MODFLOWLib.py** to create a set of internally consistent input files that incorporate the domains constructed by the GRASS-GIS workflow (Section 3.2) and conform to the simulation directory structure established through the **ReadSettings.py** utility. By default, **printMODFLOWInputs.py** calls the MODFLOW-NWT UPW/NWT flow package instead of the MODFLOW-2005, because of the superior numerical performance

of the former in tests with steep elevation gradients (e.g., Section 4.1). If desired, users can easily switch to the LPF/PCG formulation from MODFLOW-2005 by setting *sw_2005_NWT* = 1 in **printMODFLOWInputs.py**.

Input files created outside of our toolkit for a stand-alone MODFLOW model implementation of identical discretization will for the most part be usable with the integrated GSFLOW model. However, as indicated in Table 1 of the GSFLOW manual,
some MODFLOW packages were modified for their use in GSFLOW. Advantages of implementing our toolkit over using pre-created MODFLOW input files are that it already incorporates these GSFLOW modifications, it automatically uses the GRASS-GIS builder results for the domain, and it guarantees a directory structure that is consistent with the rest of the input files and the visualization scripts.

The GSFLOW–GRASS toolkit also offers an optional script **createSpatialHydCond.py** for generating spatially distributed
hydraulic conductivity fields for the upper layer based on elevation and/or distance from the stream network, with the assumption that lower elevations and/or riparian corridors have higher hydraulic conductivity properties. Because application-specific entries cannot easily be generalized for input through the Settings file, users should directly customize elevation and stream distance thresholds, as well as corresponding hydraulic conductivity values, at the top of the **createSpatialHydCond.py** script. This script will automatically import domain information from the Settings file and export results to the file location specified
by the Settings file. **createSpatialHydCond.py** serves as a ready-to-go tool for creating physically plausible hydraulic conductivity patterns, and it provides an example for how users can create their own scripts to customize spatially distributed inputs. A similar type of script could create spatially distributed infiltration fields for the preliminary MODFLOW steady-state simulation in spin-up runs (e.g., *finf* entry in the Settings file). These tools can provide preliminary inputs to jump-start GSFLOW model implementations. However, realistic construction of hydrogeologic frameworks relies on data from sources such as well
logs, geologic maps, geophysical measurements, and pumping tests (Reilly, 2001; Reilly and Harbaugh, 2004). For these, we recommend that users import the appropriate data sources into GRASS GIS and use the **v.gsflow.mapdata** extension to map these parameters onto the appropriate GSFLOW objects (e.g., HRUs, MODFLOW cells). Properties for stream segments and reaches – such as streambed hydraulic conductivity, and unsaturated hydraulic properties below the streambed – are set to default values that can be changed through the GRASS GIS extensions. By default, the streamflow calculation is set to use
Manning's equation by assuming a wide rectangular channel (*ICALC*= 1). Spatially variable stream widths and/or Manning's $n$ values may be set through the Settings file, based on either gridded or point-based (e.g., survey) data, and **v.gsflow.segments** also supports the delineation of both channel and floodplain geometries and roughness parameters.

## 3.4   GSFLOW run file

For the user's convenience, the GSFLOW–GRASS toolkit includes an executable Run file, which is a shell script for Linux,
**goGSFLOW.sh**, and a batch file for Windows, **goGSFLOW.bat**. The Run file collects input from a specified Settings file and then runs all of the above input-file builder scripts; the script **runGSFLOW.py**, which launches the GSFLOW simulation; and the runtime visualization script **plotGSFLOWTimeSeries_Runtime.py**, further described below. If the runtime visualization is not desired, the user can comment out the corresponding execution line in the Run file. As long as the user does not wish to use more features than are exposed in the Settings file, no direct interface with the code is required to run GSFLOW–GRASS.

This permits a "quick-start" implementations of GSFLOW, which can substantially lower the barrier to entry for using this model.

The Run file may be implemented only after the model domain is generated through **buildDomainGRASS.py**. The GSFLOW–GRASS toolkit separates the GRASS domain-builder module from the Run file because users will typically only need to construct their domain once, but will perform multiple runs of the model with variable parameter inputs, for example, for model calibration or to simulate different time periods.

After preliminary quick-start simulation tests, users can further customize their runs by taking advantage of the modular structure of the toolkit, which has a separate script for each input file. For example, to target specific aspects of the model, such as the surface runoff properties, corresponding parameters may be adjusted in the PRMS parameter file by editing and re-running **printPRMSparamfile.py**. Select input-file builder scripts can be run either within Python, or by editing the executable Run file.

## 3.5 Visualization tools

Our toolkit includes post-processing Python scripts that employ the Matplotlib plotting library (Hunter, 2007) for visualizing the domain discretization, key MODFLOW inputs, and model output results. The model discretization for the PRMS component of GSFLOW is exported from GRASS GIS as a set of standard vector GIS files (shapefiles). Our Python plotting scripts use these shapefiles to create figures of the surface HRU and stream segment discretization (**plotBasin.py**), and to generate movies of HRU-distributed and stream segment-distributed variables (**plotHRUvars.py** and **plotSegmentDischarge.py**). These output variables (e.g., evapotranpsiration and streamflow) are set through *aniOutVar_names* in the GSFLOW control file (see Section 3.3.1). The exported shapefiles may also be used to visualize model results with standard GIS packages (e.g., QGIS: QGIS Development Team, 2013) outside of GSFLOW–GRASS.

For the MODFLOW component of GSFLOW, the toolkit's script **plotMODFLOW.py** plots spatially distributed layer elevations, hydraulic conductivity, and a map of active computational grid cells. The script also plots spatially distributed MODFLOW simulations results over time, including for hydraulic head, change in head, water table depth, and recharge from the unsaturated zone. For storage efficiency, the toolkit creates and reads in head and unsaturated zone output files in binary format.

For basin-total GSFLOW results, the Python script **plotGSFLOWTimeSeries.py** generates time series lines for user-selected variables from the main GSFLOW CSV output file. Names of all variables, along with their descriptions and units, are listed in **GSFLOWcsvTable.py**, which is imported into **plotGSFLOWTimeSeries.py** to ensure consistency in figure labels and axes. Our toolbox also includes the runtime visualization script **plotGSFLOWTimeSeries_Runtime.py** that is by default called by the Run file (but can be commented out if desired) and displays a continuously updated time series plot of basin-total precipitation and discharge. Tracking simulation progress with runtime plots can be very useful for complex integrated models, which can have lengthy simulation times.

The visualization scripts can be run using a command-line parser and/or by editing plot options that appear near the top of each script. More advanced users may modify the bodies of the scripts to change to features such as axis intervals or color schemes. For users who want to adjust the scripts, we suggest running them in the iPython interactive programming console

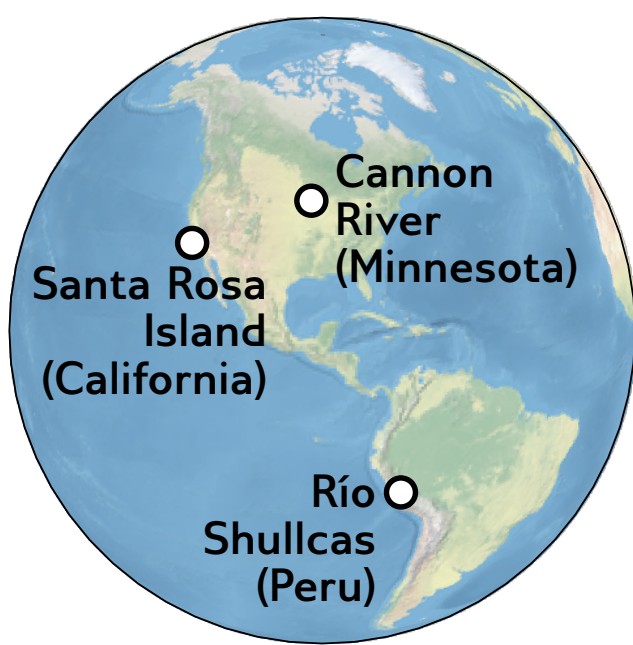

**Figure 4.** Our test sites include the high Andes, a mountainous island, and a formerly glaciated Mississippi tributary.

(Pérez and Granger, 2007), which is also incorporated into the Spyder integrated development environment (IDE). Although this visualization approach requires some familiarity with Python and/or command-line argument parsing, it accommodates a wide range of plotting preferences. All plots and videos may be displayed as on-screen figures (in raster or vector formats, using the interactive Matplotlib window), and may be saved as images (interactively) or videos (**\*.mp4** format) as defined by
5   inputs to the plotting script.

Other existing no-fee USGS GUI programs for MODFLOW also provide visualization capabilities, and using these with the input and output files produced with GSFLOW–GRASS is possible. In particular, GW Chart (Winston, 2000) can be directly implemented for plotting basin-level time series results. Additionally, Model Viewer (Hsieh and Winston, 2002) and ModelMuse (Winston, 2009) are able to read in and plot spatially variable head results from binary files with the extension
10  ".bhd," but this does require manual post-processing steps. For Model Viewer, the user needs to copy all MODFLOW input and output files to a new folder inside the Model Viewer project directory and select the namefile when prompted. For Model Muse, the user must first delete the line that starts with "IWRT" from the name file in order to load the project into the program. Once the project settings are loaded into ModelMuse, the user can use the "import model results" tool to select the binary head file.

## 4 Examples

Three example implementations demonstrate (1) the variety of hydrological processes and environments that can be explored using GSFLOW–GRASS, and (2) how the toolkit's GIS domain builder can handle diverse topographic settings, including those prone to problems with standard GIS stream network tools. Towards the first point, the specific examples chosen represent a
range of practical applications for water and land management. Towards the second point, each simulation presents a unique set of technical challenges in developing a topographically based model domain that can properly route rainfall through a network of stream segments and sub-basins as well as a connected groundwater-flow grid. It is important to note, however, that no calibration effort was made to match field observations for these test cases. The simulation results thus serve as purely schematic examples based on certain settings and do not aim to capture actual conditions at the corresponding sites.

The examples are based on the water-stressed Shullcas River Watershed, Junín Region, Peru, which is experiencing rapid glacier retreat; Water Canyon on Santa Rosa Island off the coast of California, USA, which has undergone land-cover change impacts; and the formerly glaciated Cannon River watershed, in which water flows from intensely farmed uplands into an incised bedrock valley in Minnesota, USA (Figure 4; Tables 2 and 3). All regions contain complex hydrology with interactions between surface water and groundwater and are exemplars of practical management concerns. Together they span a range of
environments: high to low elevations, steep to low-gradient catchments, coastal to inland settings, tectonically active to cratonal, and with partially to fully integrated drainage. Their catchment areas range from 12.5 km$^2$ to 3723.0 km$^2$, covering the range of scales that GSFLOW was developed to simulate. They are affected by modern climate and land-use change impacts on glaciers and agricultural (water and soil) resources (Shullcas) (Gómez et al., 2014; Arroyo Aliaga et al., 2015; Travezan Adauto, 2015; Somers et al., 2018), grazing-induced erosion (Santa Rosa) (Schumann et al., 2016), and agricultural runoff and fertilizers
(Cannon River) (Kreiling and Houser, 2016). Our choice of an example in the Peruvian Andes demonstrates how our entirely open-source modeling system may be applied to problems in the developing world, where financial limitations faced by local environmental researchers and practitioners make it difficult to use commercial software solutions.

Figures 5–7 display sample inputs and outputs of the model simulations using the default GSFLOW–GRASS toolkit for the three test cases. These applications show that even before any parameter adjustments, the GSFLOW–GRASS toolkit can
readily generate GSFLOW model domains and parameter inputs that produce numerically convergent simulations in a variety of topographies and hydroclimatic conditions.

Preliminary simulations with the default GSFLOW–GRASS provide a valuable springboard for the next step of performing the calibration needed to generate realistic model outputs for specific sites. The GSFLOW–GRASS toolbox can be customized to quickly generate additional model runs with varying input values to expedite the parameter calibration. It can also facilitate
the implementation of sensitivity or other Monte Carlo-type analyses that are critical for identifying issues such as equifinality and over-parameterization and for determining uncertainty estimates (Beven, 2006; Gallagher and Doherty, 2007; Razavi and Gupta, 2015; Song et al., 2015).

**Table 2.** Catchment and hydrological characteristics of the GSFLOW–GRASS example sites.

| Site | Drainage area [km$^2$] | Flow-routing cellsize [m$^2$] | MODFLOW cellsize [km$^2$] | Min. HRU area [km$^2$] | Elevation range [m] | Mean annual rainfall [mm] | Daily rainfall CV |
|---|---|---|---|---|---|---|---|
| **Shullcas River**, Junín Region, Peru | 161.4 | 930.93 | 0.25 | 1 | 3526–5527 | 1076 | 1.4 |
| **Water Canyon**, Santa Rosa Island, California, USA | 12.5 | 8100 | 0.0324 | 0.4 | 23–378 | 265 | 5.4 |
| **Cannon River**, Minnesota, USA | 3723 | 225 | 1 | 10 | 203–413 | 756 | 3.2 |

Precipitation statistics from 2013-08-26 to 2016-09-29 (Shullcas); 1990-04-23 to 2017-09-27 (Water Canyon); 1938-05-12 to 1943-11-05 (Cannon). "Flow-routing cellsize" is the resolution used to construct the stream network and irregular HRU cells, which are ultimately coarser-sized ("Min. HRU area"). CV = coefficient of variation.

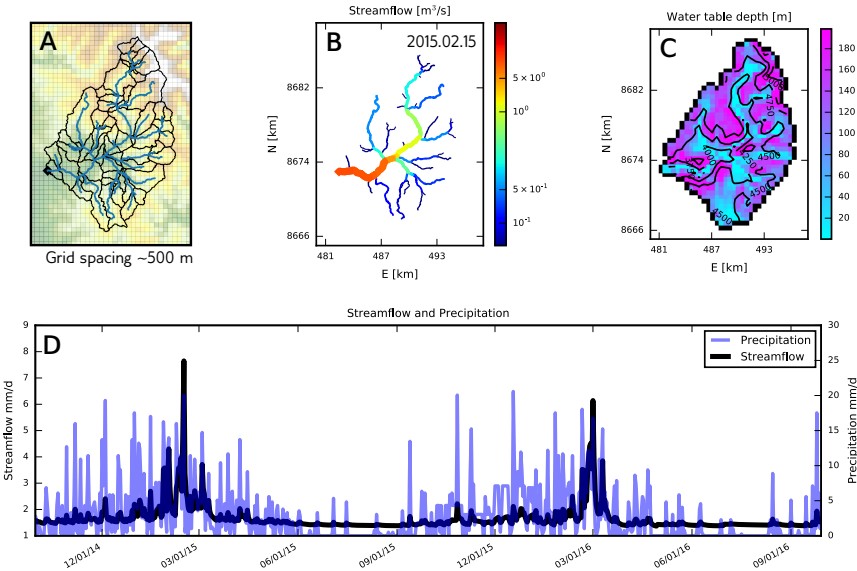

**Figure 5.** Model based on Río Shullcas, Peru. **(A)** Map with MODFLOW grid, HRU outlines, stream segments (blue), and digital elevation model. **(B)** Streamflow accumulation through the mountainous drainage network. **(C)** The modeled water table distribution with elevation contours (m.a.s.l.). **(D)** Seasonally variable precipitation and streamflow.

## 4.1 Shullcas River watershed, Peru

The first test case is based on the Shullcas River Watershed, located in the central Peruvian Andes. Precipitation is highly seasonal, and water shortages are common during the dry season from May to September. The Huaytapallana Glacier, which

supplies meltwater to the Shullcas River, is rapidly retreating (López-Moreno et al., 2014), causing concern over future water resources (Somers et al., 2018). However, in glacierized watersheds of the Peruvian Andes, a large proportion of the dry season stream discharge can be composed of groundwater (Baraer et al., 2015), driving the need to better understand groundwater-surface water interactions in the catchment.

The simple hydrologic model based on the Shullcas watershed covers an area of 161.4 km$^2$ and ranges in elevation from 3626 to 5527 m above sea level (a.s.l.) (Table 2). Using the GRASS domain-builder, the watershed was divided into 59 sub-basin HRUs based on an ASTER nominal 30 m resolution DEM (Tachikawa et al., 2011). The subsurface was represented by a single 200 m thick MODFLOW layer, with a horizontal discretization of 46 rows, each with a length of 485 m, by 33 columns, each with a width of 492 m. Meteorological data were obtained from the Peruvian Meteorological Office (SENMAHI) online
database.

The steep topography makes the Shullcas River Watershed an apt testbed for examining the ability of GSFLOW–GRASS to represent surface-water–groundwater links in challenging terrain. The major obstacle with Shullcas' steep topography and narrow canyons is the need for high resolution to represent surface flow, which leads to an impractically high number of computational units and expense if using a standard gridded model domain. An irregular surface grid can provide a much more
efficient discretization, but this then entails painstaking indexing to link it to the subsurface grid, which must be a regular rectangular grid for the MODFLOW component of GSFLOW. Also, as the most computationally expensive part of GSFLOW, practical MODFLOW implementation typically needs a coarser resolution grid than that used for resolving the stream network, but simple coarsening of DEMs with steep gradients (e.g., by using mean elevations from the higher-resolution DEM) can result in hydrologically incorrect groundwater flow directions. GSFLOW–GRASS addresses these problems by computing flow paths
using high-resolution topography ($\sim 2 \times 10^5$ grid cells of 900-m$^2$ size) and converting these into a far smaller number of larger computational surface cells (79 HRUs that are $\geq$1 km$^2$ in area) that convey the same fundamental surface-flow information (see Table 2); this efficiency is possible because the surface is discretized along topographically defined surface-water hydrologic units – stream segments and sub-basins (Section 3.2.1). To create a coarsened subsurface rectangular grid domain for the MODFLOW component, GSFLOW–GRASS' hydrological correction to enforce integrated subsurface drainage (Section 3.2.2)
proved essential for preventing unrealistic results. Early model tests for Shullcas showed that simple grid coarsening using the mean value of the elevations from the higher-resolution grid could, for example, average elevations between flat valley floors and steep canyon walls. This caused cells containing the stream to be higher in elevations than the surrounding surface on the groundwater flow grid, leading to lateral flow out of these "dams" that formed as a numerical artifact of averaging. As the final step of the domain-building solution, GSFLOW–GRASS extensions seamlessly link the hydrologically corrected coarse-scale
MODFLOW domain with the irregular surface HRUs (Section 3.2.3).

The Shullcas-based simulation does not represent glacier melt, but spatiotemporal results in Figure 5 show that GSFLOW can be useful for evaluating the potential for groundwater to buffer surface water resources in mountainous watersheds with high seasonal precipitation variability. In simulations, discharge varies seasonally in response to precipitation, with peak discharge occurring late in the wet season, after significant antecedent moisture has built up within the catchment, and essentially constant
baseflow supporting low but reliable discharge throughout the dry season (Figure 5D). The GSFLOW–GRASS post-processing

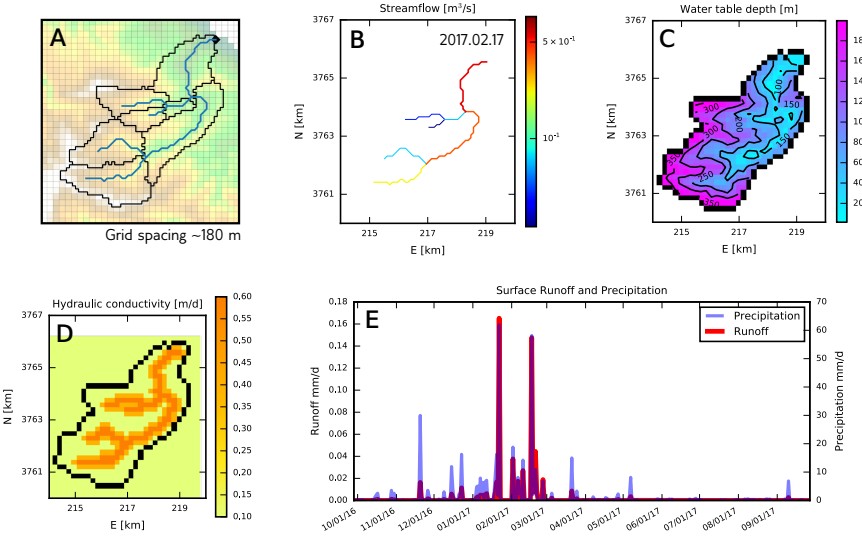

**Figure 6.** Model based on Water Canyon, Santa Rosa Island, California, USA. **(A)** Map with MODFLOW grid, HRU outlines, stream segments (blue), and digital elevation model. **(B)** Streamflow accumulation through the drainage network. **(C)** The modeled water table distribution with elevation contours (m.a.s.l.). **(D)** Spatially variable hydraulic conductivity structure, with hydraulic conductivity increasing near the channel to represent alluvium and colluvium. **(E)** Simulated surface runoff contributions to catchment-wide discharge compared with precipitation.

visualization tools were useful for depicting the accumulation of streamflow throughout the drainage network (Figure 5B) and water table depths that were shallowest in low and flat areas (Figure 5C). The model simulates that most rainfall infiltrates to recharge the aquifer, with relatively little overland flow. This result likely underestimates actual surface runoff, considering the significant erosive overland flow events have occurred in the recent past (Wagner et al., 2004). Nevertheless, preliminary results
5   depict groundwater converging at the stream network that can give information about whether baseflow can sustain discharge at the catchment outlet during dry periods.

### 4.2   Santa Rosa Island, California, USA

Santa Rosa Island is one of the Channel Islands of California, USA, and is part of the Channel Islands National Park. The island has an area of approximately 214 km$^2$ and is characterized by mountainous topography, with its highest point at 484 m
10   a.s.l. (Clark et al., 1990). Hydrologic modeling of Santa Rosa Island has previously been performed by Jazwa et al. (2016), who applied the PIHM hydrologic model (Qu and Duffy, 2007) to the island in order to understand the relationship between prehistoric human settlement patterns and surface water availability. They reported streamflow characteristics modeled for the 19 major drainages around the island during hypothetical climate regimes that are wet, dry, and of average wetness when compared to modern conditions. Unlike PIHM, GSFLOW–GRASS employs a regular three-dimensional groundwater grid that

**Table 3.** Model implementations based on three sites serve to test GSFLOW–GRASS capabilities and demonstrate applications.

| Site | GSFLOW–GRASS capabilities | Applications |
|---|---|---|
| **Shullcas River**, Junín Region, Peru | Efficient discretization of steep topography; Hydrologically corrected coarsening | Water resources in mountain catchments; Groundwater-surface water interactions under seasonally variable precipitation |
| **Water Canyon**, Santa Rosa Island, California, USA | Irregular / coastline boundaries; Hydrologically corrected coarsening; Spatially distributed hydraulic conductivity | Management of eroding hill slopes; Semi-arid climate with losing streams |
| **Cannon River**, Minnesota, USA | Two-layer hydraulic conductivity; Least-cost flow algorithm for poorly integrated drainage | Mixed agricultural—recreational watersheds; Strong temperature seasonality |

Precipitation statistics from 2013-08-26 to 2016-09-29 (Shullcas); 1990-04-23 to 2017-09-27 (Water Canyon); 1938-05-12 to 1943-11-05 (Cannon). CV = coefficient of variation.

does not align with the irregular surface domain; this makes the integrated domain building more complicated but allows for a flexible representation of the surface-water and aquifer systems.

Here we apply the GSFLOW–GRASS toolbox to model Water Canyon (Tables 2 and 3), one of the island's many drainages. We generated the surface flow routing system with topography derived from a 3 arcsecond SRTM DEM (Farr et al., 2007) projected to a UTM coordinate system at 90 m resolution, and we down-sampled the DEM to 180 m resolution for the MODFLOW grid. We drove simulations shown in Figure 6 using weather data from the Western Regional Climate Center (wrcc.dri.edu).

Water Canyon is unique among the three example sites in that its outflow drains to the ocean. It therefore requires GSFLOW–GRASS to accommodate irregular boundaries (coastlines) by properly assigning boundary conditions and routing flow through them. Users identify ocean pixels by assigning NULL values to them; this causes flow routing from **r.watershed** to stop at the shoreline. To allow flow out of pour-point at the mouth of the river, the immediately downgradient MODFLOW cell can be set as a constant-head boundary, but this cell must be chosen carefully. The finite-difference scheme in MODFLOW dictates that the constant head boundary condition must be supplied along one of the four cardinal directions of the pour-point. Therefore, if the river flows diagonally to the sea, its constant-head boundary must be moved to the closest non-diagonal cell. **v.gsflow.grid** finds the proper constant-head boundary cell to set for the coastal case, as well as for any inland drainage case in which the pour point also requires a downgradient constant-head boundary.

Losing streams such as those in the steep and semi-arid Water Canyon catchment often run dry Jazwa et al. (2016). If this causes MODFLOW cells to lose all of their water, GSFLOW will fail to numerically converge. Thus, the Water Canyon example also serves to demonstrate GSFLOW–GRASS' ability to prevent this problem by (1) incorporating MODFLOW-NWT, which uses a Newton–Raphson solver for increased stability (Niswonger et al., 2011); (2) allowing the user to specify an adequately deep MODFLOW discretization in the Settings file (Section 3.1) to supply sufficient water through the dry season; and (3) hydrologically correcting the elevations of coarsened MODFLOW cells to enforce integrated drainage through the stream network. Focusing on the third approach that is specific to the GSFLOW–GRASS toolbox (Section 3.2.2), the narrow and steep

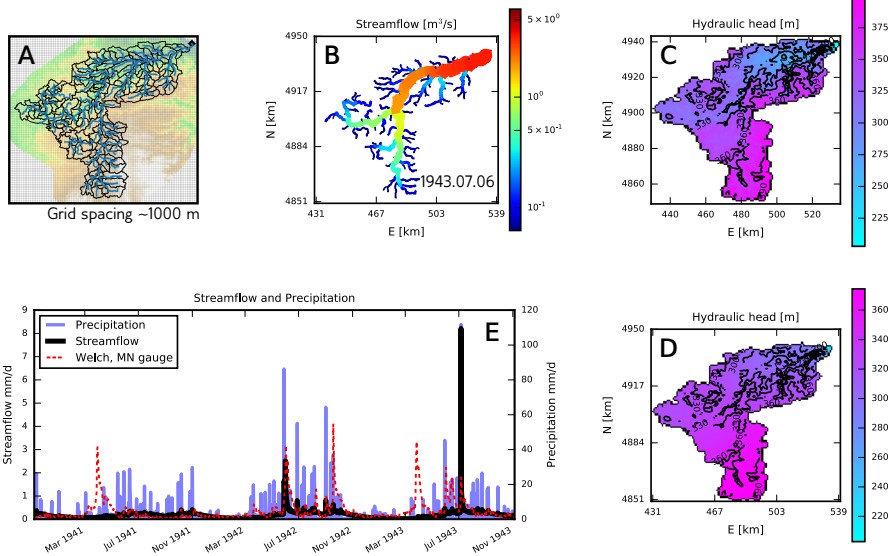

**Figure 7.** Model based on Cannon River, Minnesota, USA. **(A)** Map with MODFLOW grid, HRU outlines, stream segments (blue), and digital elevation model. **(B)** Simulated discharge after an 11 cm rainfall event. **(C, D)** Relatively low-gradient hydraulic head distributions in two MODFLOW layers representing an upper glacial till unit (low hydraulic conductivity) and lower fractured carbonate bedrock (higher hydraulic conductivity), with elevation contours (m.a.s.l.). **(E)** Three-year hydrograph showing uncalibrated discharge simulations matching observations reasonably well during non-peak flood times but poorly during many of the actual peaks.

Water Canyon requires the same hydrologic corrections that were applied in the Shullcas case, above, to maintain downslope-integrated drainage. Under losing stream conditions, artificially increased channel elevation would steepen the hydraulic head gradient away from the channel and cause it to over-simulate water flow to the surrounding landscape. Therefore, hydrologic correction of the coarse MODFLOW grid is necessary to simulate appropriate head gradients and maintain water in cells,
5   which is further required for any attempt to match stream-gauge records.

The Santa Rosa example demonstrates an application in which GSFLOW–GRASS can be used to investigate and manage erosion associated with hydrological conditions. Erosion of upland areas moves sediment downslope to the areas flanking the stream channel, which contains coarser-grained alluvial sediments. We represented this heterogeneity using a spatially distributed hydraulic conductivity field (Figure 6D) generated with the example model input script included in GSFLOW–
10   GRASS (Section 3.3.3). Figure 6E demonstrates how the post-processing tools can be used to evaluate surface runoff, a driver of erosion on the island (Schumann et al., 2016). Simulations show precipitation events triggering surface runoff (Figure 6E), which could denude the hillslopes and transport eroded sediment through the drainage network (Figure 6B).

## 4.3 Cannon River, Minnesota, USA

The Cannon River is a tributary to the upper Mississippi River in Minnesota, USA. Its headwaters cross low-relief uplands that are capped by low-hydraulic-conductivity glacial deposits (Patterson and Hobbs, 1995) and are intensively farmed (Kreiling and Houser, 2016). Its lower reaches pass through a valley cut into fractured carbonate bedrock that is popular for recreation. This combination of agricultural and recreational uses and its transient geomorphology (low-gradient headwaters above a high-gradient river) are common in the formerly glaciated Upper Midwest (Blumentritt et al., 2009; Carson et al., 2018). This leads to a suite of management concerns related to agricultural nutrients and fine sediments (Czuba and Foufoula-Georgiou, 2014), and their interactions with both the surface water and the bedrock aquifer systems that underlie them (Tipping, 2006; Steenberg et al., 2013), thus motivating the need for integrated hydrologic modeling tools.

We implemented GSFLOW–GRASS for the Cannon River watershed using the Minnesota statewide 1 m LiDAR data set (http://www.mngeo.state.mn.us/chouse/elevation/lidar.html), which we resampled to 15 m resolution. We discretized the sub-surface of the Cannon River watershed into 1 km MODFLOW grid cells. Meteorological data from nearby Zumbrota, Minnesota was obtained from the Midwestern Regional Climate Center (Adresen et al., 2014). The flexible GSFLOW–GRASS input builder allows for easy implementation of two MODFLOW layers to represent an upper glacial till unit (low hydraulic conductivity) and the underlying fractured carbonate bedrock (higher hydraulic conductivity).

Covering 3723 km$^2$, the Cannon River watershed is by far the largest of the three model implementations (Table 2) and benefits from the efficiency of the topographically based surface grid and the hydrologic robustness of the grid coarsening method in GSFLOW–GRASS. 17,455,046 flow-routing grid cells, each of which is 225 m$^2$ in area, were converted to only 610 irregular HRUs of $\geq$10 km$^2$ in area. For the groundwater domain, the elevation data were coarsened and hydrologically corrected to a 1 km regular MODFLOW grid.

Over the 3723 km$^2$ drainage area, there is only 210 m of total relief, and Pleistocene glaciation produced a significant amount of non-integrated drainage that presents very different computational challenges than those in the steep watersheds discussed previously. Much of the Cannon River watershed's post-glacial topography is characterized by small localized hills and enclosed basins that have not yet been organized (or integrated) by fluvial erosion into a linked valley network, in which water flows directly to a stream without encountering an enclosed depression (such as a lake, wetland, or dry basin). In such settings that lack integrated drainage, downslope flow-routing and "pit-filling" algorithms that are typically used to build hydrologic model domains (e.g., Bhatt et al., 2014; Maxwell et al., 2017; Gardner et al., 2018) can fail or produce spurious results by inappropriately modifying the real topography. As described in Section 3.2.1, GSFLOW–GRASS determines surface-water flow using the GRASS GIS's efficient and accurate **r.watershed** extension, which implements a least-cost path algorithm designed to produce drainage networks that route flow in the long-range path of steepest descent regardless of the degree of local drainage integration. By using **r.watershed** alongside a set of new GRASS-GIS extensions that integrate it into the GSFLOW framework, GSFLOW–GRASS is able to automatically create a topologically correct and linked drainage network in settings that lack integrated drainage for hydrologic model simulations. While this successfully builds the computational domain for the

watershed, the user must still put significant effort into adjusting the HRU parameters in the uplands to appropriately partition rainfall, storage, infiltration, and runoff.

The northern, mid-continental temperate setting makes the Cannon River watershed the example application with the most evenly distributed precipitation across seasons and strongest seasonal temperature differences. In the model, this low-relief catchment generally exhibits low hydraulic head gradients in both MODFLOW layers, except around the river gorge near the outlet, where head levels drop (Figure 7C-D). Comparisons between the simulated streamflow at the watershed outlet and corresponding observations at Welch, MN over the three-year model run reveal that without any parameter calibrations, the model produces realistic discharge during non-peak flood times and during one of the observed peaks during July 1942 (Figure 7E). The severely over-simulated discharge in July 1943 may be evidence for a local convective summer storm system passing over the Zumbrota weather station, which is located outside of the watershed boundary. Recurring failure of the model to capture April discharge indicates that snowmelt-related parameters require adjustment. Once the model is calibrated, which can be facilitated by applying parameter-estimation approaches (e.g., Doherty, 1994; Poeter and Hill, 1998, 1999) together with the automated GSFLOW–GRASS toolkit, results can be used to evaluate infiltration from overlying agricultural plots to shallow and low-gradient water tables, as well as subsequent flushing of impacted shallow groundwater into the river channels during major storms, as shown in Figure 7B.

## 5    Conclusions

To address the need for a fully automated and freely accessible software that handles the complete workflow for implementing complex hydrologic models, we have created GSFLOW–GRASS, a bundled toolkit for the coupled surface-water and groundwater model GSFLOW, using open-source Python scripts and GRASS GIS commands. GSFLOW–GRASS allows users equipped with a DEM, precipitation and temperature data, and basic knowledge about land-surface and subsurface properties to efficiently construct watershed-scale hydrologic simulations. In order to create a robust tool that can be widely implemented over diverse hydro(geo)logic settings, we built a set of GRASS GIS extensions that automatically discretizes a topological surface-water flow network that is linked with the underlying gridded groundwater domain. Our fully automated and generalized toolbox advances the accessibility of complex hydrologic software and will thus broaden the reach of integrated hydrologic models and their usage in both scientific research and practical resource management.

We have demonstrated GSFLOW–GRASS using three diverse examples based on topographies and climates from the water-stressed Andes, Santa Rosa Island off the coast of California, USA, and the intensively farmed Upper Midwest region of the United States. The results show that the new and automated GRASS GIS extensions can automatically and consistently build topologically complete linked surface and subsurface flow domains in settings that are typically challenging for standard GIS tools, including steep topographies, irregular coastal boundaries, and low-relief terrains that lack integrated drainage. Although uncalibrated, these examples further demonstrate that GSFLOW–GRASS is a flexible tool for investigating the role of groundwater-surface water interactions in modulating dry-season discharge, controlling runoff in erosion-prone landscapes, and imposing possible water-quality threats in agricultural and recreational watersheds.

We designed GSFLOW–GRASS to strike a balance between direct "out-of-the-box" functionality and full flexibility for customizing model runs. A default implementation can be launched with no programming required by the user to readily produce preliminary uncalibrated simulations that can serve as a springboard for further model-parameter adjustment through the fully commented toolkit scripts. A key feature of GSFLOW–GRASS is its use of all open-source software, enabling users anywhere to apply GSFLOW. We believe that the open-source platform will facilitate future toolbox enhancements through efforts by not only the original GSFLOW–GRASS developer team, but also new model users. We envision a number of new capabilities to tackle the grand challenge of handling spatial heterogeneity in integrated hydrologic models. Higher resolution land-surface variability could be achieved by further subdividing sub-basins according to vegetation, soil type, or other geographic features to produce HRUs. Obtaining spatially variable information can be facilitated by linking GSFLOW–GRASS to existing regional to international databases for meteorology, soil and geologic properties, and land cover. Further calibration of spatially distributed parameters can be carried out by directly setting up GSFLOW–GRASS with a flexible inverse modeling code (e.g., Doherty, 1994; Poeter and Hill, 1998, 1999). It is our hope that with its generalized form and open-access, GSFLOW–GRASS can become a community tool that continues to grow to better solve hydrologic and water resources problems of both scientific and general management concerns.

*Code availability.* The version of GSFLOW–GRASS used for this paper is available at https://github.com/UMN-Hydro/GSFLOW--GRASS/releases. Updated versions of our code are downloadable directly from the UMN-Hydro repository on GitHub, at https://github.com/UMN-Hydro/GSFLOW--GRASS. The user's manual is available as the README.md file in the repository. The GSFLOW executable and source code are available in the UMN-Hydro repository https://github.com/UMN-Hydro/GSFLOW-1.2.0 and from the USGS website https://water.usgs.gov/ogw/gsflow/. GRASS GIS 7.3+ is available from https://grass.osgeo.org/download/software/.

*Author contributions.* G.-H. C. Ng and A. D. Wickert contributed equally to this work; they together conceived of the project idea, wrote the majority of the paper, and built the visualization scripts. G.-H. C. Ng wrote the input file builder and plotting scripts. A. D. Wickert built the GRASS GIS extensions and integrated them with Ng's workflow. L. D. Somers and C. Cronkite-Ratcliff tested the code during its development and contributed their study cases. L. Saberi helped code the input-file builder scripts. R. Niswonger provided advice and support from the U.S. Geological Survey team that develops GSFLOW. J. M. McKenzie provided support for the Shullcas test case.

*Competing interests.* The authors declare that they have no conflict of interest.

*Acknowledgements.* The authors acknowledge all contributors to MODFLOW, PRMS, and GSFLOW, without whom this work would not be possible. Helpful comments from two anonymous referees guided manuscript improvements.

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
