# Peer review of "GSFLOW-GRASS v1.0.0: GIS-enabled hydrologic modeling of coupled groundwater–surface-water systems"

_Geoscientific Model Development, 2017_

## Referee Comment (RC1) · Anonymous Referee #1 · 26 Feb 2018

General comments: The manuscript presents the development of a suite of tools for preparing the input, submitting the simulation runs, and visualizing the output of the groundwater-surface water coupled GSFLOW model. The proposed suite of tools is developed exploiting the functionalities of the open-source GIS software GRASS and ad-hoc Python scripting. Authors tested the developed toolkit presenting test cases based on three catchments having different physiographic features. The manuscript is generally well written and with a logical and easy-to-follow structure.

While I concur with the authors on the potential of such kind of efforts to encourage the use of complex surface-subsurface coupled hydrological models, I question the actual

novelty and technical advancements presented in their work. Besides a suite of GIS extensions and scripts, the manuscript does not propose new technical solutions for the problem at-hand. For this reason, and for those elaborated below, I consider this contribution not suitable for GMD standard.

Specific comments:

1. In presenting/justifying their work, I think authors overlooked a bit too much the key technical issues preventing the widespread use of complex, physically based surface-subsurface coupled hydrological models in a decision-making framework. Here, I would argue that preparing the input is certainly a necessary and important step in the modeling exercise but not the most challenging one. In fact, if we agree that computationally efficient and numerically stable codes are needed to "promote science-driven decision making" then ad-hoc tools allowing a dynamic (e.g. in-situ visualization) inspection of such physical and numerical model response are probably much more needed, especially when we approach big-data problems. Saying that, I do not see the positioning of the effort presented in this manuscript with respect to these grand challenging tasks.

2. The outcome of the presented developments is clearly reflected in the results section. Here, authors describe three test cases illustrating the physical settings of each study area and discussing the potential outcome of a surface-subsurface coupled modeling approach. However, these results appear the repetition of the same exercise without much insight on the novelty of the proposed approach. For instance one could argue that such kind of plots can be simply obtained with some visualization scripts developed from scratch.

3. In a similar vein to the previous point, at the end of the introduction authors argue that the developments of such automated toolkit will enable rigorous testing. Absolutely true but a concrete path forward and tangible results are not presented in this context. Wouldn't it be an interesting way to demonstrate the utility of such kind of tools?

4. In several parts of the manuscript, authors refer to a similar work, i.e., Gardner et

al., which is currently under review for another journal. As the content of the cited work cannot be evaluated, these statements are unverifiable by the reader/reviewer, which is obviously not acceptable. Moreover, considering the potential overlap between the two contributions, as also acknowledged by the authors, it is not possible to weight the actual contribution of this work. For instance, one may ask if moving from ArcGis to GRASS or using ungridded versus gridded data would be enough to motivate an additional publication.

5. It appears that for some of the most critical parameters (e.g., Manning's parameter) authors present their approach referring to homogeneous values. In so doing, they advocate that field data on channel geometries come in a variety of forms difficult to accommodate in a generalized approach. Wouldn't it be the motivating reason for such geoscientific developments as the one presented here? Data fusion tools are in my opinion the key for facilitating the coherent ingestion of large source of information into a distributed model input data structure. An example along this line is represented by the work of Leonard and Duffy, 2013.

References:

Lorne Leonard, Christopher J. Duffy, Essential Terrestrial Variable data workflows for distributed water resources modeling, Environmental Modelling & Software,50, 85-96, doi: 10.1016/j.envsoft.2013.09.003, 2013.

Technical corrections:

1. Authors argue that models using triangulated irregular networks show better water balance performance over steep catchments. This is a quite interesting statement but ad-hoc citation is needed to substantiate this.

2. According to the author's opinion, PRMS does not implement Richards equation but instead applies an 'efficient' calculation to determine input and output for HRU. What's the meaning of 'efficient' here?

3. I do not see the precipitation lines in Figure 5-6-7.

---

## Referee Comment (RC2) · Anonymous Referee #2 · 7 Mar 2018

**General comments:**

Ng et al. present a GIS-based tool, GSFLOW-GRASS, that prepares input and runs the USGS hydrologic model GSFLOW. The authors provide comprehensive description of the GIS-based software, as well as the USGS hydrologic model. The paper is well-organized and well-written. As a modeler, I highly appreciate the authors' efforts in developing such tools, because "developing inputs to these models is usually time-consuming and requires extensive knowledge of software engineering, often prohibiting their use by many researchers and water managers". However, I do feel that GSFLOW-GRASS has limited capability in handling spatially-distributed, realistic input data (see

specific comments #2). All examples are shown without any measured discharge data. I acknowledge that model calibration is beyond the scope of this study, but it would be helpful if measured data could be shown, to demonstrate that the generated input can yield reasonable, if not accurate predictions. The authors also need to do a better job describing what are the "substantial new/novel concepts, ideas, or methods" in developing GSFLOW-GRASS, as required by GMD.

**Specific comments:**

1. Besides GSFLOW-GRASS and Gardner et al. (2017), there is another for-free software by Earthfx that can generate the inputs for GSFLOW. It is surprising that the authors did not review or describe this software. What are the differences between GSFLOW-GRASS and Earthfx software?

2. The GSFLOW-GRASS tool has limited capability in handling spatially-distributed, realistic input data. For example, P10, L24 "In its current form, v.gsflow.segments . . . allows the user to set a single channel width and Manning's n (in-channel roughness coefficient for flow resistance) across the whole domain;" P13, L29 "we do provide a script for uniformly applying a single climate data series over all HRUs to create climate_hru files;" and P14, L24 "most parameter values in printPRMSparamfile.py are preset. . .. This includes various soil and land-cover inputs, such as soil_type, cov_type, transp_end, and pt_alpha." There are GIS-based hydrologic model input tools that can take all different types of GIS input to generate spatially-distributed input data from national data-base or in situ measurements. For example, PIHMgis (Bhatt, G., Kumar, M. and Duffy, C. J., 2014: A tightly coupled GIS and distributed hydrologic modeling framework, *Environmental Modelling & Software*. **62**, 70—84.). The spatially-uniform approach and the preset default parameter values may prohibit GSFLOW from generating accurate predictions.
3. Page 3, L10 The authors state that triangulated irregular networks have better water balance performance. Why is that?

4. I am interested in the spin-up process described between L6 and L11 on Page 9. The authors describe the initial conditions as preliminary steady-state initial conditions. Usually the spin-up process is aimed to bring models to steady-state. If so, what is the difference between the before- and after-spin-up initial conditions?

5. Some of the parameters are shown without their definitions, for example pref_flow_den and sat_threshold. It would be add definitions.

**Technical comments:**

1. Figure 2 caption "Duncan runoff and fast interflow occurs in the preferential-flow reservoir." Should be "occur."

2. P10 L4 "This approach is complementary to the grid-cell HRU approach of (Gardner et al., 2017)." \citet command should be used instead of \citep.

3. P22 L8 "This allow users . . .." Should be "allows."

---

## Author Comment (AC1) · 7 Mar 2018

We thank the referee for their time in reviewing our manuscript and providing feedback. We were about to post this first reply (which addresses many of your comments) exactly when the review of the second referees came in. While we go through their review, we thought it could still be useful to post this in case it can generate some addition discussion. Also, we hoped that the referee can clarify: (1) in Specific Comment 1, what do you mean by "dynamic (e.g. in-situ visualization) inspection" and (2) in Technical Correction 3, do the blue precipitation lines not show up at all in your figures? See our responses below for more details related to these two questions

of clarification. We will post a complete response after going through all reviews and completing a full revision of the manuscript.

General comments: **The manuscript presents the development of a suite of tools for preparing the input, submitting the simulation runs, and visualizing the output of the groundwater–surface-water coupled GSFLOW model. The proposed suite of tools is developed exploiting the functionalities of the open-source GIS software GRASS and ad-hoc Python scripting. Authors tested the developed toolkit presenting test cases based on three catchments having different physiographic features. The manuscript is generally well written and with a logical and easy-to-follow structure. While I concur with the authors on the potential of such kind of efforts to encourage the use of complex surface-subsurface coupled hydrological models, I question the actual novelty and technical advancements presented in their work. Besides a suite of GIS extensions and scripts, the manuscript does not propose new technical solutions for the problem at-hand. For this reason, and for those elaborated below, I consider this contribution not suitable for GMD standard.**

We are glad the manuscript was found to be generally well-written and well-structured, but we are obviously very disappointed that the referee did not consider it to be suitable for GMD standard. Overall, we believe our practical, new utility toolbox for the USGS's GSFLOW model fits very well within GMD's solicitation for Model Description Papers: *"this type of paper may also describe model components and modules, as well as frameworks and utility tools used to build practical modelling systems."* However, this review brings to our attention that the current manuscript version fails to adequately explain the novel aspects and technical advances provided by our toolbox. We will clarify these points here and revise the manuscript accordingly.

While some of the individual scripting components within the toolbox may appear straightforward, our work's innovation is the entire bundled package. This includes

fully automated, integrated, robust, and open-source codes that cover everything from building topologically linked and robust hydrologic sub-basin domains and assembling model input parameters to visualizing model outputs within a self-consistent and efficient framework. The USGS's GSFLOW model couples two already-complex hydrologic models in a way that retains distinct structures of each, resulting in an integrated model of even greater complexity that presents new obstacles to the user. Our toolbox offers a solution that seamlessly handles the heterogeneity of this coupled model, thus tackling the grand challenge of accessibility plaguing many integrated modeling systems. As an important additional feature of accessibility, our toolbox is written using entirely free and open-source programming languages and software.

Within the toolbox, the major technical advancement is the development of a toolkit of streamlined GRASS-GIS extensions for building stream networks and sub-basins. While overland flow routing and the calculation of drainage basins from topography are standard GIS capabilities, their implementation is typically only semi-automated: systemic issues in most flow-routing algorithms require users to manually perform error-checks and corrections, which add a source of subjectivity and laborious processing time. Even more significantly, we know of no standard GIS tool predating ours that automatically builds topologically structured vectorized drainage networks that include information on adjacency and routing pathways through the network. We have now developed robust and automated algorithms that address these issues and have been tested with diverse DEMs as part of our model implementation examples. Furthermore, we have developed tools to link the irregular fluvial network to the regular grid used for the groundwater model component, permitting water to appropriately flow between the surface and subsurface. These advances now enable rapid, automated delineation of surface-water drainage networks across any generalized topography and any practical resolution, and this is conveniently done within a framework that readily links to implementation in a coupled hydrologic model.

We recognize that these new advances with our toolbox were not adequately expressed in the current manuscript version. They will be more fully described and clearly emphasized in our revision.

**Specific comments:**
**1. In presenting/justifying their work, I think authors overlooked a bit too much the key technical issues preventing the widespread use of complex, physically based surface-subsurface coupled hydrological models in a decision-making framework. Here, I would argue that preparing the input is certainly a necessary and important step in the modeling exercise but not the most challenging one. In fact, if we agree that computationally efficient and numerically stable codes are needed to "promote science-driven decision making" then ad-hoc tools allowing a dynamic (e.g. in-situ visualization) inspection of such physical and numerical model response are probably much more needed, especially when we approach big-data problems. Saying that, I do not see the positioning of the effort presented in this manuscript with respect to these grand challenging tasks.**

We believe that the need to create long model input files does present a major impediment to many potential users who may lack the necessary software skills or who might wish to carry out initial model tests before committing time to its use. Our toolbox offers a solution for this, which can greatly expand the reach of integrated hydrologic modeling. We think "dynamic (e.g. in-situ visualization) inspection" refers to run-time visualization of model results, but if this is incorrect, we ask that the referee please clarify. We acknowledge that run-time model visualization tools could help users decide whether to terminate a simulation early. However, given that there are currently no GSFLOW visualization modules available, our post-processing visualization scripts are already filling a key gap. Further, run-time plotting would slow down simulations and are typically difficult to implement on remotely accessed computer clusters. As such, while dynamic inspection is an unresolved challenge, our toolbox first addresses a more immediate need with this model.

**2. The outcome of the presented developments is clearly reflected in the results section. Here, authors describe three test cases illustrating the physical settings of each study area and discussing the potential outcome of a surface-subsurface coupled modeling approach. However, these results appear the repetition of the same exercise without much insight on the novelty of the proposed approach. For instance one could argue that such kind of plots can be simply obtained with some visualization scripts developed from scratch.**

A user can indeed develop from scratch similar visualization scripts, but we believe that the need to do so presents a major impediment to many potential users who may lack the necessary software skills or who might wish to carry out initial model tests before committing time to its use. Our toolbox includes pre- and post-processing capabilities that make the GSFLOW model widely accessible.

Further, we would like to clarify that the examples we present are not simply meant to showcase plotting capabilities, but they are also to demonstrate the robustness of our toolbox for diverse watershed settings. In particular, we show that the GIS extensions work out-of-the-box for a wide range of topographies. Each of the three test cases demonstrates particular technical challenges that our toolbox overcame.

The steep topography and narrow canyons of the Shullcas case would require an impractically high resolution to model using a regular gridded surface domain, leading to lengthy compute times. Our irregular HRU-based surface-water representation allows us to compute flow paths using high-resolution topography but reduce this into its fundamental surface-water hydrologic units, stream segments and subcatchments, for the model computations. While MODFLOW is run using a regular grid, this example case also tests our method of integrating the vectorized drainage network into the MODFLOW grid cell elevations in order to accurately simulate groundwater flow at lower (and more computationally-efficient) spatial resolutions. Assigning grid cell elevations

averaged over steep gradients produces spurious dams and lakes. Our toolbox's use of irregular drainage networks solved this problem by linking coarse-resolution subsurface grid cells to surface sub-basin units that give those cells a continuous downstream elevation gradient.

The Santa Rosa Island example represents another numerically challenging steep watershed. It also provided an opportunity to test our toolbox with small drainages covering just a few DEM grid cells and with irregular boundary geometries created by the coastline. This test case ensures that we can appropriately handle NULL values (for the ocean) while providing validation of the same methods required for the Shullcas watershed.

Finally, the Cannon River watershed in Minnesota includes deglacial topography that rivers have not yet organized into a linked valley network. Simple downslope flow-routing algorithms would typically fail for such settings, and "pit filling" can produce spurious results by inappropriately modifying the real topography. Our toolbox routes surface-water flow using the GRASS GIS "r.watershed" least-cost path algorithm, which is designed for such complex topography; we demonstrate that we are able to extend its capabilities to creating an automated topologically-correct and linked drainage network.

With these three distinct test scenarios, we therefore ensured that our toolbox passes stress tests in both very steep and very flat landscapes. In addition, the examples illustrate the range of hydrological processes that can be readily assessed with GSFLOW when facilitated by our toolbox, including tightly-coupled groundwater–surface-water interactions (Shullcas), episodic runoff driven by climate (Santa Rosa Island), and low-relief controls on water table depths (Cannon River). The reasons for the test case examples, and the toolbox capabilities that they highlight, were not made clear in the manuscript, and we will update it to ensure that readers recognize the purpose and extent of our testing.

[Figure]

**3. In a similar vein to the previous point, at the end of the introduction authors argue that the developments of such automated toolkit will enable rigorous testing. Absolutely true but a concrete path forward and tangible results are not presented in this context. Wouldn't it be an interesting way to demonstrate the utility of such kind of tools?**

We believe that once we clarify the novel aspects and technical contributions of the current work, it can be seen that this toolbox incorporates significant development work that constitutes a first, stand-alone manuscript. However, we agree with reviewer that a concrete path forward is key, and so we will revise the manuscript to provide greater detail on the types of rigorous model implementations and testing now possible with our toolbox.

**4. In several parts of the manuscript, authors refer to a similar work, i.e., Gardner et al., which is currently under review for another journal. As the content of the cited work cannot be evaluated, these statements are unverifiable by the reader/reviewer, which is obviously not acceptable. Moreover, considering the potential overlap between the two contributions, as also acknowledged by the authors, it is not possible to weight the actual contribution of this work. For instance, one may ask if moving from ArcGis to GRASS or using ungridded versus gridded data would be enough to motivate an additional publication.**

One of our co-authors, Rich Niswonger, is also a co-author of the Gardner et al. submitted manuscript and is also one of the GSFLOW developers at USGS. The Gardner et al. manuscript received encouraging reviews and is currently in revision. It may become available during the review of this current manuscript; otherwise, we may share it with permission from the first author. Our toolbox was created independently from the Gardner et al. work and can be distinguished in three important ways: 1) it creates a topography and stream network-based (ungridded) domain, offering alternative conceptual and computational implementations; 2) it fully automates all

aspects of the model pre- and post-processing, from building ALL model inputs to plotting model inputs and results; and 3) it utilizes entirely free and open-source tools, which significantly extends its reach to resource-limited users. As a developer of GSFLOW at the USGS, Niswonger views our GRASS-GSFLOW toolbox as a highly valuable complement to the alternative utilities by Gardner et al., which uses proprietary software to generate gridded domains for a subset of required model inputs. Having multiple approaches for developing models helps to serve a broader community of users, and future work using these different toolboxes will allow the community to evaluate different approaches (irregular vs. gridded domains, GRASS vs. ArcGIS algorithms, different stream network development methods, etc.).

**5. It appears that for some of the most critical parameters (e.g., Manning's parameter) authors present their approach referring to homogeneous values. In so doing, they advocate that field data on channel geometries come in a variety of forms difficult to accommodate in a generalized approach. Wouldn't it be the motivating reason for such geoscientific developments as the one presented here? Data fusion tools are in my opinion the key for facilitating the coherent ingestion of large source of information into a distributed model input data structure. An example along this line is represented by the work of Leonard and Duffy, 2013.**

Heterogeneous hydraulic conductivity is implemented in the Santa Rosa Island example. We agree with the reviewer that a heterogeneous Manning's $n$ parameter would also be important to include, and in response, we will develop a method to easily incorporate field data into a model for Manning's $n$ values in our revision. We intend to use a set of point measurements of Manning's $n$ values and assign values in the model domain units (HRU's) based on the nearest data point. This simple approach will provide the framework for more complex data integration procedures that could be developed later.

**References**: Lorne Leonard, Christopher J. Duffy, Essential Terrestrial Variable data workflows for distributed water resources modeling, Environmental Modelling & Software,50, 85-96, 10.1016/j.envsoft.2013.09.003, 2013.

**Technical corrections:**

**1. Authors argue that models using triangulated irregular networks show better water balance performance over steep catchments. This is a quite interesting statement but ad-hoc citation is needed to substantiate this.**

Triangulated irregular networks are often implemented with finite volume methods, which are guaranteed to conserve mass (Leveque et al. 2002). We intend to add references to the sentence as follows: "Models such as tRIBS (Vivoni et al., 2004) and PIHM (Qu and Duffy, 2007) utilize triangulated irregular networks for more computationally efficient representation of complex terrain (Goodrich et al., 1991) and for better water balance performance through the mass-conserving finite volume method (Leveque et al. 2002)."

New references:

Leveque, R. J. (2002), Finite Volume Methods for Hyperbolic Problems, Cambridge Univ. Press, New York.

Goodrich D.C., Woolhiser D.A., Keefer T.O. (1991). Kinematic routing using finite elements on a triangular irregular network. Water Resources Research 27(6): 995–1003.

**2. According to the author's opinion, PRMS does not implement Richards equation but instead applies an 'efficient' calculation to determine input and output for HRU. What's the meaning of 'efficient' here?**

By "efficient," we mean computationally more efficient. We will clarify this in our revision.

**3. I do not see the precipitation lines in Figure 5-6-7.**

We see the blue precipitation lines clearly in these figures. We are unsure why they do not appear for the referee and wonder if there is an issue with the file conversion. Could the referee please clarify whether the blue lines fail to appear at all, or whether they do but the referee does not find them to be clear enough?

---

## Referee Comment (RC3) · Anonymous Referee #3 · 8 Mar 2018

This article presents a user interface for the community hydrologic model GS-Flow using the community GIS package GRASS. This manuscript is well written and clearly presented. The interface is well documented. However, I am having trouble seeing the primary goal or take-home message for the readership of GMD. Is there a science or educational motivation for this work that allows users to do something they can't already do with the existing PRMS / Modflow approach? I like this manuscript and think it's well written but as currently framed, for me, misses this key point and reads much more like a user manual than a scientific article. I think revisions are needed to bring this critical point forward.

minor comments p1. lines 1-6. I think a better firs paragraph can help motivate this work's main takeaway point more clearly.

p1. line 9. GS-flow isn't an integrated model, it is coupled. Integrated models are defined to solve 3D richards' equation and the shallow water equations in an implicit framework to capture these coupled, nonlinear processes. This should be clarified in the revised manuscript.

p3. lines 7-11. Is this platform run in parallel? My understanding is not, nor is GS-flow parallel. I'm confused by this statement.

p3. line 10. I think the comment about triangulated grids providing better water balance is unsubstantiated and perhaps false. Most triangulated formulations are not even locally mass conservative which leads to local water balance error. GS-flow also uses structured grinding, which seems contradictory to these statements.

p3. line 24. again, GS-flow isn't integrated (or "integrated") and I don't know what 'integrated-coupled' even means.

p4. line 26+. This paragraph is short and confusing. Please reword.

p22. lines 7+. These don't strike me as conclusions and read a bit like an advertisement. To my central point, what is the scientific motivation and conclusions reached by this work. Reworking this paragraph would help that substantially.

---

## Referee Comment (RC4) · Anonymous Referee #1 · 12 Mar 2018

I would like to thank the authors for the first round of responses. I provide below some additional clarifications on few issues raised in the authors reply according to my previous comments.

1. I evaluated the paper not suitable for GMD for the lack of novelty and technical advancements. I did not question the utility itself of the proposed toolkit and I did not express any issue concerning the fit of the subject addressed in this work with the scope of the journal.

2. I highlighted some of the grand challenges (e.g., big-data problems) that, in my opinion, modelers are facing when performing large-scale high-resolution surfacesubsurface coupled simulations. In this context, in-situ visualization (i.e., the use of libraries to dynamically connect running simulations and graphical outputs) is of particular interest in the geoscience community. My concern was that the paper did not even mention/discuss how the methodology they are proposing reconcile with such grand challenge.

3. I questioned the insights gained from the three test cases. Authors reply that each of them demonstrates particular technical challenges solved by the proposed toolkit where 'other' approaches would fail. If this is the case, you need to provide evidence, from a simple visual inspection of Figure 5-6-7 I do not see it.

4. I raised the issue of a cited publication, which is currently under review for another journal. Authors' argumentation is that the work received positive comments and it will be likely out very soon. At this time it is not. Therefore, it is not possible for any reviewer or person eager to comment on the manuscript to have an idea on the content of the cited work. In other words, being aware of these positive comments on the contribution, you should have included in the discussion later in the review process. . ...

---

## Author Comment (AC2) · 26 Apr 2018

RESPONSE TO REVIEWER 1

**Boldface: Reviewer 1's original comments and additional clarifications (latter added to the corresponding original comment)**

*Italics:* *Our response*

*We thank the referee for their time in reviewing our manuscript and providing feedback.*

[Figure]

**General comments:** **The manuscript presents the development of a suite of tools for preparing the input, submitting the simulation runs, and visualizing the output of the groundwater–surface-water coupled GSFLOW model. The proposed suite of tools is developed exploiting the functionalities of the open-source GIS software GRASS and ad-hoc Python scripting. Authors tested the developed toolkit presenting test cases based on three catchments having different physiographic features. The manuscript is generally well written and with a logical and easy-to-follow structure. While I concur with the authors on the potential of such kind of efforts to encourage the use of complex surface-subsurface coupled hydrological models, I question the actual novelty and technical advancements presented in their work. Besides a suite of GIS extensions and scripts, the manuscript does not propose new technical solutions for the problem at-hand. For this reason, and for those elaborated below, I consider this contribution not suitable for GMD standard.**

**(Additional clarification:) I evaluated the paper not suitable for GMD for the lack of novelty and technical advancements. I did not question the utility itself of the proposed toolkit and I did not express any issue concerning the fit of the subject addressed in this work with the scope of the journal.**

*We are glad the reviewer found that our manuscript was generally well-written and fits within the GMD scope, but we are obviously very disappointed that the referee did not consider it to be suitable for GMD standard. Our work does offer new technical solutions for making integrated hydrologic modeling more accessible; this review brought to our attention that the first manuscript version indeed failed to explain these novel aspects and technical contributions and instead focused too much on simply documenting the contents of the toolbox. We appreciate that the reviewer raised this issue, and we have substantially revised the manuscript to address this serious shortcoming in the original presentation.*

*In particular, we first clarified that while some of the individual scripting components within the toolbox may appear straightforward, our work's innovation is the entire bundled package. Our substantially edited Introduction now emphasizes that existing software for integrated hydrologic models fail to provide freely accessible toolkits that fully cover pre- to post-processing steps (p. 2 Lines 22-30), and that GSFLOW-GRASS addresses that critical gap stymieing the use of integrated hydrologic models (p.2 Line 31- p. 3 Line 6).*

*We then explained that the major technical advancement of our work was to create a new set of GRASS-GIS tools that can robustly and automatically generate surface and subsurface model domains suitable for hydrological modeling – these are critical for GSFLOW-GRASS to be widely applicable to a diverse range of hydro(geo)logical settings. We now realize that the original manuscript version documented these new GRASS GIS extensions but provided almost no background on the challenges of creating robust and automated tools, which have led to a general unavailability of such solutions predating our toolbox. A new paragraph has been added to the Introduction to present the technical advancements with these GRASS GIS extensions (p. 3 Lines 7-21). Further, we have entirely re-written Section 3.2 on the GRASS GIS domain builder (p. 10-12), so that it now explicitly describes what was implemented to solve specific known problems with stream network delineation. Finally, we also made major changes to Section 4 on the Examples, in order to explain how each example demonstrates a different strength and capability of the domain builder (specifically, p. 21 Lines 3-10 for Shullcas, p. 21 Line 32- p. 22 Line 4 for Santa Rosa, and p. 23 Line 10- p. 24 Line 4 for Cannon River). These examples demonstrate how GSFLOW-GRASS handles known challenges with various degrees of drainage integration, landscape relief, and grid resolution, as well as the presence of irregular coastal boundaries.*

*The technical advancements of our GRASS GIS tools were recently highlighted as a new release feature on the GRASS GIS website: https:// trac.osgeo.org/ grass/ wiki/ Grass7/ NewFeatures74 (including a figure with our Cannon River watershed example)*

*– see the screenshot in Figure 1.*

*In addition to major revisions to the Introduction, GRASS GIS Domain Builder section, and Examples section, we essentially re-wrote the Abstract and Conclusion to highlight these new and technical contributions.*

**Specific comments:**

**1. In presenting/justifying their work, I think authors overlooked a bit too much the key technical issues preventing the widespread use of complex, physically based surface-subsurface coupled hydrological models in a decision-making framework. Here, I would argue that preparing the input is certainly a necessary and important step in the modeling exercise but not the most challenging one. In fact, if we agree that computationally efficient and numerically stable codes are needed to "promote science-driven decision making" then ad-hoc tools allowing a dynamic (e.g. in-situ visualization) inspection of such physical and numerical model response are probably much more needed, especially when we approach big-data problems. Saying that, I do not see the positioning of the effort presented in this manuscript with respect to these grand challenging tasks.**

**(Additional clarification:) I highlighted some of the grand challenges (e.g., big-data problems) that, in my opinion, modelers are facing when performing large-scale high-resolution surface-subsurface coupled simulations. In this context, in-situ visualization (i.e., the use of libraries to dynamically connect running simulations and graphical outputs) is of particular interest in the geoscience community. My concern was that the paper did not even mention/discuss how the methodology they are proposing reconcile with such grand challenge.**

*We believe that the need to create long model input files does in fact present a critical challenge for many potential users who may lack the necessary software skills or who might wish to carry out initial model tests before committing time to its use. In support*

*of the value of our toolbox, we would like to share that over just April 9 to 22 (the maximum length of time tracked by GitHub), our GSFLOW-GRASS repository received 173 views and 22 unique visitors, and one user from a major research university sent us an email that opened with "Thank you so much for sharing the GSFLOW-GRASS toolkit. This toolkit really relieves my struggle of preparing inputs." - and all of this is with absolutely no effort to advertise our toolkit.*

*However, we do acknowledge that there are other grand challenges to integrated hydrologic modeling, and we appreciate the reviewer's suggestion about in-situ visualization. In response, we have expanded our toolbox to include an additional tool "plotGSFLOWTimeSeries_Runtime.py", which is now incorporated into our Run script to generate runtime time series plots of simulated discharge and precipitation. This new capability is described in the revised manuscript on p. 17 Lines 12-13 and p. 18 Lines 10-13.*

**2. The outcome of the presented developments is clearly reflected in the results section. Here, authors describe three test cases illustrating the physical settings of each study area and discussing the potential outcome of a surface-subsurface coupled modeling approach. However, these results appear the repetition of the same exercise without much insight on the novelty of the proposed approach. For instance one could argue that such kind of plots can be simply obtained with some visualization scripts developed from scratch.**

**(Additional clarification:) I questioned the insights gained from the three test cases. Authors reply that each of them demonstrates particular technical challenges solved by the proposed toolkit where 'other' approaches would fail. If this is the case, you need to provide evidence, from a simple visual inspection of Figure 5-6-7 I do not see it.**

*A user can indeed develop from scratch similar visualization scripts, but we believe that*

*the need to do so presents a major impediment to many potential users who may lack the programming knowledge or may not be able or wish to invest the time for it. Our toolbox includes pre- and post-processing capabilities that make the GSFLOW model widely accessible.*

*We do acknowledge, however, that the original manuscript version presented the 3 examples in a way that did not describe how each one presents a particular challenge that the new GRASS GIS extensions address. As mentioned earlier in this response, we addded paragraphs to each example to do so, specifically: p. 21 Lines 3-10 for Shullcas, p. 21 Line 32- p. 22 Line 4 for Santa Rosa, and p. 23 Line 10- p. 24 Line 4 for Cannon River. These aspects are also summarized in the Conclusions: "The results show that the new and automated GRASS GIS extensions can automatically and consistently build topologically complete linked surface and subsurface flow domains in settings that are typically challenging for standard GIS tools, including steep topographies, irregular coastal boundaries, and low-relief terrains that lack integrated drainage." (p. 24 Lines 31-33).*

*We further realized that we should have more clearly highlighted the types of hydrologic / hydrogeologic processes of management concern that can be evaluated with aid of GSFLOW-GRASS through each example; we have edited the last paragraph of each example to better express these types of processes and how they are depicted with the GSFLOW-GRASS visualization tools. These processes are also summarized in the edited Conclusion: "these examples further demonstrate that GSFLOW-GRASS is a flexible tool for investigating the role of groundwater-surface water interactions in modulating dry-season discharge, controlling runoff in erosion-prone landscapes, and imposing possible water-quality threats in agricultural and recreational watersheds." (p. 25 Lines 1-3).*

**3. In a similar vein to the previous point, at the end of the introduction authors argue that the developments of such automated toolkit will enable rigorous test-**

ing. **Absolutely true but a concrete path forward and tangible results are not presented in this context. Wouldn't it be an interesting way to demonstrate the utility of such kind of tools?**

*As we discussed in our response to the previous comment 2., we realize that our original manuscript version failed to adequately explain how the 3 examples demonstrated the utility of GSFLOW-GRASS. We have substantially edited our manuscript to now explain how each of these examples showcase a particular capability of the domain builder as well as a different scientific and/or resource management concern affected by groundwater-surface interactions that can be probed with the aid of GSFLOW-GRASS (see manuscript lines referenced above). As a concrete path forward, we also suggest future tests of the performance of ungridded surface domains with GSFLOW-GRASS (p. 9 Lines 6-7), and we list potential future extensions of GSFLOW-GRASS in the Conclusion (p. 25 Lines 9-17).*

**4. In several parts of the manuscript, authors refer to a similar work, i.e., Gardner et al., which is currently under review for another journal. As the content of the cited work cannot be evaluated, these statements are unverifiable by the reader/reviewer, which is obviously not acceptable. Moreover, considering the potential overlap between the two contributions, as also acknowledged by the authors, it is not possible to weight the actual contribution of this work. For instance, one may ask if moving from ArcGis to GRASS or using ungridded versus gridded data would be enough to motivate an additional publication.**

**(Additional clarification:) I raised the issue of a cited publication, which is currently under review for another journal. Authors' argumentation is that the work received positive comments and it will be likely out very soon. At this time it is not. Therefore, it is not possible for any reviewer or person eager to comment on the manuscript to have an idea on the content of the cited work. In other words, being aware of these positive comments on the contribution, you should have**

[Figure]

**included in the discussion later in the review process.**

*We now recognize that it is unreasonable to expect a reader to follow detailed comparisons with an unpublished and unavailable manuscript. One of our co-authors, Rich Niswonger, is also a co-author of the Gardner et al. submitted manuscript (as well as one of the GSFLOW developers at USGS). He reports that the manuscript is still in re-review at this time. Because the Gardner et al. work is not actually central to GSFLOW-GRASS, we minimized our discussion of that work – mostly in the Introduction and the "User-specified settings and model inputs" section. We now only mention it as one of the other software options to facilitate integrated hydrologic model implementation that do not offer a complete pre- to post-processing set of tools. We view it as a real benefit for the community to have these two new packages with different features (differences in discretization, handling of input data, availability of post-processing tools, and software platforms), so that users can choose the one most suitable for their application. Rich Niswonger's role as co-author has not been to develop the GSFLOW-GRASS software, but it has been to ensure that GSFLOW-GRASS is not overly duplicative of the package by Gardner et al. (of which he is a developer), and that GSFLOW-GRASS is constructed in a way that the USGS considers will be effective for increasing the accessibility of GSFLOW. The multiple softwares (free and proprietary) available for implementing MODFLOW serve as an example that having more than one software package for a model can be valuable for supporting an extensive user-base.*

**5. It appears that for some of the most critical parameters (e.g., Manning's parameter) authors present their approach referring to homogeneous values. In so doing, they advocate that field data on channel geometries come in a variety of forms difficult to accommodate in a generalized approach. Wouldn't it be the motivating reason for such geoscientific developments as the one presented here? Data fusion tools are in my opinion the key for facilitating the coherent ingestion**

**of large source of information into a distributed model input data structure. An example along this line is represented by the work of Leonard and Duffy, 2013.**

**References: Lorne Leonard, Christopher J. Duffy, Essential Terrestrial Variable data workflows for distributed water resources modeling, Environmental Modelling & Software, 50, 85-96, 10.1016 /j.envsoft.2013.09.003, 2013.**

*Our original GSFLOW-GRASS version did include an option for spatially heterogeneous hydraulic conductivity inputs, which was implemented in the Santa Rosa Island example. However, we agree with the reviewer that a heterogeneous channel width and Manning's n parameter would also be important to include, and in response, we modified the toolbox to accommodate this through the Settings file (as described on p. 9 Line 29 and on p. 17 Line 5).*

*We acknowledge the value of linking integrated modeling with existing databases for model inputs, but we consider this beyond the scope of our current work, which aims to provide a generalized solution for implementing GSFLOW-GRASS. We reference software tools that do fuse data products with hydrologic models, including Leonard and Duffy (2013) as suggested by the reviewer (p. 10 Lines 3-5); we then point out that these databases are generally only available in observation-rich places and thus we do not include any in the first GSFLOW-GRASS version, which serves as a general basis for further development (p. 10 Lines 7-8). Our revised conclusion discusses future extensions of GSFLOW-GRASS to include links to spatial databases to generate model inputs (p. 25 Lines 12-13).*

*Although GSFLOW-GRASS currently does not offer spatially heterogeneous solutions for inputs beyond hydraulic conductivity and Manning's n, we created a new GRASS GIS tool, v.gsflow.mapdata, in response to the reviewer's valid concern about it. This tool can take any spatially variable data in a raster or vector GIS format and map it to one of the GSFLOW discretization structures: sub-basin HRUs for PRMS surface-water processes, regular grid cells for MODFLOW groundwater processes,*

*gravity reservoirs that link the HRUs and MODFLOW grid cells, or stream segments or reaches for MODFLOW streamflow processes. This helps users add data from any source to the GSFLOW-GRASS data structures for input into the model. The new v.gsflow.mapdata tool is presented on p. 9 Line 30- p. 10 Line 8. Throughout the rest of the revised manuscript, we also mention how this tool can be implemented to create specific spatially distributed inputs, including the climate inputs and soil / land-cover parameters (p. 12 Line 26, p. 14 Lines 19 and 29, p. 15 Lines 16 and 29, and p. 17 Line 1).*

**Technical corrections:**

**1. Authors argue that models using triangulated irregular networks show better water balance performance over steep catchments. This is a quite interesting statement but ad-hoc citation is needed to substantiate this.**

*We realize that we left out some details and should have specified that TINs show better water balance performance IF they are implemented with the finite volume method (because the finite volume method is mass-conserving), and that TINs cover complex surface domain more efficiently (fewer units) than grid cells. We edited the text to say all of this on p. 4 Line 15-16.*

**2. According to the author's opinion, PRMS does not implement Richards equation but instead applies an 'efficient' calculation to determine input and output for HRU. What's the meaning of 'efficient' here?**

*By "efficient," we mean computationally fast. We clarified this on p. 6 Lines 5-6.*

**3. I do not see the precipitation lines in Figure 5-6-7.**

*As we mentioned in our preliminary response to the reviewer: we see the blue pre-*

*cipitation lines clearly in these figures. We are unsure why they do not appear for the reviewer and wonder if there is an issue with the file conversion. If more information could be provided (e.g., do the blue lines fail to appear at all, or do they appear but just not clearly?), we can address it.*

——————————————————

[Figure]

**Fig. 1.** Screenshot of GRASS GIS' new features - includes GSFLOW-GRASS add-ons

---

## Author Comment (AC3) · 26 Apr 2018

RESPONSE TO REVIEWER 2

**Boldface: Reviewer 2's original comments**

*Italics: Our response*

*We thank the referee for their time in reviewing our manuscript and providing feedback.*

**General comments:**

Ng et al. present a GIS-based tool, GSFLOW-GRASS, that prepares input and runs the USGS hydrologic model GSFLOW. The authors provide comprehensive description of the GIS-based software, as well as the USGS hydrologic model. The paper is well-organized and well-written. As a modeler, I highly appreciate the authors' efforts in developing such tools, because "developing inputs to these models is usually time-consuming and requires extensive knowledge of software engineering, often prohibiting their use by many researchers and water managers". However, I do feel that GSFLOW-GRASS has limited capability in handling spatially-distributed, realistic input data (see specific comments #2). All examples are shown without any measured discharge data. I acknowledge that model calibration is beyond the scope of this study, but it would be helpful if measured data could be shown, to demonstrate that the generated input can yield reasonable, if not accurate predictions. The authors also need to do a better job describing what are the "substantial new/novel concepts, ideas, or methods" in developing GSFLOW-GRASS, as required by GMD.

*We are glad that the reviewer found our manuscript to be generally well-organized and appreciated our effort to make integrated hydrologic modeling more accessible. Indeed, we believe model-users are already seeing the utility of our work; just over April 9 to 22 (the maximum length of time tracked by GitHub), our GSFLOW-GRASS repository received 173 views and 22 unique visitors, and one user from a research university sent us an email that opened with "Thank you so much for sharing the GSFLOW-GRASS toolkit. This toolkit really relieves my struggle of preparing inputs." - and all of this is with absolutely no effort to advertise our toolkit.*

*After this review, we do now recognize that our original manuscript version failed to adequately explain the novel aspects and new technical advances provided by our toolbox. We appreciate that the reviewer raised this issue, and we have substantially revised the manuscript to address this serious shortcoming in the presentation.*

*In particular, we first clarified that while some of the individual scripting components*
*within the toolbox may appear straightforward, our work's innovation is the entire bundled package. Our substantially edited Introduction now emphasizes that existing software for integrated hydrologic models fail to provide freely accessible tool-sets that fully cover pre- to post-processing steps (p. 2, lines 22-30), and that GSFLOW-GRASS addresses that critical gap stymieing the use of integrated hydrologic models (p.2 lines 31-34, p. 3 lines 1-6).*

*We then explained that the major new method advancement was to create a new set of GRASS-GIS tools that can robustly and automatically generate surface and subsurface model domains suitable for hydrological modeling, which was critical for GSFLOW-GRASS to be widely applicable to a diverse range of hydro(geo)logical settings. We now realize that the original manuscript version documented these new GRASS GIS extensions but provided almost no background on the challenges of creating robust and automated tools, which has led to a general unavailability of such solutions predating our toolbox. A new paragraph has been added to the Introduction to present this technical advancement (p. 3 Lines 7-21). Further, we have entirely re-written Section 3.2 on the GRASS GIS domain builder (p. 10-12), so that it now explicitly describes what was implemented to solve specific known challenges for stream network delineation. Finally, we also made major changes to Section 4 on the Examples, in order to explain how each example demonstrates a different strength and capability of the domain builder (specifically, p. 21 Lines 3-10 for Shullcas, p. 21 Line 32- p. 22 Line 4 for Santa Rosa, and p. 23 Line 10- p. 24 Line 4 for Cannon River). These examples demonstrate how GSFLOW-GRASS handles known challenges with various degrees of drainage integration, landscape relief, and grid resolution, as well as the presence of irregular coastal boundaries.*

*The new technical advancements of our GRASS GIS tools were recently highlighted as a new release feature on the GRASS GIS website: https:// trac.osgeo.org/ grass/ wiki/ Grass7/ NewFeatures74 (including a figure with our Cannon River watershed example) - see screenshot in Figure 1.*

*In addition to major revisions to the Introduction, GRASS GIS Domain Builder section, and Examples section, we essentially re-wrote the Abstract and Conclusion to highlight these new and technical contributions.*

*The reviewer also brought up an excellent point about the importance of spatially distributed, realistic input data. This prompted us to add new capabilities to the toolbox to accommodate spatially variable inputs, which we describe in greater detail under "Specific comment #2."*

*We appreciate the reviewer acknowledging that model calibration is beyond the scope of this work. However, the reviewer's comment about comparing against observations made us realize that we needed to be clearer that the aim of the examples is not simulate realistic results for each site but is instead to demonstrate the robustness of the domain builder for a range of settings (as pointed out above) and the types of processes that can be explored with GSFLOW-GRASS. We re-wrote the opening of the Examples section to reflect this (p. 18 Line 32- p. 19 Line 5), and we added a brief discussion of how the automated GSFLOW-GRASS toolbox jump-starts the model implementation and thus facilitates additional calibration and sensitivity analysis (p. 19 Line 18- p. 20 Line 7).*

*Although the examples are not meant to serve as realistic results, we do recognize the value of comparing our example simulations against actual observations. The Cannon River site is the only one of the 3 example sites that has publicly available discharge data, and so we now show it in our revised Figure 7. In addition to discussing where the uncalibrated model does reasonably match observations, we also point out how the major discrepancies found in the default implementation can be useful for guiding parameter calibration (p. 24 Lines 9-15).*

**Specific comments:**

**1. Besides GSFLOW-GRASS and Gardner et al. (2017), there is another for-free**

software by Earthfx that can generate the inputs for GSFLOW. It is surprising that
the authors did not review or describe this software. What are the differences
between GSFLOW-GRASS and Earthfx software?

*We were aware of the Earthfx software and did mention it in the original manuscript
("However, with the exception of a single for-fee software product by Earthfx (http:
// www.earthfx.com/ ), these tools neither help users conceptually link the different pro-
cess domains represented in PRMS and MODFLOW nor generate the files that link
them." original p. 2 Lines 21-23). The reviewer wrote that Earthfx is "for-free" but
we wonder if that is a typo and is supposed to be "for-fee"? Looking again at the
Earthfx website, we do not see anywhere that one can freely download the software
(http:// www.earthfx.com/ VIEWLOG/ Components/ VLGSFLOW.aspx) and only see a
page pointing interested users to call for pricing information (http:// www.earthfx.com/
VIEWLOG/ Details/ Pricing.aspx says: "Call for pricing," "University discount 30%" for
the VIEWLOG software, which includes VL-GSFLOW).*

*Given that the software is fee-based and cannot be accessed without contacting the
vendor, we found it difficult to review in detail. We did add to the text that Earthfx offers
"full support" for GSFLOW, specifically through its "VIEWLOG" software package (p.
2 Lines 25-26). We then point out that "the community still lacks a free and complete
package spanning pre- to post-processing for heterogeneous surface and subsurface
domains" (p. 2 Lines 27-28), motivating the development of GSFLOW-GRASS.*

2. The GSFLOW-GRASS tool has limited capability in handling spatially-
distributed, realistic input data. For example, P10, L24 "In its current form,
v.gsflow.segments ... allows the user to set a single channel width and Man-
ning's n (in-channel roughness coefficient for flow resistance) across the whole
domain;" P13, L29 "we do provide a script for uniformly applying a single climate
data series over all HRUs to create climate_hru files;" and P14, L24 "most param-
eter values in printPRMSparamfile.py are preset. . .. This includes various soil

**and land-cover inputs, such as soil_type, cov_type, transp_end, and pt_alpha."
There are GIS-based hydrologic model input tools that can take all different types
of GIS input to generate spatially-distributed input data from national data-base
or in situ measurements. For example, PIHMgis (Bhatt, G., Kumar, M. and Duffy,
C. J., 2014: A tightly coupled GIS and distributed hydrologic modeling frame-
work, Environmental Modelling & Software. 62, 70-84.). The spatially-uniform
approach and the preset default parameter values may prohibit GSFLOW from
generating accurate predictions.**

*Our original GSFLOW-GRASS version did include an option for spatially heteroge-
neous hydraulic conductivity, which was implemented in the Santa Rosa Island exam-
ple. However, we agree with the reviewer that the general lack of ability for our original
toolbox to generate spatially variable data inputs (beyond hydraulic conductivity) was
a weakness in the original submission. To address this, we modified the toolbox to
accommodate heterogeneous channel width and Manning's n parameter through the
Settings file (as explained on p. 9 Line 29 and on p. 17 Line 5). For other inputs, we cre-
ated a new GRASS GIS tool, v.gsflow.mapdata, which can take any spatially variable
data in a raster or vector GIS format and map it to one of the GSFLOW discretization
structures: sub-basin HRUs for PRMS surface-water processes, regular grid cells for
MODFLOW groundwater processes, gravity reservoirs that link the HRUs and MOD-
FLOW grid cells, or stream segments or reaches for MODFLOW streamflow processes.
This allows users to incorporate data from any source to the GSFLOW-GRASS data
structures for input into the model. The new v.gsflow.mapdata tool is presented on p. 9
Line 30- p. 10 Line 8. Throughout the rest of the paper, we also mention how this tool
can be implemented to create specific spatially distributed inputs, including the climate
inputs and soil / land-cover parameters mentioned by the reviewer (p. 12 Line 26, p.
14 Lines 19 and 29, p. 15 Lines 16 and 29, and p. 17 Line 1).*

*For linking integrated hydrologic modeling with existing databases for model inputs,
we do see the clear utility but consider this to be beyond the scope of our current work,*

*which aims to provide a generalized solution for implementing GSFLOW-GRASS. We reference software tools that focus on this capability, including Bhatt et al. (2014) as suggested by the reviewer (p. 10 Lines 3-5); we then point out that these databases are typically only available in observation-rich places and thus we do not include it in the first GSFLOW-GRASS version, which provides a general basis for further development (p. 10 Lines 7-8). Our revised conclusion discusses future extensions of GSFLOW-GRASS to include links to spatial databases to generate model inputs (p. 25 Lines 11-12).*

**3. Page 3, L10 The authors state that triangulated irregular networks have better water balance performance. Why is that?**

*We realized that we left out some details and should have specified that triangulated irregular networks (TINs) show better water balance performance IF they are implemented with the finite volume method (because the finite volume method is mass-conserving), and that TINs cover complex surface domain more efficiently (fewer units) than grid cells. We edited the text to say all of this on p. 4 Line 15-16.*

**4. I am interested in the spin-up process described between L6 and L11 on Page 9. The authors describe the initial conditions as preliminary steady-state initial conditions. Usually the spin-up process is aimed to bring models to steady-state. If so, what is the difference between the before- and after-spin-up initial conditions?**

*We realize we could have explained the steady-state vs. spin-up process much more clearly. The preliminary steady-state run is just for the MODFLOW (groundwater) component; the steady-state groundwater head results are then used to initialize the spin-up for the fully coupled (surface AND subsurface model domains). We edited the text to better explain this on p. 10 Lines 13-17.*

**5. Some of the parameters are shown without their definitions, for example pref_flow_den and sat_threshold. It would be add definitions.**

*We added Table 1 to define all GSFLOW parameters mentioned in the manuscript.*

**Technical comments:**

**1. Figure 2 caption "Duncan runoff and fast interflow occurs in the preferential-flow reservoir." Should be "occur."**

*We implemented this edit.*

**2. P10 L4 "This approach is complementary to the grid-cell HRU approach of (Gardner et al., 2017)." \citet command should be used instead of \citep.**

*The text was edited in the revision and no longer has this issue.*

**3. P22 L8 "This allow users ...." Should be "allows."**

*We implemented this edit.*

[Figure]

**Fig. 1.** Screenshot of GRASS GIS' new features - includes GSFLOW-GRASS add-ons

---

## Author Comment (AC4) · 26 Apr 2018

RESPONSE TO REVIEWER 3

**Boldface: Reviewer 3's original comments**

*Italics: Our response*

*We thank the referee for their time in reviewing our manuscript and providing feedback.*

**This article presents a user interface for the community hydrologic model GS-**

[Figure]

**Flow using the community GIS package GRASS. This manuscript is well written and clearly presented. The interface is well documented. However, I am having trouble seeing the primary goal or take-home message for the readership of GMD. Is there a science or educational motivation for this work that allows users to do something they can't already do with the existing PRMS / Modflow approach? I like this manuscript and think it's well written but as currently framed, for me, misses this key point and reads much more like a user manual than a scientific article. I think revisions are needed to bring this critical point forward.**

*We are glad that the reviewer liked our manuscript and found it to be well-written. We also appreciate the reviewer bringing to our attention that we needed to clarify the scientific merit of the work, which we realize was very inadequately described in the original manuscript. Scientific understanding of integrated hydrologic processes has been stymied by the inaccessibility of complex models for many researchers and resource managers; the major advancement of our work is to provide a robust and flexible software for implementing the USGS's groundwater and surface-water flow model - GSFLOW - across diverse hydro(geo)logic settings; importantly, this required the development of new GRASS GIS extensions that overcome common obstacles in creating automated and reproducible surface and subsurface model domains for integrated hydrologic models. We now realize that many of these points were almost entirely missing from the original manuscript version, and we have substantially revised the manuscript to address this major shortcoming.*

*Our edited Introduction now emphasizes that existing software for integrated hydrologic models do not provide freely accessible toolkits that fully cover pre- to post-processing steps (p. 2, lines 22-30), and that GSFLOW-GRASS addresses that gap (p.2 lines 31-34, p. 3 lines 1-6). The original manuscript version documented the new GRASS GIS extensions, but admittedly, it did so much like a manual and provided almost no background on the challenges of creating robust and automated tools – which have led to a general unavailability of such solutions predating our toolbox. A new*

*paragraph has been added to the Introduction to present the technical advancements with these GRASS GIS extensions (p. 3 Lines 7-21). Further, we have entirely re-written Section 3.2 on the GRASS GIS domain builder (p. 10-12), so that it now explicitly describes what was implemented to solve specific known problems with stream network delineation. Finally, we also made major changes to Section 4 on the Examples, in order to explain how each model implementation demonstrates a different capability of the domain builder (specifically, p. 21 Lines 3-10 for Shullcas, p. 21 Line 32- p. 22 Line 4 for Santa Rosa, and p. 23 Line 10- p. 24 Line 4 for Cannon River). In particular, these examples demonstrate how GSFLOW-GRASS handles known challenges with various degrees of drainage integration, landscape relief, and grid resolution, as well as the presence of irregular coastal boundaries.*

**minor comments p1. lines 1-6. I think a better firs paragraph can help motivate this work's main takeaway point more clearly.**

*We have entirely rewritten the first paragraph (as well as most of the rest of the Introduction section) to highlight key motivations for "streamlined access to models that integrate surface and subsurface processes," which includes tools that address "challenges of of generating computationally robust surface and sub-surface model domains" (p. 1 Lines 1-8).*

**p1. line 9. GS-flow isn't an integrated model, it is coupled. Integrated models are defined to solve 3D richards' equation and the shallow water equations in an implicit framework to capture these coupled, nonlinear processes. This should be clarified in the revised manuscript.**

*We appreciate the reviewer's rigor in distinguishing between the use of the terms "integrated" and "coupled." Indeed, GSFLOW is a "coupled" model in that it that employs an iterative method to link the base codes of PRMS and MODFLOW. We chose to also*

*refer to GSFLOW as an "integrated" model following the USGS's use of that term. The GSFLOW manual (Markstrom et al., 2008) distinguishes between two types of "integrated" models - "fully integrated" models that simultaneously solve surface and subsurface domain equations (what the reviewer calls "integrated") and "coupled regions" models that iterate between solutions for each set of equations (what the reviewer calls "coupled"). Thus, the USGS presents GSFLOW as an "integrated" model of the "coupled regions" type.*

*In order not to confuse model users, who will likely also be looking at the GSFLOW manual, we elected to adopt the same terminology as the USGS. However, we do now clarify that GSFLOW is not "fully integrated" but is instead "coupled" on p. 2 Lines 18-20. In fact, as a coupled model, GSFLOW still requires all the individual input files of both underlying models, which accounts for much of the laborious and time-consuming process of implementing GSFLOW. This motivation for a bundled toolkit solution for coupled models is now highlighted on p. 2 Lines 20-22.*

**p3. lines 7-11. Is this platform run in parallel? My understanding is not, nor is GS-flow parallel. I'm confused by this statement.**

*We realize from this comment that our original wording was confusing. No, GSFLOW-GRASS is not set up to run in parallel. The statement referenced by the reviewer was simply referring to the general advantages of using gridded domains, one of which is easier porting to parallel systems if desired. We edited the text to now read: "In general, gridded domains are easier to construct and extend to parallelized computational systems, and they allow flexible spatial specification of soil and land-cover heterogeneity." (p. 4 Lines 11-13).*

**p3. line 10. I think the comment about triangulated grids providing better water balance is unsubstantiated and perhaps false. Most triangulated formulations**

**are not even locally mass conservative which leads to local water balance error. GS-flow also uses structured grinding, which seems contradictory to these statements.**

*We realized that we left out a critical detail - we meant to specify that TINs show better water balance performance IF they are implemented with the finite volume method (because the finite volume method is mass-conserving). We edited the text to clarify the water balance advantage with finite volume on p. 4 Line 15-16. GSFLOW-GRASS uses rectangular grid cells for the MODFLOW subsurface component, but it uses irregular sub-basin HRUs for the PRMS surface component. We consolidated all the information about the GSFLOW-GRASS domain discretization in the first paragraph of our Methods section to make this clearer (p. 9 Lines 2-7).*

**p3. line 24. again, GS-flow isn't integrated (or "integrated") and I don't know what 'integrated-coupled' even means.**

*See our above explanation of our use of "integrated," "fully integrated," and "coupled." However, we do realize that the wording mentioned by the reviewer was awkward and removed it on p. 3 Line 26.*

**p4. line 26+. This paragraph is short and confusing. Please reword.**

*We believe the reviewer was confused by the vagueness of "different modes" and the ambiguous "they" in the original text ("Table 1 in the GSFLOW manual (Markstrom et al., 2008) lists all PRMS modules, MODFLOW stress packages, and GSFLOW modules used by GSFLOW in the different modes. This section includes a brief description of the main processes they represent."). We re-wrote the paragraph more clearly as follows: "This section includes a brief description of the main hydrologic processes represented in GSFLOW, with select parameters listed in Table 1. Full details can be found in the GSFLOW manual (Markstrom et al., 2008). In particular,*

*Table 1 from Markstrom et al. (2008) summarizes all the surface-water processes captured by PRMS modules, groundwater processes captured by MODFLOW stress packages, and model coupling procedures captured by GSFLOW." (p. 5 Lines 2-5).*

**p22. lines 7+. These don't strike me as conclusions and read a bit like an advertisement. To my central point, what is the scientific motivation and conclusions reached by this work. Reworking this paragraph would help that substantially.**

*After reading this review, we agree that the Conclusion should be re-written and have now done so. The new Conclusions section emphasizes the technical advances provided by the GRASS GIS domain builder tools, the capabilities of GSFLOW-GRASS across diverse settings demonstrated by the model examples, and the value of this new toolkit for making integrated hydrologic modeling more accessible; we also end with a list of potential future extensions of this toolbox.*

---

## Referee Report (RR1)

**General comments**

The authors have addressed all the concerns I had in my previous review, and have made substantial changes to the manuscript and added new capabilities to the model. The capability of taking spatially-distributed could be especially important for users to create model domains that represent the spatial heterogeneities in soil properties, landcover properties, and climate.

I apologize that I mistook "for-fee" ad "for-free" in my comment on Earthfx. I still feel that a comparison with Earthfx could be very helpful. However, this is just a minor issue.

---

## Author Response (AR2)

**RESPONSE TO REVIEWER 1:**
=========================

We would like to thank the reviewer for their time to review our paper. We are glad that the reviewer found several of our recent revisions to benefit the overall quality of the manuscript, and we have carefully gone over the remaining issues that the reviewer raised. In this response, we have addressed all the reviewer's comments, by editing the manuscript for many of the points, and by providing clarifications for others. We believe that these changes and explanations will now clearly present the technical advancements and value of our new toolkit.

The reviewer's comments are copied here with a gray background. Our responses are made with a white background; quotes from the text are in italics and changes are highlighted with underlines.

I would like to thank the authors for their effort in improving the manuscript according to the reviewers' comments. I see clear changes in several parts of the manuscript, most notably in the introduction part and in the description (including also a new feature) of the proposed toolkit. All of these changes are beneficial for the overall quality of the work. However, I must also admit that one of my major remarks concerning the Results section still persists in this revised version. Here my opinion is that the effort made by the authors did not really go to the 'right' direction. T**he discussion still contains many vague statements and it is not supported by illustrative figures. That is, the outcome of the results section appears disconnected from those statements (made in the Introduction) about the novelty of the toolkit.** This is, in my opinion, a major issue for a GMD paper where the reader expects to see a more in-depth discussion on the technical advancement of the proposed toolkit. I provide below some specific comments that I hope will be useful to address my major concern with the work.

Above, we boldfaced the reviewer's major general point to address. Our paper does not have sections named "Discussion" and "Results," so we assume that the reviewer was referring primarily to the "Examples" and "Conclusions" sections.

We thoroughly reviewed these sections and believe that they do support the statements in the Introduction, but the reviewer's comments make us realize that the text needed clarification in a number of places. Thus, while the reviewer recommends major revisions, we find that we can in fact address the issues through relatively straightforward edits to the text. Our revision does include the major change of adding Table 2, which summarizes the key characteristics of the diverse test catchments, including the challenging feature(s) that the GSFLOW-GRASS toolkit successfully handles to automatically produce simulations. We thank Reviewer 1 for their comments that inspired us to create this table, which we think greatly improves the presentation of our work.

We presume that the specific connections between the Introduction and the Examples and Conclusions sections that were of greatest concern to the reviewer are listed in the reviewer's

numbered comments below.  However, to ensure that we have fully addressed the entirety of the reviewer's general point, we systematically list, in order, each Introduction statement about the technical advancements of our toolkit, and explain how they are supported by rest of the manuscript, including new text edits to enhance the connections where appropriate.  For those points that we felt were already clearly demonstrated clearly in the original text, we did not to make further changes in the interest of remaining concise.

Introduction, Paragraph 4:
    We wrote: *"Our overarching goal is to develop a bundled package -- "GSFLOW-GRASS" -- to handle the complexity of the coupled GSFLOW model, thus tackling the grand challenge of accessibility plaguing many integrated modeling systems."*
    Our new toolkit does this - it can build an integrated hydrologic model run in a matter of minutes. The results are demonstrated by the Examples section.  No change made.

Introduction, Paragraph 4:
    We wrote that our toolkit is: *"fully automated"*
    The code is indeed fully automated and based on a single DEM input and a single "*.ini" file, as is depicted in Figure 3 and demonstrated through the applications in the Examples section.  No change made.

Introduction, Paragraph 4:
    We wrote that our toolkit is: *"robust"*
    Here we acknowledge that we should (1) be more careful with our use of the term "robust" (noting the reviewer's comment #1 below), and (2) summarize the various features of the example watersheds to demonstrate our point more readily.  By "robust," we mean that the toolkit does not fail, even for a diverse range of implementations.  To demonstrate this, the Examples section shows that GSFLOW-GRASS successfully handles three watersheds that each presents very distinct types of computational challenges to implementing GSFLOW.  To more clearly depict at a glance that GSFLOW-GRASS can work for diverse settings, we now summarize the different features of each example watershed in the new Table 2.  Also, to avoid confusion with other senses of "robust," we revised the text to not use this word in reference to model simulation results, where the reader can mistakenly interpret it to mean a lack of sensitivity to model inputs (reviewer's comment #1); however, we retain the term elsewhere when it is used in reference to the toolkit, where we feel that its meaning of "not failing" is unambiguous.

Introduction, Paragraph 4:
    We wrote: *"Python scripts generate model input files and model output graphics, and extensions using the open-source GRASS GIS platform build topographically defined sub-watersheds linked to subsurface grid cells."*
    The overall workflow described here is schematically depicted in Figure 1, and the output graphics are demonstrated in Figures 5-7. The Methods section defines in detail the GRASS GIS connection. No change made.

Introduction, Paragraph 4:

We wrote: *"Our use of only free and open-source programming languages and software is a key feature of the toolbox's accessibility... Open-source software facilitates implementation of GSFLOW-GRASS by diverse academic, government, and individual entities, enables further community development of GSFLOW-GRASS, and aligns with the USGS's goal to make its resources publicly accessible."*

The accessibility of the software used for the toolkit is discussed in the code development sections (including Section 2.2 GRASS GIS), highlighted in Shullcas example case, and re-emphasized in the Conclusions.  The open-source aspect of our toolkit is verifiable with our GitHub repository, which is referenced in the manuscript.  No change made.

Introduction, Paragraph 5:

We wrote: *"Whereas overland flow routing and the calculation of drainage basins from topography are standard GIS capabilities, our tool improves upon these by automatically building topologically structured vectorized drainage networks without manual corrections using a least-cost path approach (Metz et al., 2011), while also including information on adjacency and routing pathways through the network that is required by integrated hydrologic models."*

The Methods section shows that this is accomplished using r.watershed (Metz et al, 2011's tool) and our newly developed r.stream.network.  No change made.

Introduction, Paragraph 5:

We wrote: *"The main technical advancement of GSFLOW-GRASS is the development of streamlined GRASS GIS extensions that have passed a diverse range of stress tests, including steep to low-relief topographies, large and intricate to small and simple drainage systems, incomplete to full topographic drainage integration, and mountainous to coastal watersheds."*

This is exactly the purpose of the Examples section. To make this clear, we have added Table 2 to summarize the different test cases and their distinct features.

Introduction, Paragraph 5:

We wrote: *"These new computational capabilities enable rapid, automated delineation of surface-water drainage networks linked to subsurface domains across any generalized landscape and practical resolution."*

In the Examples section, the implementations cover two orders of magnitude in drainage basin area, one order of magnitude in landscape relief, a factor of 30 in MODFLOW grid cell size, and a factor of 20 in HRU area. They include all of the possible complications now tabulated in Table 2. We feel that this is a broad-ranging and complete test of the capabilities of GSFLOW-GRASS, thus supporting our statement in the Introduction.  To make sure that this broad range is not lost among the details in the Examples section, we added Table 2 to show at a glance how the each example demonstrates different capabilities of GSFLOW-GRASS.  We also slightly edited the above sentence in the Introduction to be more exact that the range of capabilities we test with the toolkit was set by the range of capabilities of the GSFLOW model.  The revised sentence is: *"These new capabilities enable rapid, automated delineation of surface-water drainage networks linked to subsurface domains across any generalized landscape and computationally feasible resolution -- within the range of scales that can be*

Introduction, Paragraph 5:

We wrote: *“By doing this all within a framework that also includes open-source model input and post-processing tools, GSFLOW-GRASS presents a solution toward more accessible integrated hydrologic modeling.”*

As noted in the Introduction, the only roughly comparable software that exists for GSFLOW is commercial (Earthfx). The USGS is working on a similar product (Gardner et al., in review), but one that includes more manual work and has a different gridding structure.  Thus, this toolkit does indeed fill a gap.  No change made.

Finally, in response to the reviewer's concern about whether these statements are "supported by illustrative figures": we believe that the new Table 2 and text clarifications now highlight the distinct challenges presented with each example watershed, such that the successfully executed simulation results displayed in Figures 5-7 demonstrate the technical contribution of GSFLOW-GRASS of handling coupled hydrologic model implementations across a diverse range of conditions.

In summary, we hope this exhaustive list of points, along with new revisions to clarify the text, shows that the body of our manuscript does support the introductory statements about the technical advances of the new toolkit.  We do greatly appreciate the reviewer raising their concern, because it prompted us to improve various parts of the manuscript and to create Table 2, which showcases the diverse range of challenges that we demonstrate the toolkit to successfully handle.

1. Authors state that the scope of the three examples is to demonstrate the robustness of the developed GRASS GIS domain builder across diverse topographic settings. I have problems in assessing this 'robustness' from Figures 5-6-7. In my opinion robust means that the output of the model does not change 'much' with small perturbations of the input. According to this I would frame the exercise in a different way. For each test case you identify a major challenge and then you exploit the toolkit to automatically generate a suite of model grid configurations that are tested in terms of different model output. This exercise will provide the opportunity to demonstrate the validity of the approach in guiding users during preliminary (i.e., grid definition) still critical modeling steps.

We believe the reviewer is referring to the following original text (and following passages):
*“Three implementations of GSFLOW-GRASS serve the dual purposes of demonstrating (1) the robustness of the newly developed GRASS GIS domain builder across diverse topographic settings, including those prone to problems with standard GIS stream network tools, and (2) the variety of hydrological processes that can be assessed with the use of GSFLOW-GRASS. The implementations utilize topographic and climatic inputs from widely differing sites to show the range of applicability of the new toolbox.”*

We thank the reviewer for bringing to our attention the ambiguity of the term "robust."  We agree that when referring to model runs, "robust" can have a similar meaning as "stable," where small perturbations do not cause much change in results.  We instead intended to say that our toolkit (not model runs) is robust, in that it is difficult to "break" -- it can be successfully applied (without failing) to a diverse range of watersheds, including those that might pose challenges for other drainage network building workflows.  The reviewer refers to this second sense of "robust" in their suggestion that we identify major challenges for each test case and then exploit the toolkit to handle them.  This was exactly our intention with the Examples section, but based on the reviewer's comment, we could have conveyed this more clearly.  This lack of clarity was likely because our original opening sentences to the Examples section (cited above) made only generic reference to differences among the sites, rather than highlighting them as having challenges to be tackled by the toolkit. To address this, we revised the opening to (1) not use the misleading word "robust" and (2) to follow the reviewer's suggestion to first explicitly introduce the examples as representing major challenges to be handled by the toolkit:

> *"Three example implementations demonstrate (1) the variety of hydrological processes and environments that can be explored using GSFLOW-GRASS and (2) how the toolkit's GIS domain builder can handle diverse topographic settings, including those prone to problems with standard GIS stream network tools.  For this second point, each simulation presents a unique set of technical challenges in developing a topographically based model domain that can properly route rainfall through a network of stream segments and sub-basins as well as a connected groundwater-flow grid." (p. 18 line 30-p. 19 line 1)*

The specific challenges were already described in the original text under the test case sections. Each test case section is organized to first introduce the watershed (where is it and why apply GSFLOW for it), **second explain the challenge it poses**, third provide details of the GSFLOW-GRASS implementation, and fourth describe the simulation results.  For the second component, the Shullcas examples opens with "*A major obstacle with the steep topography and narrow canyons of Shullcas is the need for impractically high resolution and computational expense if using a standard gridded model domain*" (p. 21 lines 1-2).  The Santa Rosa example states: "*Water Canyon is unique among the three example sites in that its outflow drains to the ocean, which is represented by cells with NULL values. This required modifications to the source code in order to properly accumulate flow across the watershed without encountering errors due to the null values offshore*" (p. 23 lines 10-12).  And the Cannon River example explains: "*Much of this watershed's post-glacial topography is characterized by small local hills and enclosed basins that have not yet been organized (or integrated) by fluvial erosion into a linked valley network, in which water would flow directly to a stream without encountering an enclosed lake or other basin.  In such settings lacking integrated drainages, simple downslope flow-routing algorithms typically fail, and "pit filling" can produce spurious results by inappropriately modifying the real topography*" (p. 24 lines 2-6).

Although we believe the original text had already described the major challenge of each example, the reviewer's comment made us realize the distinct challenges were perhaps buried within the text.  To address this, we have newly created Table 2 to present the key watershed

characteristics and the challenges that each example serves to test.  We anticipate that this Table 2 will clarify any confusion about the purpose of the different examples.

2. In the first example it seems that one of the most important features of the toolkit is the combination of flow paths extracted using high-resolution topography with the 'fewest' possible computational cells for MODFLOW. Could the authors identify (using their pre- and post-processing tool) the 'exact' value of the fewest possible computational cells?

We believe the reviewer is referring to the following original text: *"GSFLOW-GRASS's use of topographically based irregular HRUs makes it possible to compute flow paths using high-resolution topography but with the fewest possible computational cells corresponding to fundamental surface-water hydrologic units – stream segments and sub-basins. These irregular drainage units are then linked to a relatively coarse regular groundwater (MODFLOW) grid whose cell elevations are assigned based on our hydrologically corrected downscaling method (Section 3.2)."*

We would like to first point out that our mention of "computational cells" referred to the HRUs and stream segments, and not to the MODFLOW grid cells.  We modified the second part of this passage to make the distinction clearer.  Substantively, this comment did make us realize that the original text inappropriately stressed an absolute "fewest" number of cells, while the main idea we wanted to convey is that high-resolution topography may be used to define a relatively computationally inexpensive surface-water grid. Thus, it is not meaningful to calculate a "fewest" number of cells as the reviewer requests, and we have corrected our text by removing the absolute term of "fewest."   The revised passage is:

> *"GSFLOW-GRASS's use of topographically based irregular HRUs makes it possible to compute flow paths using high-resolution topography, and then convert these into a relatively small number of computational cells corresponding to fundamental surface-water hydrologic units -- stream segments and sub-basins. In contrast, a surface discretization not automatically aligned with flow direction would require a greater number of cells to accurately resolve flow paths. The irregular drainage units generated by GSFLOW-GRASS are then linked to a groundwater (MODFLOW) grid, whose regular finite-difference grid can also be coarse relative to the high-resolution DEM. Cell elevations for the MODFLOW grid are assigned based on our hydrologically corrected downscaling method (Section 3.2)." (p. 20 line 18-p. 21 line 6)*

3. In the second example, it seems that one of the most challenging task for the toolkit is to appropriately handle NULL value cells. I am not sure about it.

We believe the reviewer is referring to the following original text: *"Water Canyon is unique among the three example sites in that its outflow drains to the ocean, requiring that GSFLOW-GRASS appropriately handles NULL values for ocean grid cells."*

We thank the reviewer for pointing out that we were not clear here. We have now revised the

text to more clearly emphasize the toolkit's important feature of properly handling NULL values:

> *"Water Canyon is unique among the three example sites in that its outflow drains to the ocean, which is represented by cells with NULL values. This required modifications to the source code in order to properly accumulate flow across the watershed without encountering errors due to the null values offshore." (p. 23 lines 7-9)*

The relevant code modifications for GSFLOW-GRASS to function with NULL-valued ocean cells may be found at:
https://github.com/UMN-Hydro/GSFLOW-GRASS/commit/9e1f14f54d797e4e634d1b6b27f59f9c1b3cbc6c

4. Again in the second example, I am really confused by those statements connecting the numerical convergence of MODFLOW with the merit of the proposed toolkit. Specifically, I do not see how the suite of tools can avoid numerical convergence issues when MODFLOW cells run dry. Overall, I think that numerical convergence problems may still appear in MODFLOW and this won't make the proposed tool useless.

We believe the reviewer is referring to the following original text: *"The semi-arid climate can lead to losing streams that may run dry (Jazwa et al., 2016); the GSFLOW-GRASS implementation correspondingly shows low streamflow in simulations (Figure 6B) but avoids numerical convergence problems that can arise when MODFLOW cells run dry."*

The reviewer's comment made us realize this sentence was poorly worded. Indeed, when a MODFLOW cells runs dry, using the GSFLOW-GRASS toolkit won't magically prevent numerical convergence issues. We meant that using GSFLOW-GRASS could help prevent the case of cells running dry -- and consequently help to avoid the corresponding numerical convergence issues. We have revised this sentence as follows:

> *"The semi-arid climate can lead to losing streams that may run dry (Jazwa et al., 2016), which may make GSFLOW simulations susceptible to having entire MODFLOW cells become dry and cause numerical convergence issues. The GSFLOW-GRASS implementation does simulate low streamflow as expected (Figure 6B), but the domain-builder generated sufficiently thick and hydrologically corrected MODFLOW cells that maintain water and avoid any computational problems." (p. 23 lines 16-20)*

5. In the third example, authors state that under smooth topographic gradients simple downslope flow-routing algorithms fail and they use this as an example to demonstrate the ability of their toolkit under such low-relief conditions. However, it is not clear to which extent this ability is due to a new feature of the proposed toolkit or to an already existing GRASS (r.watershed) functionality.

We believe the reviewer is referring to the following original text: *"The Cannon River watershed covers 3720 km$^2$ , with only 215 m of total relief, making it the largest and lowest-gradient of*

*these study cases, leading to very different computational challenges than those in the steep watersheds. Its deglacial topography has not yet been organized by fluvial erosion into a linked valley network. In such settings, simple downslope flow-routing algorithms typically fail, and "pit filling" can produce spurious results by inappropriately modifying the real topography. GSFLOW-GRASS routes surface-water flow using the GRASS GIS r.watershed least-cost path algorithm, which is designed to produce drainage networks that route flow in the long-range path of steepest descent regardless of the degree of local drainage integration. Its success in this example demonstrates the ability of GSFLOW-GRASS to automatically create a topologically correct and linked drainage network in low-relief settings for hydrologic model simulations."*

Smooth topographic gradients make flow-routing problems easy to solve, and therefore we did NOT state that they would fail under these conditions. We believe the reviewer may have confused "low-relief" with "smooth topographic gradients," which are not the same.  The problem presented with the Cannon River watershed is that formerly glaciated settings require time to develop a fully integrated drainage network that is expressed in the topography, and before this point, they are full of local depressions and pits. Most GIS tools will artificially fill these pits, but r.watershed's least-cost-path algorithm is able to route flow across them in a more realistic way. This is therefore an existing GRASS GIS functionality, and our contribution has been to link it to the GSFLOW domain builder.  To better explain that the issue is related to drainage integration, as well as our contribution with r.watershed, we have revised this paragraph:

*"The Cannon River watershed covers 3723 km$^2$ with only 210 m of total relief (Table 2), including a significant amount of non-integrated drainage that leads to very different computational challenges than those in the steep watersheds. Much of this watershed's post-glacial topography is characterized by small local hills and enclosed basins that have not yet been organized (or integrated) by fluvial erosion into a linked valley network, in which water would flow directly to a stream without encountering an enclosed lake or other basin.  In such settings lacking integrated drainages, simple downslope flow-routing algorithms typically fail, and "pit filling"' can produce spurious results by inappropriately modifying the real topography. GSFLOW-GRASS routes surface-water flow using the GRASS GIS's **r.watershed** least-cost path algorithm, which is designed to produce drainage networks that route flow in the long-range path of steepest descent regardless of the degree of local drainage integration. By using **r.watershed**, GSFLOW-GRASS is able to automatically create a topologically correct and linked drainage network in settings that lack integrated drainage for hydrologic model simulations." (p. 23 line 31-p. 24 lines 6)*

6. Finally, all applications presented in this work are at relatively small catchment scale. Can the authors proof the utility of their tool at large, i.e. continental, scale using for instance the CONUS domain?

GSFLOW is a watershed model developed for use at scales of a few square kilometers to a several thousand square kilometers.  Thus, GSFLOW-GRASS should not be applied at the

continental scale.  We realize that this would not be obvious for readers not closely familiar with GSFLOW, and so we added the following sentence to the Background section:

> "It [GSFLOW] is designed for simulations of watersheds with areas of a few square kilometers to several thousand square kilometers (p. 2, Markstrom et al., 2008)." (p. 3 lines 24-25)

With our test cases, we do in fact span nearly this full range of scales applicable for GSFLOW, and we added a sentence to the "Examples" section to highlight this:

> "Their catchment areas range from 12.5 $km^2$ to 3723.0 $km^2$, covering the range of scales that GSFLOW was developed to simulate." (p. 19 lines 10-11)

According to the aforementioned points I would encourage authors to make a substantial effort to augment the insights from the results section, which will bring the overall quality of the manuscript suitable for GMD standard.

We greatly appreciate the reviewer highlighting unclear parts of our manuscript, and we can see how the corresponding interpretations by the reviewer would seem to require major revision.  We believe that our edits to clarify the text and additional Table 2 fully address the reviewer's concerns, and that additional substantive changes are not necessary (e.g., no need to reorganize examples around major challenges - this was already done and just required clarification and a summary Table; no need to calculate absolute "fewest" number of cells - just needed to correct the text to refer to "relatively few;" no need to implement GSFLOW-GRASS on a continental basin scale - just needed to explain that the GSFLOW manual says it should only be applied to much smaller scales).

Other points:

1. The discussion in the Introduction connecting surface-subsurface coupling approaches to model input parameters is misleading. You may have that a fully coupled and a sequentially coupled approach requires the same amount of input information.

We believe the reviewer is referring to the following original text: *"GSFLOW is not "fully integrated" in the sense that it does not simultaneously solve surface and subsurface flow equations; instead it consists of an iterative coupling between MODFLOW and PRMS that requires nearly all the individual input files for each of the two original models as well as an additional GSFLOW-specific linkage file.  This means that to run GSFLOW, the user bears the burden of first generating a multitude of formatted ASCII files while ensuring that the three sets of input files are consistent with each other and can produce convergent coupled simulations."*

The reviewer is correct that fully-integrated/coupled and iteratively/sequentially-coupled approaches can require the same amount of input information.  We meant that GSFLOW is iteratively coupled in a way that requires a complicated mess of heterogeneous input files, and

this is what makes the model burdensome to use.  We have revised the text to clarify this:
> *"GSFLOW is not "fully integrated" in the sense that it does not simultaneously solve surface and subsurface flow equations; instead it consists of an iterative coupling between MODFLOW and PRMS that requires nearly all the individual input files for each of the two original models as well as an additional GSFLOW-specific linkage file.  While a fully integrated model may have all the input information streamlined into a small number of internally consistent and efficiently organized files, to run GSFLOW, the user bears the burden of generating a multitude of diversely formatted ASCII files and ensuring that they contain inputs that correctly correspond with each other and can produce convergent coupled simulations." (p. 2 lines 18-23)*

2. I would avoid statements about the potential of the tool in developing countries where computational resources may be limited. I do not see how the proposed tool may help with respect to this limitation.

We believe the reviewer is referring to the following original text: *"Our choice of an example in the Peruvian Andes demonstrates how our entirely open-source modeling system may be applied to problems in the developing world, where computational resources may be limited for local environmental researchers and practitioners."*

We realize that we were unclear: the reviewer probably understood "computational resources" as referring to processing power.  We meant it in the sense of monetary resources for software, which our toolkit does address through its use of all open-source and freely available software. We have revised the text as follows:
> *"Our choice of an example in the Peruvian Andes demonstrates how our entirely open-source modeling system may be applied to problems in the developing world, where financial limitations faced by local environmental researchers and practitioners make it difficult to use commercial software solutions." (p, 19 lines 14-16)*

3. Usually captions do not contain discussion of the presented results. Please check Figures 5-6-7.

Thank you for pointing this out. Some of our co-authors are accustomed to sub-disciplines in which extended captions are more common (e.g. geology), but we see that they are not typical in GMD. We have moved any significant discussion from the captions to the main text, while retaining some limited information in the captions that we believe will help the reader more easily understand the figures.

**RESPONSE TO REVIEWER 2:**
========================

We greatly appreciate Reviewer 2's recommendation for our manuscript to be accepted as-is. We have not made any major substantive changes to the manuscript and thus believe that it largely remains in the form approved by the reviewer. The edits that we have made in response to Reviewer 1 should only improve the presentation.

[revised manuscript text omitted]

---

## Author Response (AR3)

*All of the referee's review comments are copied below and shaded in gray. Our responses are in italics. New text is quoted and/or referenced in boldface.*

I would like to thank the authors for their effort in clarifying several points made in my last round of review comments. In acknowledging such effort, I must recognize that my major concerns, expressed during the first and second iteration of this review process, still persist. These points cannot be addressed just reformulating some sentences in the text but they need, in my opinion, additional substantial work by the authors. Indeed, as previously stated, the Results section fails to offer a clear view of the technical advancements proposed with the GSFLOW-GRASS toolkit. This is, in my opinion, a major issue for a GMD paper. Here I must also highlight that despite this central concern, Figures 5-6-7 (and their information content) have remained nearly unchanged through the review process. I will try to re-iterate on these major issues starting from the newly included table (i.e., Table 2 of the revised manuscript) that summarizes the tests of GSFLOW-GRASS capabilities/technical challenges for the three selected test cases.

*We greatly appreciate the referee's continued attention and time to ensure that our manuscript is of suitable quality for GMD, and we are glad that the referee recognized improvements in our last revision. We are of course disappointed that the referee still finds major concerns, as we did seriously consider all previous comments. In this revision, we focus on three points that we anticipate should satisfactorily address the latest comments.*

*(1)*
*First and foremost, we address the referee's main concern about our toolbox's technical merits by substantiating with specific details the statements flagged in our 3 examples. Despite our previous efforts to clarify technical advancements, the referee's persistent comments made us realize that indeed, a number of our claims remained vague and would greatly benefit from direct support. We added precise comparison numbers, detailed literature examples, and improved explanations in the three example in Section 4, as detailed in our specific responses below.*

*Further, we substantially edited Section 3.2 ("GRASS GIS domain builder") to more clearly highlight the overall technical contributions. We had previously left some technical details vague in the examples section because we had explained the general points in Section 3.2 (GRASS-GIS Domain Builder). However, we now see that we had not adequately highlighted the innovations in Section 3.2, and further, the section was not well-organized or referenced in the examples for the reader to easily tie them together. We now provide new sub-section headings for easier references to Section 3.2. **We have almost completely rewritten the part on the surface-water network (now Section 3.2.1)** to clearly detail the computational advantages of GRASS-GIS's r.watershed extension, supported by a complete literature review; then, to address the reviewer's concern that using an existing software is an insufficient technical advancement (see below), we again provide a complete literature review to explain that ours is the first effort to incorporate this algorithm into an integrated hydrologic model toolbox, thus newly harnessing its power in this field. Other changes to Section 3.2.1 serve to more clearly explain the different features of the surface-water network builder. **The entire section on the Groundwater-flow grid (now Section 3.2.2) was also significantly edited**, in large part to better explain the hydrologic correction of coarsened MODFLOW grids - an aspect in the examples that the referee was unable to follow in the previous version (see below). **Finally, we also made major edits to the section on support for additional GSFLOW***

*functionality (now Section 3.2.3).  We added the new extension "v.gsflow.mapdata" in our previous revision in response to reviewer comments about the limited input data support provided by GSFLOW-GRASS compared to some other integrated hydrologic modeling pre-processing packages.  This revision includes more details on the types of customized inputs that users can make with this new extension.*

*Please see Section 3.2 (p. 10) - in particular, 3.2.1 (p. 11), 3.2.2 (p. 13), and 3.2.3 (p. 13). Those sections are almost entirely re-written, but we do not attempt to copy them here because of their length.*

*Finally, related to the reviewer's concern about Figures 5-7, we would like to point out that we did in fact modify Figure 7 in our previous revision to include a comparison with observed streamflow in Cannon River.  This was done to build confidence that our toolbox produces realistic preliminary simulations that can be further calibrated for specific study.*

*(2)*
*We acknowledge that our toolbox builds upon mostly existing algorithms and often straightforward programming rather than formulating brand-new methodologies, but we would like to argue that the primary technical innovation is to bring together these different components in an automated, accessible, and self-consistent package for integrated hydrologic modeling - a task that is not at all trivial and has not been previously done (specifically, using the powerful least-cost path flow-routing algorithm with integrated hydrologic modeling).  As we point out in the manuscript, providing a practical utility that enables more researchers and practitioners to implement integrated hydrologic models can help accelerate advances in hydrologic science understanding and water resource management - we anticipate this to be our major contribution.  We do appreciate that the referee wishes to uphold a high technical standard for GMD and we emphasize that we do NOT wish to water-down the technical quality of GMD articles, but we also believe that technical advancement can come in the form of innovatively integrating disparate components into a new practical and useful tool.*

*To check whether our perspective is consistent with the stated goals and requirements of GMD, we carefully reviewed the GMD editorial document (Hargreaves et al. 2015 - full reference is below) for guidance.  We appreciate that the referee acknowledged in an earlier review that our manuscript does fit within GMD's call; we would like to further share the following excerpts.*
*- Support for contributions that are accessible and practical utilities:*
> *"The papers should be detailed, complete, rigorous, and accessible to a wide community of geoscientists. In addition to complete models, this type of paper may also describe model components and modules, as well as frameworks and utility tools used to build practical modelling systems, such as coupling frameworks or other software toolboxes with a geoscientific application."*
*- Support that the scientific goal need not be a major scientific/technical discovery:*
> *"The scientific goal is reproducibility: ideally, the description should be sufficiently detailed to in principle allow for the re-implementation of the model by others, so all technical details which could substantially affect the numerical output should be described."*
> *"It is not expected from a GMD paper that it contain novel scientific discoveries."*
*We acknowledge that "scientific goal / discoveries" do not strictly equate with "technical discovery," but in this editorial document, those were the closest terms we could find related to*

*"technical advancement"; in fact, the word "technical" appears only one time (cited above) in the entire section on "Model description papers" (Section A2) - the category of our manuscript. No where in the editorial is there a statement about a required level of technical innovation. As such, we do believe that after clarifying the technical aspects of our toolbox (the referee rightfully pointed out that our explanations needed to be improved), the degree of technical discovery should not be a metric used against our submission to GMD. Our toolbox already has proven accessibility and practical application based on GitHub statistics; outside our immediate collaborators, we have had:*

- *344 visitors since April 22, 2018, when we started to track this (~3/day)*
- *2 forks (dynamic copies for further code development by third parties)*
- *4 stars (bookmarks)*
- *1 external scientist actively using and building upon GSFLOW-GRASS after finding it online (his fork is one of the two)*

*Reference*

*J. C. Hargreaves, A. Kerkweg, R. Marsh, A. Ridgwell, D. M. Roche, and R. Sander, "Editorial: The publication of geoscientific model developments v1.1," Geosci. Model Dev., vol. 8, no. 10, pp. 3487–3495, Oct. 2015.*

*(3)*

*We would like to clarify that only one purpose of the examples is to demonstrate technical aspects; they also serve to demonstrate applications with GSFLOW that can be facilitated with our toolbox. We feel that in our effort to clarify the technical points in response to the referee's previous comments, this other important purpose is becoming buried. Again, we share the following excerpts from Hargreaves et al. (2015) to support the appropriateness of using examples for this reason, and not solely for demonstrating technical advancements:*

> *"The model description should be contextualised appropriately. For example, the inclusion of discussion of the scope of applicability and limitations of the approach adopted is expected."*
>
> *"Examples of model output should be provided, with evaluation against standard benchmarks, observations, and/or other model output included as appropriate."*

*(Note that as mentioned above, we added streamflow observations to Figure 7 in the previous revision to show that out-of-the-box, results and plots from GSFLOW-GRASS can show realistic simulations as well as provide insight into how certain model parameters need to be further calibrated.) Some of the referee's comments made us realize that we were not always clear that some aspects of the examples served to highlight possible applications, and not necessarily technical advancements, and we have edited the text accordingly. **We have also added the column "Applications" in Table 3 to summarize these applications for each example**. Specific details are provided below.*

The technical challenges identified for the first test case (i.e., Shullcas River, Peru) are the (i) steep topography and (ii) seasonal rainfall.

(i)        As concerns the first technical challenge authors argue that a steep topography requires the definition of the surface flow path using high-resolution topography with the option of resolving the subsurface at relatively coarse resolution. Therefore, the technical challenge consists in handling a surface discretization not automatically aligned with subsurface flow path. This technical challenge reflects the sketch of Figure 1d, if I am right. Therefore, I think that a

good way to test the toolkit would be to generate different subsurface configurations (in terms of grid resolution) with the same number of HRUs obtained with the high-resolution DEM (30 m for this test case). Here you could show the great potential of the automated workflow in generating consistent modeling scenarios that potentially subtend different level of information. Some concrete numbers on the computational gain would substantiate some too-vague statements still present in the manuscript ("…steep topography and narrow canyons of Shullcas is the need for impractically high resolution and computational expense if using a standard gridded model domain…")

*We really appreciate the referee's comment, because it made clear that we inadequately explained the challenges of Shullcas and GSFLOW-GRASS' solutions for them. The steep topography and narrow canyons lead to a three different but related challenges, only one of which is the one that the referee was able to readily pick out (misaligned grids) based on how the previous version was written. The full list of challenges include: (1) need for a high-resolution surface flow representation that is computationally efficient, (2) need to link misaligned grids, and (3) need to coarsen MODFLOW regular grid without causing hydrologically incorrect flow. We have largely rewritten the Shullcas section to clarify this.*

*We also now see that our statements about how GSFLOW-GRASS addresses challenges were indeed vague. The referee had an excellent suggestion to provide concrete metrics to support our claims. The referee mentioned testing different grid alignments, but actually this was the most straightforward of the challenges and was solved with code that handles painstaking indexing between the domains, so there is not much to test. Instead, we decided it would be more meaningful to substantiate the other two solutions. To demonstrate the computational efficiency from the irregular grid, we now report that only 79 HRU cells were needed to capture nearly all of the essential flow-routing information from the original high-resolution $2 \times 10^5$ cells.* **We added a column with flow-routing grid cell size to Table 2 to facilitate the comparison**. *To explain the value of hydrologically corrected MODFLOW cells in steep canyons, we now explain that without it, simulations of the Shullcas watershed showed water converging next to instead of within the stream channel network.*

*The third paragraph of the Shullcas section was almost entirely rewritten to incorporate the above changes:*

> ***(p. 23 line 11-30)***
> ***"The steep topography makes the Shullcas River Watershed an apt testbed for examining the ability of GSFLOW-GRASS to represent surface-water-groundwater links in challenging terrain. The major obstacle with Shullcas' steep topography and narrow canyons is the need for high resolution to represent surface flow, which leads to an impractically high number of computational units and expense if using a standard gridded model domain. An irregular surface grid can provide a much more efficient discretization, but this then entails painstaking indexing to link it to the subsurface grid, which must be a regular rectangular grid for the MODFLOW component of GSFLOW. Also, as the most computationally expensive part of GSFLOW, practical MODFLOW implementation typically needs a coarser resolution grid than that used for resolving the stream network, but simple coarsening of DEMs with steep gradients (e.g., by using mean elevations from the higher-resolution DEM) can result in hydrologically incorrect groundwater flow directions. GSFLOW--GRASS addresses these problems by computing flow paths***

*using high-resolution topography (~2x10^5 grid cells of 900-m^2 size) and converting these into a far smaller number of larger computational surface cells (79 HRUs that are >=1 km^2 in area) that convey the same fundamental surface-flow information (see Table 2); this efficiency is possible because the surface is discretized along topographically defined surface-water hydrologic units -- stream segments and sub-basins (Section 3.2.1). To create a coarsened subsurface rectangular grid domain for the MODFLOW component, GSFLOW-GRASS' hydrological correction to enforce integrated subsurface drainage (Section 3.2.2) proved essential for preventing unrealistic results. Early model tests for Shullcas showed that simple grid coarsening using the mean value of the elevations from the higher-resolution grid could, for example, average elevations between flat valley floors and steep canyon walls. This caused cells containing the stream to be higher in elevations than the surrounding surface on the groundwater flow grid, leading to lateral flow out of these "dams" that formed as a numerical artifact of averaging. As the final step of the domain-building solution, GSFLOW-GRASS extensions seamlessly link the hydrologically corrected coarse-scale MODFLOW domain with the irregular surface HRUs (Section 3.2.3)."*

(ii)    The aim of this work is not to present any scientific validation of the results obtained by implementing the GSFLOW-GRASS workflow, as also stated by authors. Therefore, I do not see any technical challenge implied in the term "Seasonal rainfall" included by authors in this table.

*This referee is correct that "seasonal rainfall" was inappropriately identified as a technical challenge. We realized that we were conflating the two different purposes of the examples: (1) demonstrating technical advances and capabilities of the toolbox and (2) demonstrating hydrological processes and applications possible with GSFLOW-GRASS. The point of "seasonal rainfall" should fall in the 2nd category - it is an example of a use case of GSFLOW-GRASS.* **We have now modified Table 2 so that it is split into two separate tables.** *Table 2 contains only the numerical data for the watersheds, while Table 3 contains the "GSFLOW-GRASS Capabilities" and "Applications" demonstrated with each watershed example.* **"Seasonal rainfall" is no longer placed under "GSFLOW-GRASS Capabilities" for Shullcas and is instead listed under "Applications" in Table 3.**

The technical challenges identified for the second test case (i.e., Water Canyon, USA) are the (i) NULL cells defining irregular coastline, (ii) small basin, and (iii) small number of coarse-resolution cells.

(i)    I must admit that I still do not see the challenge of handling NULL cells. Authors show the modification of the code but this remains for me a basic feature of any terrain analysis tool.

*The referee's comment made us realize that we did not adequately explain the problem with coastal boundaries. The referee is correct that assigning NULL values alone is not an issue; instead, the complication is in how to set boundary conditions for flow at a coastal pour-point where the NULL value is encountered. We rewrote and greatly expanded this section about the coastline to explain the need to carefully set constant head boundary conditions, and* **we have reworded the issue in Table 3 on "GSFLOW-GRASS capabilities" to refer to the irregular coastline boundary condition rather than to NULL values**. *The revised main text reads:*

*(p. 25 line 7-15)*
*"Water Canyon is unique among the three example sites in that its outflow drains to the ocean. It therefore requires GSFLOW-GRASS to accommodate irregular boundaries (coastlines) by properly assigning boundary conditions and routing flow through them. Users identify ocean pixels by assigning NULL values to them; this causes flow routing from r.watershed to stop at the shoreline. To allow flow out of pour-point at the mouth of the river, the immediately downgradient MODFLOW cell can be set as a constant-head boundary, but this cell must be chosen carefully. The finite-difference scheme in MODFLOW dictates that the constant head boundary condition must be supplied along one of the four cardinal directions of the pour-point. Therefore, if the river flows diagonally to the sea, its constant-head boundary must be moved to the closest non-diagonal cell. v.gsflow.grid finds the proper constant-head boundary cell to set for the coastal case, as well as for any inland drainage case in which the pour point also requires a downgradient constant-head boundary."*

(ii)      Why a small basin is a challenging case for GSFLOW-GRASS?
(iii)     What's the technical challenge associated to a small number of coarse-resolution cells? What's the meaning of small?

*Compared to the other two examples, Santa Rosa has both the coarsest resolution for determining the flow-routing and the smallest drainage area, which means that there is the sparsest amount of information to resolve how water is moving in the smallest watershed. That makes it the most difficult to accurately predict flow of the three examples, which is why we previously pointed out this aspect of Santa Rosa. However, we now realize this is more related to the choice of datasets and watersheds rather than the toolbox method, and so highlighting this "challenge" is a distraction to the reader - just as the referee found.* **We have removed any mention of the small drainage size and coarse flow-routing resolution from the Santa Rosa example** *and instead now focus on the important features that the example showcases relevant to the toolbox capabilities. These include the NULL values (details above), and the hydrologically corrected downscaling method (details below).*

*In addition to the technical capabilities demonstrated with Santa Rosa, we also expanded our discussion of the hydrological processes demonstrated with this example. As mentioned in our overall response, we wish to better emphasize that there is dual purpose to our examples, and showing technical capabilities is only one, and example applications is the other. We heavily edited the final paragraph to further explain the potential of using GSFLOW-GRASS to investigate and manage erosion through simulations of surface runoff, and we demonstrate how the toolbox's example hydraulic conductivity script was used to create heterogeneous conditions found with eroded watersheds:*
*(p. 26 line 6-12)*
*"The Santa Rosa example demonstrates an application in which GSFLOW--GRASS can be used to investigate and manage erosion associated with hydrological conditions. Erosion of upland areas moves sediment downslope to the areas flanking the stream channel, which contains coarser-grained alluvial sediments. We represented this heterogeneity using a spatially distributed hydraulic conductivity field (Figure 6D) generated with the example model input script*

*included in GSFLOW-GRASS (Section 3.3.3).  Figure 6E demonstrates how the post-processing tools can be used to evaluate surface runoff, a driver of erosion on the island (Schumann et al. 2016).  Simulations show precipitation events triggering surface runoff (Figure 6E), which could denude the hillslopes and transport eroded sediment through the drainage network (Figure 6B)."*

**We have filled out "GSFLOW-GRASS Capabilities" and "Applications" in Table 3 accordingly for the Santa Rosa example.**

According to point (i), (ii), and (iii) I have problems in identifying the technical capabilities of the proposed workflow from this second test case. Again, there are confusing statements in the discussion of the results that do not bring much insight:

"The GSFLOW-GRASS implementation does simulate low streamflow as expected (Figure 6B), but the domain-builder generated sufficiently thick and hydrologically corrected MODFLOW cells that they maintain water and avoid any computational problems."

What do you mean with "sufficiently thick" and "hydrologically corrected" MODFLOW cells?

*The referee's comment helped us realize the above sentence was poorly worded, and we needed to provide more detail, because the hydrologically corrected MODFLOW cells are among the key capabilities demonstrated with this example.  We essentially rewrote this paragraph to explain why the hydrologically corrected downscaling method for MODFLOW grid cells is important for preventing MODFLOW cells from becoming incorrectly dry and posing numerical problems; we also separated out the confusing statement about "sufficiently thick" MODFLOW cells and clarified that we are talking about MODFLOW layer depth:*

> *(p. 25 line 16-p. 26 line 5)*
> *"**Losing streams such as those in the steep and semi-arid Water Canyon catchment often run dry (Jazwa et al. 2016). If this causes MODFLOW cells to lose all of their water, GSFLOW will fail to numerically converge. Thus, the Water Canyon example also serves to demonstrate GSFLOW-GRASS' ability to prevent this problem by (1) incorporating MODFLOW-NWT, which uses a Newton-Raphson solver for increased stability (Niswonger et al. 2011); (2) allowing the user to specify an adequately deep MODFLOW discretization in the Settings file (Section 3.1) to supply sufficient water through the dry season; and (3) hydrologically correcting the elevations of coarsened MODFLOW cells to enforce integrated drainage through the stream network. Focusing on the third approach that is specific to the GSFLOW-GRASS toolbox (Section 3.2.2), the narrow and steep Water Canyon requires the same hydrologic corrections that were applied in the Shullcas case, above, to maintain downslope-integrated drainage. Under losing stream conditions, artificially increased channel elevation would steepen the hydraulic head gradient away from the channel and cause it to over-simulate water flow to the surrounding landscape. Therefore, hydrologic correction of the coarse MODFLOW grid is necessary to simulate appropriate head gradients and maintain water in cells, which is further required for any attempt to match stream-gauge records.**"*

***We have filled out "GSFLOW-GRASS capabilities" in Table 3 accordingly.***

The technical challenges identified for the third test case are (i) two-layer soil hydraulic conductivity, (ii) poorly integrated drainage, (iii) low relief, and (iv) large basin.

(i) The tasks associated to the first point (i.e., two-layer soil hydraulic conductivity) are not discussed in the main text.

*Thank you for pointing this out.  We now mention the two-layer soil hydraulic conductivity in the main text:*
> ***(p. 27 line 13-14)***
> ***"The flexible GSFLOW-GRASS input builder allows for easy implementation of two MODFLOW layers to represent an upper glacial till unit (low hydraulic conductivity) and the underlying fractured carbonate bedrock (higher hydraulic conductivity)."***

*We also **slightly shortened the caption text about it**, now that it is in the main text.*

(ii) and (iii) In my previous review I highlighted the problem that all the discussion for this test case is about an already existing GRASS feature (r.watershed). Therefore, one could argue that nothing new related to GSFLOW-GRASS has been discussed/presented in this test case. This issue will persist as long as authors do not expand the focus of their exercise. Authors also argue that "Most GIS tools will artificially fill these pits, but r.watershed's least-cost-path algorithm is able to route flow across them in a more realistic way". This is a quite interesting technical aspect to show with a practical example where you substantiate on relevant modeling issues using GSFLOW-GRASS. As it is right now the reader has no concrete results to get a better insight on what the authors state.

*To address the referee's comment that there was "nothing new" because we used the existing GRASS-GIS extension r.watershed: **our re-written Section 3.2.1** now explains that the r.watershed algorithm has never been used before with integrated hydrologic modeling, despite its superior performance compared to the local gradient-based tools used by others.  This should now make clear that our introduction of the least-cost flow algorithm (r.watershed) to integrated hydrologic modeling is in fact an innovative step.  The computational and performance specifications of the least-cost flow algorithm relative to other commonly used methods are now also fully documented in the rewritten Section 3.2.1 based on a thorough literature review, which addresses the referee's other request for more concrete aspects about the algorithm's advantages.*

*The actual section on the Cannon River example was heavily edited to now point to Section 3.2.1.; reiterate the fact that other domain builders for integrated hydrologic models use different flow-routing algorithms that can be problematic; and emphasize that we created various new GRASS extensions in order to integrate r.watershed with GSFLOW (so that we are not simply plug-and-chugging with an existing tool):*
> ***(p. 27 line 25-33)***
> ***"In such settings that lack integrated drainage, downslope flow-routing and "pit-filling" algorithms that are typically used to build hydrologic model domains***

*(e.g., Bhatt et al. 2014, Maxwell et al. 2017, Gardner et al. 2018) can fail or produce spurious results by inappropriately modifying the real topography. As described in Section 3.2, GSFLOW-GRASS determines surface-water flow using the GRASS GIS's efficient and accurate r.watershed extension, which implements a least-cost path algorithm designed to produce drainage networks that route flow in the long-range path of steepest descent regardless of the degree of local drainage integration. By using r.watershed alongside a set of new GRASS-GIS extensions that integrate it into the GSFLOW framework, GSFLOW-GRASS is able to automatically create a topologically correct and linked drainage network in settings that lack integrated drainage for hydrologic model simulations."*

(iv)What's the challenge associated to a large basin? Technically a small basin resolved at high resolution could be the same of a coarse-resolution large basin.

*Indeed, a coarse-resolution grid of a large basin has the same computational complexity of a fine-resolution grid of a small basin. The question, then, is how to get to a reasonable coarse-resolution grid of a large basin - and this is what we address with Cannon River.  We edited the text to clarify that as the largest of the examples watersheds, Cannon River takes advantage of the efficient irregular surface grid and a robust downscaling method of GSFLOW-GRASS:*

*(p. 27 line 16-20)*
*"Covering 3723 km$^2$, the Cannon River watershed is by far the largest of the three model implementations (Table 2) and benefits from the efficiency of the topographically based surface grid and the hydrologic robustness of the grid coarsening method in GSFLOW-GRASS. 17,455,046 flow-routing grid cells, each of which is 225 m$^2$ in area, were converted to only 610 irregular HRUs of >=10km$^2$ in area. For the groundwater domain, the elevation data were coarsened and hydrologically corrected to a 1 km regular MODFLOW grid."*

Other points:

1. Captions still contain extensive discussion of the results that should be moved in the main text.

*We wish to strike a balance between concise and informative captions and have maintained a moderate amount of the original text in the captions.  However, we do recognize that some of the original caption text had not been adequately incorporated into the main text, and we now include it in the main text and do shorten some of the captions.  **Changes were made throughout each of the three examples to reference all the figures.***

2. I disagree with this statement:

"Unlike PIHM, GSFLOW-GRASS employs regular groundwater grid cells that are distinct from the irregular surface units, which makes the integrated domain building more complicated but allows for more complex representation of the surface-water and aquifer systems."

PIHM implements a fully unstructured grid that allows for a detailed representation of surface-subsurface water interactions. See reference below:

D. Wang, Y. Liu, and M. Kumar (2018). Using nested discretization for a detailed yet computationally efficient simulation of local hydrology in a distributed hydrologic model. Scientific Reports.

*We appreciate the reviewer bringing this up and did not mean at all to imply that PIHM has fewer capabilities than GSFLOW. We only meant for this sentence to make clear that there are differences between the models, and that GSFLOW-GRASS' heterogeneous domain is more complicated to build but can be flexible. We have edited the sentence so that it no longer says "allows for more complex representation" and instead we now say that it "allows for flexible representation." The only form of complexity that could be easier represented with GSFLOW-GRASS than PIHM is vertical aquifer heterogeneity, since the MODFLOW component of GSFLOW-GRASS can have multiple layers whereas PIHM currently has a single vertically integrated saturated zone layer. However, we have no interest in comparing models in this manuscript - we in fact recognize that PIHM has a number of strengths relative to GSFLOW-GRASS. The full modified sentence is:*
> *(p. 24 line 14-p. 25 line 2)*
> ***"Unlike PIHM, GSFLOW-GRASS employs a regular three-dimensional groundwater grid that does not align with the irregular surface domain; this makes the integrated domain building more complicated but allows for a flexible representation of the surface-water and aquifer systems."***

*We also thank the referee for sharing the new PIHM reference - we had not yet seen it. We added a citation to it earlier in the paper, where we discuss PIHM:*
> *(p. 4 line 15-17)*

[revised manuscript text omitted]

---

## Author Response (AR4)

All of the reviewer's comments are copied here with gray highlighting.
Our responses are not highlighted. New text is in **boldface**.

We note that in the manuscript comparison file, it appears as though nearly the entire Examples section is changed. It is not. We reordered the three examples (as explained below, to present the new multi-configuration model tests with the Cannon River example early in the section), and latexdiff (the differencing program for latex) picks this up as almost all new text. There were substantive changes made within the reordered examples, and these are detailed in the below response.

I would like to thank the authors for their very detailed replies to my last round of comments. My overall impression is that their effort adds to the clarity and quality of some parts of manuscript (see new section 3.2.1 – section 3.2.2) while keeping partially solved some other major concerns that I had with the manuscript. In particular, as already expressed in my very first review and re-iterated over the two successive steps, the information content of Fig. 5-6-7 does not clearly reflect the capabilities and claimed features (that I do not doubt exist) of the proposed toolkit. In expressing my opinion I highlighted that such issues would have required some substantial work (including improvements of the figures) going beyond some pure text modification/clarification.

We thank the reviewer for their continued attention to our manuscript. In this revision, we have incorporated two new tests and accompanying figures (**Figures 3 and 7**): first is a set of simulations that compares results for various model domain configurations to show the robustness and efficiency of our toolkit; second is a test with and without the hydrologic correction step in the domain builder to show its effectiveness for ensuring accurate simulations. We believe that these major additions, which extend well beyond text modifications and include new figures, fully address the reviewer's main critique. Details about these new tests are included in our response to the reviewer's point #1. We have also made edits to address the reviewer's other comments.

1)      In the current form, the results demonstrate that the proposed automated workflow is able to generate a set of consistent input and output files starting from different physiographic settings. This is clearly an important and necessary achievement for the toolbox. However, this is not sufficient to demonstrate the robustness of the toolkit to handle the technical challenges identified for each test case. A rigorous approach to test the technical capabilities (as I have already suggested) would consist in the generation of different configurations to be ingested in the hydrological model. Indeed, in the implementation of the automated workflow the user has to define different "a priori" parameters values (e.g., thresholds value for the definition of the HRU, subsurface coarsening, etc.) that result in different configurations that eventually produce different results of the integrated hydrological model. This is in my opinion the way to show how the proposed toolkit could really guide users through important modeling steps. I enforce my opinion highlighting that statements like "…79 HRU cells were needed to capture nearly all the essential flow-routing information…"; "…GSFLOW-GRASS' hydrological correction to enforced integrated subsurface proved essential for preventing unrealistic results", "Early model tests for Shullcas showed that simple grid coarsening using the mean value of the elevations…" etc… are still rather vague and do not reflect the information content of Fig5-6-7.

We added two tests (and accompanying Figures 3 and 7) to the Examples section to rigorously demonstrate the capabilities and robustness of GSFLOW-GRASS.

In the first added test, we implemented the reviewer's suggested test of inputting different domain resolution configurations into the GSFLOW-GRASS workflow.  We chose to do this for the Cannon River watershed example, because its greatest area and highest resolution DEM of the three examples make it possible to test a large range of resolution choices.  We note that we have now reversed the order of  the examples so that Cannon River is first, in order to present this important test early.  The user makes a decision about two different domains, the surface irregular HRU's and the subsurface rectangular grid; the new test examines the implications of varying both.  The performance of the different configurations are evaluated based on root mean square error discharge (RMSE) (defined relative to the highest resolution run), the computational runtime of the GSFLOW execution, and the computational runtime of the domain-builder.  This test serves two purposes.  First, it demonstrates the advantages and trade-offs of implementing coarsened domains using GSFLOW-GRASS's topographically driven surface domain and hydrologically corrected subsurface domain. Second, it demonstrates the type of rigorous evaluations that can be easily carried out with the GSFLOW-GRASS toolkit.  The results are shown in the new Figure 7.  The new text in the Cannon River watershed example explaining this test is as follows:

**"The Cannon River watershed is by far the largest of the three model implementations and thus greatly benefits from the coarsened surface and subsurface domains (Table 2). To test the robustness and efficiency of GSFLOW-GRASS over different resolution configurations, we compared the accuracy and compute time of the Cannon River example case across one order of magnitude in threshold surface drainage area (for the HRU delineation) and two orders of magnitude in MODFLOW subsurface grid cell area, starting from the base case resolution shown in Figure 7. Note that the threshold surface drainage area increase was limited by the total watershed size. Figure 7A shows that coarsening the irregular HRU resolution results in little (at finer MODFLOW resolution) to negligible (at coarser MODFLOW resolution) increase in error, compared to coarsening the rectangular MODFLOW grid. This demonstrates that the accuracy of GSFLOW-GRASS' topographically based surface discretization is well-maintained even with large-sized cells. Over the 2 orders of magnitude increase in rectangular MODFLOW grid cell sizes, errors steadily grow to about 35%, but GSFLOW-GRASS' hydrological correction step (Section 3.2.2) helped prevent even greater errors. The trade-off for accuracy is compute time: GSFLOW runtime is much more sensitive to the MODFLOW resolution than to the surface-domain resolution (Figure 7B). However, the domain builder algorithm – which requires longer to compute times than the 5.5-year GSFLOW simulation for the Cannon River – is sensitive to both the surface drainage resolution and the**

**MODFLOW grid cell area, and it is even more sensitive to the surface drainage resolution (Figure 7C). GSFLOW-GRASS's fully automated surface drainage delineation thus allows users to overcome one of the most time-consuming obstacles to implementing integrated hydrologic models by constructing efficient and accurate irregular HRUs. The GSFLOW-GRASS toolkit makes it easy to carry out systematic GSFLOW configuration tests like this to assess model performance."**

In the second added test, we compare the simulation results with and without the hydrologic correction to demonstrate its importance for generating accurate results. This was implemented for the Shullcas watershed, because it is most acutely prone to domain-coarsening problems addressed by the correction step, due to its steep topography and high canyon walls. The new Figure 3 graphically shows the undesirable effect of water being trapped behind artificial "dams" without the hydrologic correction. The correction allows for the expected, continuous water table depths along the stream channel.

Also, we edited each of the specific sentences of concern raised by the reviewer (all located in the previous 3rd paragraph of the Schullcas example):
- Original text: "converting these into a far smaller number of larger computational surface cells (79 HRUs that are ≥1 km$^2$ in area) that convey the same fundamental surface-flow information."
    - Edits: We deleted the vague phrase "that convey the same fundamental surface-flow information" and instead reference the new multi-configuration test results (in the Cannon River example), which quantitatively demonstrated GSFLOW-GRASS' robustness and accuracy across surface domain resolutions.
- Original text: "GSFLOW-GRASS' hydrological correction to enforced integrated subsurface proved essential for preventing unrealistic results. Early model tests for Shullcas showed that simple grid coarsening using the mean value of the elevations…"
    - Edits: We realized that our repeated efforts to more clearly describe the hydrologic correction in words was not effective, and so we include the test with and without the correction. The results in Figure 3 unambiguously shows the improvement after the hydrologic correction. In the text, we deleted the vague phrase "unrealistic results" and now point to Figure 3 rather than reference "Eary model tests."
    - Edits: We also heavily edited Section 3.2.2 to clarify our explanation of the hydrologic correction step, which we do in part by presenting the new Figure 3 (rather than waiting until the Examples section to present Figure 3). Section 3.2.2 now reads as follows:
        > **"Following the completion of the surface-water domain, the next step is to build the groundwater domain. MODFLOW-NWT uses a rectangular finite-difference grid structure (Harbaugh, 2005; Niswonger et al., 2011). The cell size for this grid is selected by the user in the Settings file. It is often necessary to discretize the**

MODFLOW groundwater domain on a grid that is coarser than the DEM used for surface flow routing in order to increase computational efficiency while still allowing GSFLOW--GRASS to generate a complex surface-water network; the proper grid cell size depends on the size of the HRUs and the strength of the surface-water-groundwater coupling. v.gsflow.grid builds the MODFLOW grid as a set of GIS vector areas (Figure 1C) using the built-in v.mkgrid command. The resolution of this grid is approximately that desired by the user, with the constraint that the edges of each grid cell must align with the edges of each raster cell in the flow-routing DEM.

r.gsflow.hydrodem then performs a hydrologically-correct resampling of the original flow-routing DEM to the resolution of the MODFLOW grid cells. This resampling is required when users desire a MODFLOW grid that is coarser than the flow-routing DEM for computational feasibility. Without a hydrologically-correct resampling, MODFLOW cells would be assigned the overall mean elevation from the corresponding region of the flow-routing DEM. In this case, MODFLOW grid cell elevations may average across valley floors and valley walls, creating a bumpy river longitudinal profile that contains artificial dams (Figure 3A). With the hydrologic correction, MODFLOW cells that do not contain stream segments remain unchanged, but cells containing stream segments are assigned the mean elevation of only the river-channel cells in the flow-routing DEM. This enforces decreasing elevations down the drainage network, and Figure 3B demonstrates the resulting continuous flow through the catchment."

The full revised paragraph of the Shullcas example (now 2nd paragraph) reads:

"The simple hydrologic model based on the Shullcas watershed covers a large elevation range and uses a coarsened discretization based on an ASTER nominal 30 m resolution DEM (Tachikawa et al., 2011) (Table 2). Meteorological data were obtained from the Peruvian Meteorological Office (SENMAHI) online database. Located in the Andes, the Shullcas River Watershed serves as an apt testbed for examining the ability of GSFLOW-GRASS to represent surface-water-groundwater links in steep terrain. Representing flow in steep topography and narrow canyons calls for high resolution computations that can be infeasible to directly incorporate into an integrated hydrologic model. GSFLOW-GRASS solved this problem by converting high-resolution flow-routing information for Shullcas into a far smaller number of topographically defined computational surface cells and coarsened MODFLOW grid cells (see Table 2) - a strategy that the multiple-configuration Cannon River test

**demonstrated can generate an efficient and accurate discretization (Figure 7). The major challenge to domain coarsening further presented by Shullcas is its particular susceptibility to artificial "dams" due to its mountainous topography. These numerical artifacts occur when averaging elevations across flat valley floors and adjacent steep canyon walls, which can cause cells containing streams to be higher rather than lower than the surrounding cells on the MODFLOW grid. GSFLOW-GRASS's hydrological correction addresses this by enforcing integrated subsurface drainage (Section 3.2.2), thus preventing improper accumulation and, subsequently, lateral leakage of water behind "dammed" stream cells (Figure 3)."**

Finally, in response to the reviewer's concerns about Figures 6-8-9 (formally 5-6-7), we point out that those serve mostly to demonstrate the various GSFLOW-GRASS applications, which in our response to point #2 we argue is an important purpose of the three examples. For figures that demonstrate the robustness and efficiency of GSFLOW-GRASS - which we believe is what the reviewer seeks - we added the new Figures 3 and 7.

2)    In the discussion of each test case you should remove all the text (including also "Applications" column in Table 3) providing a generic description of the "potential" scientific issues (e.g., erosion) associated to the case study and "demonstrated" with GSFLOW-GRASS. These issues are not truly addressed in this work and all of these argumentations appear disconnected from the main objective of the work. Note here that the head paragraph in the "Examples" section contains already an exhaustive and general description of the three test cases.

The reviewer has helped us see that our previous manuscript version included more details than needed about the scientific applications. However, we respectfully argue that demonstrating the range of potential applications for GSFLOW-GRASS IS critical for proving its flexibility and value. The reviewer is most focused on the "technical" contribution of GSFLOW-GRASS, but we believe that some other readers may be more interested in its utility. The examples serve both purposes, as stated in the introductory section of our Examples: "Three example implementations demonstrate (1) the variety of hydrological processes and environments that can be explored using GSFLOW--GRASS, and (2) how the toolkit's GIS domain builder can handle diverse topographic settings, including those prone to problems with standard GIS stream network tools." To strike a balance between our views, we elected to significantly shorten the discussions about applications, but also kept in parts that motivate the need for integrated hydrological models and/or describe how the application demonstrates certain features of the toolkit. We chose to keep in the "Applications" column in Table 3, because this is the more concise format than the main text. We explain where we made cuts and consolidated text, and we justify the remaining text here:

- We eliminated almost all details about applications in the introductory portion of the "Examples" section - nearly an entire paragraph. We agree with the reviewer that it was somewhat repetitive to have the applications described both here and in each individual example section.  The only relevant sentences kept were: "The three examples all contain complex hydrology with interactions between surface water and groundwater and are exemplars of practical management concerns. Together they span a range of environments: high to low elevations, steep to low-gradient catchments, coastal to inland settings, tectonically active to cratonal, and with partially to fully integrated drainage. Their catchment areas range from 12.5 km$^2$ to 3723.0 km$^2$, covering the range of scales that GSFLOW was developed to simulate."

- Schullcas example: A sentence about discharge variability is removed from the last paragraph.  We kept in discussion about the groundwater-surface water interactions, because that motivates the use of an integrated hydrological modeling, as well as showcases our visualization scripts for showing spatially distributed processes.
- Santa Rosa example: In the last paragraph, 3 sentences are entirely or nearly entirely deleted.  We edited the remaining text to emphasize that the erosion application serves to demonstrate the toolkit's script for creating spatially distributed hydraulic conductivity and the toolkit's post-processing scripts for comparing precipitation and surface runoff time series.
- Cannon River example: We shortened the first paragraph by removing some details about the geomorphology and condensing the text.  We also shortened the last paragraph by removing a couple of sentences about climate and head gradient results and deleting details about infiltration.  We did keep in the comparison between simulated and observed discharge to demonstrate that GSFLOW-GRASS can produce reasonable simulations. We also made edits to better emphasize how the toolkit can facilitate in addressing the calibration needs highlighted by the comparison.  Lastly, we left in discussion about water quality threats in agricultural areas posed by groundwater-surface water interactions to motivate the need for integrated hydrologic models.

3)     In the last revised version authors cite the work of Gardner et al. (2018), which proposes a series of technical (pre-processing) solutions for the same integrated hydrological model (i.e., GSFLOW-Arcpy). As the work of Gardner et al. is now published and available online (it was not during the first, second, and third review iteration), I think it is quite important to:

- Better highlight the contribution (need) of GSFLOW-GRASS in the introduction.
- Have a dedicated section presenting the differences and similarities between the solutions implemented in both toolkit (i.e., GSFLOW-GRASS and GSFLOW-Arcpy).

We appreciate having the opportunity to explain the distinction between GSFLOW-GRASS and GSFOW-Arcpy.  We had taken out much of the discussion in response to the reviewer's earlier comment, but we agree that it is now appropriate to include it now that the paper on GSFLOW-Arcpy is out.

We added a new section at the end of the Introduction to explain that both GSFLOW-GRASS and GSFLOW-Arcpy are software packages that aid with creating inputs for GSFLOW, but that they have important differences. First is their domain structure - GSFLOW-GRASS creates topographically based surface water cells that are different than the subsurface grid, while GSFLOW-Arcpy creates regular rectangular surface grid cells that coincide exactly with the subsurface grid. The second is that GSFLOW-GRASS uses all open source programs and provides fully automated pre- and post-processing steps, while GSFLOW-Arcpy requires an ArcGIS license and the user must have a way to handle MODFLOW input files and post-processing separately. GSFLOW-Arcpy does provide support for accessing surface datasets for model inputs. Overall, GSFLOW-GRASS fills the needs of new model-users who seek a complete and fully automated package that does not require commercial software licenses, and of model-users working with steep and complex terrain that can be more efficiently covered by topographically based domain units.

Note that we do not attempt to promote GSFLOW-GRASS over the USGS's GSFLOW-Arcpy, because that is not our goal. The original development of the two toolkits began independently, but we learned of each other's work before both toolkits' completion. We quickly realized that these were complementary approaches that would (1) together reach more potential users based on their particular needs and backgrounds and (2) facilitate future rigorous testing to help resolve the debate on domain types. We then asked Rich Niswonger, one of the USGS developers of GSFLOW and GSFLOW-Arcpy, to join as a co-author to ensure that our toolkit would be valuable to the large community of USGS model users and not overly duplicate GSFLOW-Arcpy.

The added section to the Introduction is copied here:

[revised manuscript text omitted]

---

## Author Response (AR5)

*We thank the reviewer for their comments and have responded below. Reviewer comments are copied in roman text, and our responses are italicized and indented.*

I would like to thank the authors for the substantial effort in improving the manuscript. I have just two minor comments before recommending the manuscript for publication in GMD.

*The authors thank the reviewer for all of their comments.*

- The discussion of Figure 6 and Figure 7 is flipped in the main text. Please change the order of the figures for consistency.

*In Section 4, Figure 6 is first mentioned in the main text (p. 23 Line 2) prior to the first mention of Figure 7 (p. 24 line 19). Therefore, the figures are in the appropriate order, which we will maintain.*

- I appreciate the authors effort in generating the new Figure 7. I think it is quite illustrative. For that reason I would also add the same plot you're showing for the river discharge (i.e., Figure 7a) considering some subsurface model output (e.g., hydraulic head) at arbitrary points. Please include also a color scale for the plots.

*We appreciate the reviewer's suggestion to analyze hydraulic head and recognize that this could be a useful way to characterize the scale dependence of the general GSFLOW (or MODFLOW) model. However, this is not an appropriate method to analyze the specific GSFLOW-GRASS software, because of the way we constructed the domain-builder algorithm to most accurately correspond to the finest resolution topographic information from the DEM. This comment from the reviewer makes us realize that we had not sufficiently explained this beneficial feature of GSFLOW-GRASS domain-builder.*

*When the user specifies a desired grid cell size, GSFLOW-GRASS implements the closest grid spacing to the specified size that also meets two criteria. First, the MODFLOW grid cells must align with the edges of the grid cells in the flow-routing DEM in order to limit smearing of the original fine-resolution information. Second, the extent of the MODFLOW grid is set to conform to the watershed shape; the extent of an initial inner MODFLOW grid is set to match the first flow-routing DEM cell edge beyond the margins of the study watershed, and then three MODFLOW cells are added to each side to produce appropriate boundary conditions. The final domain configuration varies with the chosen resolution to ensure that the most compact -- and thus computationally efficient -- possible MODFLOW grid domain is created.*

*Because this topographically accurate and efficient algorithm design modifies the size of the MODFLOW grid cells and changes the extent of the MODFLOW grid itself, implementing the reviewer's suggested comparison of simulated hydraulic head across grid resolutions is difficult to impossible. Carrying out this comparison would require that coarse resolution grids have cell edges that perfectly align with cell edges of fine resolution grids, because unaligned grid cells spanning steep topographic gradients will contain very different and incompatible head values. Picking a set of perfectly aligned nested grids across resolutions cannot be readily done with our algorithm, which automatically adjusts cell sizes to accurately and efficiently correspond with topographic information in the DEM. Performing the suggested comparison test would thus require that we remove a feature that aids in the performance of GSFLOW-GRASS and would*

*not be a productive test.*

*We have now added the following description to Section 3.2.2 (p. 13, lines 20-25) to more clearly describe this feature of the domain-builder:*

> *"The resolution of this grid is approximately that desired by the user and meets two criteria designed to most accurately and efficiently capture the topographic information represented in the fine-scale DEM.  First, the grid size must allow for MODFLOW grid-cell edges to align perfectly with flow-routing DEM cell edges. Second, the grid size must be able to generate a domain that fits exactly within a tight-fitting bounding box snapped to the closest flow-routing DEM cell edges outside of the study watershed.  As a final step, this tight-fitted domain is padded by three cells on each side to ensure appropriate boundary-condition handling."*

*Regarding the color scale, we would like to point out that the changes in color exactly coincide with the labeled black contour lines in Figure 7.  The colors were added simply to enhance the plain contour plot and do not give additional information beyond the labelled contours.  We chose not to include a color scale because it would be redundant information to the labelled contour lines, and we wished to avoid unnecessary clutter in the figure.  We do recognize that the contour labels may have appeared small to the reviewer, prompting their request for the color scale.  So, to directly address this underlying concern, we increased the font size of the contour line labels.  We believe this should address any concerns about the readability of the figure.  We show the previous and current plots below for comparison (the different figures do not show up in the latex file-comparison).*

[revised manuscript text omitted]